# Multi-Player Zero-Sum Markov Games with Networked Separable Interactions

**Chanwoo Park**[♮]
MIT
cpark97@mit.edu

**Kaiqing Zhang**[♮]
University of Maryland, College Park
kaiqing@umd.edu

**Asuman Ozdaglar**
MIT
asuman@mit.edu

## Abstract

We study a new class of Markov games, *(multi-player) zero-sum Markov Games* with *Networked separable interactions* (zero-sum NMGs), to model the local interaction structure in non-cooperative multi-agent sequential decision-making. We define a zero-sum NMG as a model where the payoffs of the auxiliary games associated with each state are zero-sum and have some separable (i.e., polymatrix) structure across the neighbors over some interaction network. We first identify the necessary and sufficient conditions under which an MG can be presented as a zero-sum NMG, and show that the set of Markov coarse correlated equilibrium (CCE) collapses to the set of Markov Nash equilibrium (NE) in these games, in that the product of per-state marginalization of the former for all players yields the latter. Furthermore, we show that finding approximate Markov *stationary* CCE in infinite-horizon discounted zero-sum NMGs is PPAD-hard, unless the underlying network has a "star topology". Then, we propose fictitious-play-type dynamics, the classical learning dynamics in normal-form games, for zero-sum NMGs, and establish convergence guarantees to Markov stationary NE under a star-shaped network structure. Finally, in light of the hardness result, we focus on computing a Markov *non-stationary* NE and provide finite-iteration guarantees for a series of value-iteration-based algorithms. We also provide numerical experiments to corroborate our theoretical results.

## 1 Introduction

Nash Equilibrium (NE) has been broadly used as a solution concept in game theory, since the seminal works of [1, 2]. Perhaps equally important, NE is also deeply rooted in the prediction and analysis of *learning dynamics* in multi-agent strategic environments: it may appear as a natural outcome of many non-equilibrating learning processes of multiple agents interacting with each other [3, 4]. A prominent example of such learning processes is fictitious play (FP) [5, 6], in which myopic agents estimate the opponents' play using history, and then choose a best-response action (based on their payoff matrix) against this estimate, as if the opponents use it as their stationary strategy. The focus of these studies has initially been on the convergence to NE in zero-sum games (see [5, 6], and also [7, 4]), and in games with aligned objective (identical-interest and potential games, see [8]). Since then, FP has been shown to converge to NE in more important classes of games, including $2 \times n$ games [9, 10], "one-against-all" games [11], and zero-sum polymatrix games [12], justifying the prediction power of NE in learning in normal-form/matrix games.

Some of these results have recently been extended to the *stochastic game* (also known as the *Markov* game (MG)) setting, a model for multi-agent sequential decision-making with state transition dynamics, first introduced in [13]. In particular, [14, 15, 16, 17, 18] have studied best-response-type learning dynamics in two-player zero-sum MGs, and [19, 17] have studied that in multi-player

---

[♮]Alphabetical order.

37th Conference on Neural Information Processing Systems (NeurIPS 2023).

identical-interest games. Following the same path as studying matrix games, one natural question arises: *Are there other classes of MGs beyond two-player zero-sum and identical-interest cases that allow natural learning dynamics, e.g., fictitious play, to justify NE as the long-run emerging outcome?*

On the other end of the spectrum, it is well-known that for *general-sum* normal-form games, the special case of MGs without the state transition dynamics, *computing* an NE is intractable [20, 21]. Relaxed solution concepts as (coarse) correlated equilibrium ((C)CE) have thus been favored when it comes to equilibrium computation for general-sum, multi-player games [22, 3]. Encouragingly, when the interactions among players have some networked separable structure, also known as being *polymatrix*, computing NE may be made tractable even in multi-player settings. This has been instantiated in the seminal works [23, 24] in the normal-form game setting, which showed that any CCE collapses to the NE in such games when the payoffs are zero-sum. Thus, any algorithms that can efficiently compute the CCE in such games will lead to the efficient computation of the NE.

In fact, besides being of theoretical interest, multi-player zero-sum games with networked separable interactions also find a range of applications, including security games [24], fashion games [25, 26, 27], and resource allocation problems [28]. These examples, oftentimes, naturally involve some *state* transition that captures the dynamics of the evolution of the environment in practice. For example, in the security game, the protection level or immunity of a target increases as a function of the number of past attacks, leading to a smaller probability of an attack on the target being successful. Hence, it is imperative to study such multi-player zero-sum games with state transitions. As eluded in the recent results [29, 30], such a transition from *stateless* to *stateful* cases may not always yield straightforward and expected results, e.g., computing stationary CCE can be computationally intractable in stochastic games, in stark contrast to the normal-form case where CCE can be efficiently computed. This naturally prompts another question: *Are there other types of (multi-player) MGs that may circumvent the computational hardness of computing NE/CCE?*

In an effort to address these two questions, we introduce a new class of Markov games – *(multi-player) zero-sum Markov games with networked separable interactions* (zero-sum NMGs). We summarize our contributions as follows, and defer a more detailed literature review to Appendix A.

**Contributions.** First, we introduce a new class of non-cooperative Markov games: (multi-player) zero-sum MGs with Networked separable interactions (zero-sum NMGs), wherein the payoffs of the *auxiliary-games* associated with each state, i.e., the sum of instantaneous reward and expectation of any estimated state-value functions, possess the multi-player zero-sum and networked separable (i.e., polymatrix) structure as in [31, 28, 23, 24] for normal-form games, a strict generalization of the latter. We also provide structural results on the reward and transition dynamics of the game, as well as examples of this class of games. Specifically, for a Markov game to qualify as a zero-sum NMG, if and only if its reward function has the zero-sum polymatrix structure, and its transition dynamics is an *ensemble* of multiple single-controller transition dynamics that are sampled randomly at each state (see Remark 3 for more details). This transition dynamics covers the common ones in the MG literature, including the single-controller and turn-based dynamics. Second, we show that Markov CCE and Markov NE *collapse* in that the product of per-state marginal distributions of the former yields the latter, making it sufficient to focus on the former in equilibrium computation. We then show the PPAD-hardness [32] of computing the *Markov stationary* equilibrium, a natural solution concept in infinite-horizon discounted MGs, unless the underlying network has a star-topology. This is in contrast to the normal-form case where CCE is always computationally tractable. Third, we study the fictitious-play property [8] of zero-sum NMGs, showing that the fictitious-play dynamics [16, 19] converges to the Markov stationary NE, for zero-sum NMGs with a star-shaped network structure. Finally, in light of the hardness of computing stationary equilibria, we develop a series of value-iteration-based algorithms for computing a Markov *non-stationary* NE of zero-sum NMGs, with finite-iteration guarantees. We also provide numerical experiments to corroborate our theoretical results in Section G. We hope our results serve as a starting point for studying this networked separable interaction structure in non-cooperative Markov games.

**Notation.** For a real number $c$, we use $(c)_+$ to denote $\max\{c, 0\}$. For an event $\mathcal{E}$, we use $\mathbf{1}(\mathcal{E})$ to denote the indicator function such that $\mathbf{1}(\mathcal{E}) = 1$ if $\mathcal{E}$ is true, and $\mathbf{1}(\mathcal{E}) = 0$ otherwise. We define multinomial distribution with probability $(w_i)_{i \in \mathcal{N}}$ as $\text{Multinomial}\big((w_i)_{i \in \mathcal{N}}\big)$. We denote the uniform distribution over a set $\mathcal{S}$ as $\text{Unif}(\mathcal{S})$. We denote the Bernoulli distribution with probability $p$ as $\text{Bern}(p)$. The sgn function is defined as $\text{sgn}(x) = 2 \times \mathbf{1}(x \geq 0) - 1$. The KL-divergence between two probability distributions $p, q$ is denoted as $\text{KL}(p, q) = \mathbb{E}_p[\log(p/q)]$. For a graph $\mathcal{G} = (\mathcal{N}, \mathcal{E})$, we denote the set of neighboring nodes of node $i \in \mathcal{N}$ as $\mathcal{E}_i$ (without including $i$). The maximum norm of a matrix $X \in \mathbb{R}^{m \times n}$, denoted as $\|X\|_{\max}$, is defined as $\|X\|_{\max} := \max_{i \in [m], j \in [n]} |X_{i,j}|$.

## 2  Preliminaries

### 2.1  Markov games

We define a Markov game as a tuple $(\mathcal{N}, \mathcal{S}, \mathcal{A}, H, (\mathbb{P}_h)_{h \in [H]}, (r_{h,i})_{h \in [H], i \in \mathcal{N}}, \gamma)$, where $\mathcal{N} = [n]$ is the set of players, $\mathcal{S}$ is the state space with $|\mathcal{S}| = S$, $\mathcal{A}_i$ is the action space for player $i$ with $|\mathcal{A}_i| = A_i$ and $\mathcal{A} = \prod_{i \in \mathcal{N}} \mathcal{A}_i$, $H \leq \infty$ is the length of the horizon, $\mathbb{P}_h : \mathcal{S} \times \mathcal{A} \to \Delta(\mathcal{S})$ captures the state transition dynamics at timestep $h$, $r_{h,i} \in [0, R]$ is the reward function for player $i$ at timestep $h$, bounded by some $R > 0$, and $\gamma \in (0, 1]$ is a discount factor. An MG with a finite horizon ($H < \infty$) is also referred to as an *episodic* MG, while an MG with an infinite horizon ($H = \infty$) and $\gamma < 1$ is referred to as an infinite-horizon $\gamma$-discounted MG. When $H = \infty$, we will consider the transition dynamics and reward functions, denoted by $\mathbb{P}$ and $(r_i)_{i \in \mathcal{N}}$, respectively, to be independent of $h$. Hereafter, we may use *agent* and *player* interchangeably.

**Policy.**  Consider the stochastic Markov policy for player $i$, denoted by $\pi_i$, as $\pi_i := \{\pi_{h,i} : \mathcal{S} \to \Delta(\mathcal{A}_i)\}_{h \in [H]}$. A *joint* Markov policy is a policy $\pi := \{\pi_h : \mathcal{S} \to \Delta(\mathcal{A})\}_{h \in [H]}$, where $\pi_h : \mathcal{S} \to \Delta(\mathcal{A})$ decides the joint action of all players that can be potentially correlated. A joint Markov policy is a *product* Markov policy if $\pi_h : \mathcal{S} \to \prod_{i \in \mathcal{N}} \Delta(\mathcal{A}_i)$ for all $h \in [H]$, and is denoted as $\pi = \pi_1 \times \pi_2 \times \cdots \times \pi_n$. When the policy is independent of $h$, the policy is called a *stationary* policy. We let $\boldsymbol{a}_h$ denote the joint action of all agents at timestep $h$. Unless otherwise noted, we will work with Markov policies throughout. We denote $\pi(s) \in \Delta(\mathcal{A})$ as the joint policy at state $s \in \mathcal{S}$.

**Value function.**  For player $i$, the value function under joint policy $\pi$, at timestep $h$ and state $s_h$ is defined as $V_{h,i}^\pi(s_h) := \mathbb{E}_\pi \big[ \sum_{h'=h}^H \gamma^{h'-h} r_{h',i}(s_{h'}, \boldsymbol{a}_{h'}) \,\big|\, s_h \big]$, which denotes the expected cumulative reward for player $i$ at step $h$ if all players adhere to policy $\pi$. We also define $V_{h,i}^\pi(\rho) := \mathbb{E}_{s_h \sim \rho}[V_{h,i}^\pi(s_h)]$ for some state distribution $\rho \in \Delta(\mathcal{S})$. We denote the $Q$-function for the $i$-th player under policy $\pi$, at step $h$ and state $s_h$ as $Q_{h,i}^\pi(s_h, \boldsymbol{a}_h) := \mathbb{E}_\pi \big[ \sum_{h'=h}^H \gamma^{h'-h} r_{h',i}(s_{h'}, \boldsymbol{a}_{h'}) \,\big|\, s_h, \boldsymbol{a}_h \big]$, which determines the expected cumulative reward for the $i$-th player at step $h$, when starting from the state-action pair $(s_h, \boldsymbol{a}_h)$. For the infinite-horizon discounted setting, we also use $V_i^\pi$ and $Q_i^\pi$ to denote $V_{1,i}^\pi$ and $Q_{1,i}^\pi$ for short, respectively.

**Approximate equilibrium.**  Define an $\epsilon$-approximate *Markov perfect Nash equilibrium* as a product policy $\pi$, which satisfies $\max_{i \in \mathcal{N}} \max_{\mu_i \in (\Delta(\mathcal{A}_i))^{|\mathcal{S}| \times H}} (V_{h,i}^{\mu_i, \pi_{-i}}(\rho) - V_{h,i}^\pi(\rho)) \leq \epsilon$ for *all* $\rho \in \Delta(\mathcal{S})$ and $h \in [H]$, where $\pi_{-i}$ represents the marginalized policy of all players except player $i$. Define an $\epsilon$-approximate *Markov coarse correlated equilibrium* as a joint policy $\pi$, which satisfies $\max_{i \in \mathcal{N}} \max_{\mu_i \in (\Delta(\mathcal{A}_i))^{|\mathcal{S}| \times H}} (V_{h,i}^{\mu_i, \pi_{-i}}(\rho) - V_{h,i}^\pi(\rho)) \leq \epsilon$ for *all* $\rho \in \Delta(\mathcal{S})$ and $h \in [H]$. In the infinite-horizon setting, they can be equivalently defined as satisfying $\max_{s \in \mathcal{S}} \max_{i \in \mathcal{N}} \max_{\mu_i \in (\Delta(\mathcal{A}_i))^{|\mathcal{S}|}} (V_i^{\mu_i, \pi_{-i}}(s) - V_i^\pi(s)) \leq \epsilon$. If the above conditions only hold for certain $\rho$ and $h = 1$, we refer to them as Markov *non-perfect* NE and CCE, respectively. Unless otherwise noted, we hereafter focus on Markov perfect equilibria, and sometimes refer to them simply as *Markov equilibria* when it is clear from the context. In the infinite-horizon setting, if additionally, the policy is *stationary*, then they are referred to as a *Markov stationary* NE and CCE, respectively.

### 2.2  Multi-player zero-sum games with networked separable interactions

As a generalization of *two-player* zero-sum matrix games, *(multi-player)* zero-sum polymatrix games have been introduced in [31, 28, 23, 24]. A *polymatrix game*, also known as a *separable network game* is defined by a tuple $(\mathcal{G} = (\mathcal{N}, \mathcal{E}_r), \mathcal{A} = \prod_{i \in \mathcal{N}} \mathcal{A}_i, (r_{i,j})_{(i,j) \in \mathcal{E}_r})$. Here, $\mathcal{G}$ is an undirected connected graph where $\mathcal{N} = [n]$ denotes the set of players and $\mathcal{E}_r \subseteq \mathcal{N} \times \mathcal{N}$ denotes the set of edges describing the rewards' networked structures, where the graph neighborhoods represent the interactions among players. For each edge, a two-player game is defined for players $i$ and $j$, with action sets $\mathcal{A}_i$ and $\mathcal{A}_j$, and reward functions $r_{i,j} : \mathcal{A}_i \times \mathcal{A}_j \to \mathbb{R}$, and similarly for $r_{j,i}$. The reward for player $i$ for a given joint action $\boldsymbol{a} = (a_i)_{i \in \mathcal{N}} \in \prod_{i \in \mathcal{N}} \mathcal{A}_i$ is calculated as the sum of the rewards for all edges involving player $i$, that is, $r_i(\boldsymbol{a}) = \sum_{j:(i,j) \in \mathcal{E}_r} r_{i,j}(a_i, a_j)$. To be consistent with our terminology later, hereafter, we also refer to such games as *(multi-player) Games with Networked separable interactions (NGs)*.

In a *zero-sum* polymatrix game (i.e., a (multi-player) *zero-sum* Game with Networked separable interactions (zero-sum NG)), the sum of rewards for all players at any joint action $\boldsymbol{a} = (a_i)_{i \in \mathcal{N}} \in \prod_{i \in \mathcal{N}} \mathcal{A}_i$ equals zero, i.e., $\sum_{i \in \mathcal{N}} r_i(\boldsymbol{a}) = 0$. One can define the policy of agent $i$, i.e., $\pi_i \in \Delta(\mathcal{A}_i)$, so that the agent takes actions by sampling $a_i \sim \pi_i(\cdot)$. Note that $\pi_i$ can be viewed as the reduced case of the policy defined in Section 2.1 when $\mathcal{S} = \emptyset$ and $H = 1$. The expected reward for player $i$

under $\pi$ can then be computed as:

$$r_i(\pi) := \sum_{j:(i,j)\in\mathcal{E}_r} \sum_{a_i\in\mathcal{A}_i, a_j\in\mathcal{A}_j} r_{i,j}(a_i, a_j)\, \pi_i(a_i)\pi_j(a_j) = \pi_i^\intercal r_i \pi, \tag{1}$$

where $r_i$ denotes the matrix $r_i := (r_{i,1}, \dots, r_{i,(i-1)}, \mathbf{0}, r_{i,(i+1)}, \dots, r_{i,n}) \in \mathbb{R}^{|\mathcal{A}_i| \times \sum_{i\in\mathcal{N}}|\mathcal{A}_i|}$ and $\pi := (\pi_1^\intercal, \pi_2^\intercal, \dots, \pi_n^\intercal)^\intercal \in \mathbb{R}^{\sum_{i\in\mathcal{N}}|\mathcal{A}_i|}$. We define $r := (r_1^\intercal, r_2^\intercal, \dots, r_n^\intercal)^\intercal \in \mathbb{R}^{\sum_{i\in\mathcal{N}}|\mathcal{A}_i| \times \sum_{i\in\mathcal{N}}|\mathcal{A}_i|}$. Then in this case we have $\sum_{i\in\mathcal{N}} r_i(\pi) = 0$ for any policy $\pi$. See more prominent application examples of zero-sum polymatrix games in [23, 24].

# 3 Multi-Player (Zero-Sum) MGs with Networked Separable Interactions

We now introduce our model of multi-player zero-sum MGs with networked separable interactions.

## 3.1 Definitions

**Definition 1.** An infinite-horizon $\gamma$-discounted MG is called a *(multi-player) MG with Networked separable interactions (NMG)* characterized by a tuple $(\mathcal{G} = (\mathcal{N}, \mathcal{E}_Q), \mathcal{S}, \mathcal{A}, \mathbb{P}, (r_i)_{i\in\mathcal{N}}, \gamma)$ if for any function $V : \mathcal{S} \to \mathbb{R}$, defining $Q_i^V(s, a) := r_i(s, a) + \gamma \sum_{s'\in\mathcal{S}} \mathbb{P}(s' \mid s, a)V(s')$, there exist a set of functions $(Q_{i,j}^V)_{(i,j)\in\mathcal{E}_Q}$ and an undirected connected graph $\mathcal{G} = (\mathcal{N}, \mathcal{E}_Q)$ such that $Q_i^V(s, a) = \sum_{j\in\mathcal{E}_{Q,i}} Q_{i,j}^V(s, a_i, a_j)$ holds for every $i \in \mathcal{N}$, $s \in \mathcal{S}$, $a \in \mathcal{A}$, where $\mathcal{E}_{Q,i}$ denotes the neighbors of player $i$ induced by the edge set $\mathcal{E}_Q$ (without including $i$). When it is clear from the context, we represent the NMG tuple simply as $\mathcal{G} = (\mathcal{N}, \mathcal{E}_Q)$.

A finite-horizon MG is called a *(multi-player) MG with Networked separable interactions* if for any set of functions $V := \{V_h\}_{h\in[H+1]}$ where $V_h : \mathcal{S} \to \mathbb{R}$, defining $Q_{h,i}^V(s, a) := r_{h,i}(s, a) + \gamma \sum_{s'\in\mathcal{S}} \mathbb{P}_h(s' \mid s, a)V_{h+1}(s')$, there exist a set of functions $(Q_{h,i,j}^V)_{(i,j)\in\mathcal{E}_Q, h\in[H]}$ such that $Q_{h,i}^V(s, a) = \sum_{j\in\mathcal{E}_{Q,i}} Q_{h,i,j}^V(s, a_i, a_j)$ holds for every $i \in \mathcal{N}$, $s \in \mathcal{S}$, $h \in [H]$, $a \in \mathcal{A}$.

A (multi-player) NMG is called a *(multi-player) zero-sum MG with Networked separable interactions (zero-sum NMG)* if additionally $(\mathcal{G} = (\mathcal{N}, \mathcal{E}_Q), \mathcal{A} = \prod_{i\in\mathcal{N}} \mathcal{A}_i, (r_{i,j}(s) := Q_{i,j}^{\mathbf{0}}(s))_{(i,j)\in\mathcal{E}_Q})$ forms a zero-sum NG for all $s \in \mathcal{S}$ in the infinite-horizon $\gamma$-discounted case, or $(\mathcal{G} = (\mathcal{N}, \mathcal{E}_Q), \mathcal{A} = \prod_{i\in\mathcal{N}} \mathcal{A}_i, (r_{h,i,j}(s) := Q_{h,i,j}^{\mathbf{0}}(s))_{(i,j)\in\mathcal{E}_Q})$ forms a zero-sum NG for all $s \in \mathcal{S}$ and $h \in [H]$ in the finite-horizon case.

Regarding the assumption that the above conditions hold under any (set of) functions $V$, one may understand this as a structural requirement to inherit the polymatrix structure in the Markov game case. It is natural since $\{Q_i^V\}_{i\in\mathcal{N}}$ would play the role of the *payoff matrix* in the normal-form case, when value-(iteration) based algorithms are used to solve the MG. As our hope is to exploit the networked structure in the payoff matrices to develop efficient algorithms for solving such MGs, if we do not know *a priori* which value function estimate $V$ will be encountered in the algorithm update, the networked structure may easily break if we do not assume them to hold for *all* possible $V$. Moreover, such a definition easily encompasses the normal-form case, by preserving the polymatrix structure of the *reward functions* (when substituting $V$ to be a zero function). Some alternative definition (see Remark 2) may not necessarily preserve the polymatrix structure of even the reward functions in a consistent way (see Section B.4 for a concrete example). We thus focus on Definition 1, which at least covers the polymatrix structure of the reduced case regarding only reward functions.

Indeed, such a networked structure in Markov games may be fragile. We now propose both sufficient and *necessary* conditions for the reward function's structure and the transition dynamics of the MG, to be an NMG. Here we focus on the infinite-horizon discounted setting for a simpler exposition. For finite-horizon cases, a similar statement holds, which is deferred to Appendix B. We also defer the full statement and proof of the following result to Appendix B. We first introduce the definition of decomposability and the set $\mathcal{N}_C$, which will be used in establishing the conditions. For a graph $\mathcal{G} = (\mathcal{N}, \mathcal{E})$, we define $\mathcal{N}_C := \{i \mid (i,j) \in \mathcal{E} \text{ for all } j \in \mathcal{N}\}$, which may be an empty set if no such node $i$ exists.

**Definition 2** (Decomposability). A non-negative function $f : X^{|\mathcal{D}|} \to \mathbb{R}^+ \cup \{0\}$ is decomposable with respect to a set $\mathcal{D} \neq \emptyset$ if there exists a set of non-negative functions $(f_i)_{i\in\mathcal{D}}$ with $f_i : X \to \mathbb{R}^+ \cup \{0\}$, such that $f(x) = \sum_{i\in\mathcal{D}} f_i(x_i)$ holds for any $x \in X^{|\mathcal{D}|}$. A non-negative function $f : X^{|\mathcal{D}|} \to \mathbb{R}^+ \cup \{0\}$ is decomposable with respect to a set $\mathcal{D} = \emptyset$, if there exists a non-negative constant $f_o$ such that $f(x) = f_o$ holds for any $x \in X^{|\mathcal{D}|}$.

**Proposition 1.** For a given graph $\mathcal{G} = (\mathcal{N}, \mathcal{E}_Q)$, an MG $(\mathcal{N}, \mathcal{S}, \mathcal{A}, \mathbb{P}, (r_i)_{i \in \mathcal{N}}, \gamma)$ with more than two players is an NMG with respect to $\mathcal{G}$ if and only if: (1) $r_i(s, a_i, \cdot)$ **is decomposable with respect to** $\mathcal{E}_{Q,i}$ for each $i \in \mathcal{N}, s \in \mathcal{S}, a_i \in \mathcal{A}_i$, i.e., $r_i(s, \boldsymbol{a}) = \sum_{j \in \mathcal{E}_{Q,i}} r_{i,j}(s, a_i, a_j)$ for a set of functions $\{r_{i,j}(s, a_i, \cdot)\}_{j \in \mathcal{E}_{Q,i}}$, and (2) **the transition dynamics** $\mathbb{P}(s' \mid s, \cdot)$ **is decomposable with respect to** the set $\mathcal{N}_C$ of this $\mathcal{G}$, i.e., $\mathbb{P}(s' \mid s, \boldsymbol{a}) = \sum_{i \in \mathcal{N}_C} \mathbb{F}_i(s' \mid s, a_i)$ for a set of functions $\{\mathbb{F}_i(s' \mid s, \cdot)\}_{i \in \mathcal{N}_C}$ if $\mathcal{N}_C \neq \emptyset$, or $\mathbb{P}(s' \mid s, \boldsymbol{a}) = \mathbb{F}_o(s' \mid s)$ for some constant function (of $\boldsymbol{a}$) $\mathbb{F}_o(s' \mid s)$ if $\mathcal{N}_C = \emptyset$. Moreover, an MG qualifies as a zero-sum NMG if and only if it satisfies an additional condition: the NG, characterized by $(\mathcal{G}, \mathcal{A}, (r_{i,j}(s))_{(i,j) \in \mathcal{E}_Q})$, is a zero-sum NG for all $s \in \mathcal{S}$. In the case of two players, every (zero-sum) Markov game becomes a (zero-sum) NMG.

**Remark 1** (Stronger sufficient condition). We note that for an MG $(\mathcal{N}, \mathcal{S}, \mathcal{A}, \mathbb{P}, (r_i)_{i \in \mathcal{N}}, \gamma)$, if for every agent $i$, $r_i(s, a_i, \cdot)$ is decomposable with respect to some $\mathcal{E}_r \subseteq \mathcal{E}_Q$, and $\mathbb{P}(s' \mid s, \cdot)$ is decomposable with respect to some $\mathcal{N}_P \subseteq \mathcal{N}_C$, then one can still prove the *if* part, i.e., there exists some $\mathcal{G} = (\mathcal{N}, \mathcal{E}_Q)$ such that the game is an NMG with respect to this $\mathcal{G}$. See Figure 1 for the illustration. This is because by our definition, being decomposable with respect to a subset implies being decomposable with respect to a larger set, as one can choose the functions $f_i$ for the $i$ in the complement of the subset to be simply zero. We chose to state as in Proposition 1 just for the purpose of presenting both the *if* and *only if* conditions in a concise and unified way.

**Remark 2** (An alternative NMG definition). Another reasonable definition of NMG may be as follows: if for any *policy* $\pi$, there exist a set of functions $(Q_{i,j}^\pi)_{(i,j) \in \mathcal{E}_Q}$ and an undirected connected graph $\mathcal{G} = (\mathcal{N}, \mathcal{E}_Q)$ such that $Q_i^\pi(s, \boldsymbol{a}) = \sum_{j \in \mathcal{E}_{Q,i}} Q_{i,j}^\pi(s, a_i, a_j)$ holds for every $i \in \mathcal{N}, s \in \mathcal{S}$, $\boldsymbol{a} \in \mathcal{A}$. Note that such a definition can be useful in developing *policy*-based algorithms (while Definition 1 is more amenable to developing *value*-based algorithms), e.g., policy iteration, policy gradient, actor-critic methods, where the $Q$-value under certain *policy* $\pi$ will appear in the updates and may need to preserve certain decomposability structure, for any policy $\pi$ encountered in the algorithm updates. However, in this case, we cannot always guarantee the decomposability of $\mathbb{P}(s' \mid s, \cdot)$ or $r_i(s, a_i, \cdot)$. For example, if we assume that $r_i(s, \boldsymbol{a}) = 0$ for every $i \in \mathcal{N}, s \in \mathcal{S}, \boldsymbol{a} \in \mathcal{A}$, then $Q_i^\pi(s, \boldsymbol{a}) = \sum_{j \in \mathcal{E}_{Q,i}} 0$ and thus $Q_i^\pi(s, a_i, \cdot)$ is always decomposable regardless of $\mathbb{P}(s' \mid s, \boldsymbol{a})$. However, interestingly, we can show that the decomposability of the transition dynamics and the reward function as in Proposition 1 can still be guaranteed, as long as some *degenerate* cases as above do not occur. In particular, if there exist no $i \in \mathcal{N}$ and $s \in \mathcal{S}$ such that $Q_i^\pi(s, \boldsymbol{a})$ is a constant function of $\boldsymbol{a}$ for any $\pi$, then the results in Proposition 1 and hence after still hold. We defer a detailed discussion on this alternative definition to Appendix B.

**Remark 3** (Implication of decomposable transition dynamics). For an MG to be an NMG, by Proposition 1 the transition dynamics should be decomposable, i.e., $\mathbb{P}(\cdot \mid s, \boldsymbol{a}) = \sum_{j \in \mathcal{N}_C} \mathbb{F}_j(\cdot \mid s, a_j)$ or $\mathbb{P}(\cdot \mid s, \boldsymbol{a}) = \mathbb{F}_o(s' \mid s)$. We first focus on the discussion of the former case. Define $w_j(s, a_j) := \sum_{s' \in \mathcal{S}} \mathbb{F}_j(s' \mid s, a_j)$. If we fix the value of $s$ and $a_{-j}$, then $w_j(s, a_j)$ has to be the same for different values of $a_j$ due to the fact $\sum_{j \in \mathcal{N}_C} w_j(s, a_j) = 1$. Also note that by definition, $w_j(s, a_j)$ does not depend on the choice of this fixed $a_{-j}$. Therefore, such a $w_j(s, a_j)$ can be written as $w_j(s)$, where $w_j(s) = \sum_{s' \in \mathcal{S}} \mathbb{F}_j(s' \mid s, a_j)$ for all $a_j \in \mathcal{A}_j$. We can thus rewrite $(\mathbb{F}_j)_{j \in \mathcal{N}_C}$ using some actual probability distributions $(\mathbb{P}_j)_{j \in \mathcal{N}_C}$, such that if $w_j(s) \neq 0$, then we rewrite $\mathbb{F}_j$ as $\mathbb{F}_j(s' \mid s, a_j) = w_j(s) \frac{\mathbb{F}_j(s' \mid s, a_j)}{w_j(s)} = w_j(s) \mathbb{P}_j(s' \mid s, a_j)$, and if $w_j(s) = 0$, we rewrite $\mathbb{F}_j$ as $\mathbb{F}_j(s' \mid s, a_j) = w_j(s) \mathbb{P}_j(s' \mid s, a_j)$ for an arbitrary probability distribution $\mathbb{P}_j(\cdot \mid s, a_j)$. Notice that $\sum_{s' \in \mathcal{S}} \mathbb{P}_j(s' \mid s, a_j) = 1$ for any $j \in \mathcal{N}_C$. Then, the decomposable transition dynamics can be represented as $\mathbb{P}(\cdot \mid s, \boldsymbol{a}) = \sum_{j \in \mathcal{N}_C} w_j(s) \mathbb{P}_j(\cdot \mid s, a_j)$, i.e., an *ensemble* of the transition dynamics that is only controlled by single controllers. The model's transition dynamics thus act according to the following two steps: (1) sampling the controller according to the distribution $\text{Multinomial}\big((w_i(s))_{i \in \mathcal{N}_C}\big)$, and (2) transitioning the state following the sampled controller's dynamics. Such a model has also been investigated under the name of transition dynamics with *additive structures* in [33]. Note that our model is more general and thus covers the single-controller MG setting [34], where there is only one agent controlling the transition dynamics at *all* states. It also covers the setting of turn-based MGs [34], where in each round, depending on the current state $s$, the transition dynamics is by turns affected by only one of the agents. This can be captured by the proper choice of $(w_i(s))_{i \in \mathcal{N}_C}$ that takes value 1 only for one agent at each state $s$ (while takes value 0 for all other non-controller agents at each state $s$). Additionally, the second case where $\mathbb{P}(\cdot \mid s, \boldsymbol{a}) = \mathbb{F}_o(s' \mid s)$ corresponds to the one with no ensemble of controller agents.

**Proposition 2** (Decomposition of $(Q_i^V)_{i \in \mathcal{N}}$). For an infinite-horizon $\gamma$-discounted NMG with $\mathcal{G} = (\mathcal{N}, \mathcal{E}_Q)$ such that $\mathcal{N}_C \neq \emptyset$, if we know that $\mathbb{P}(s' \mid s, \boldsymbol{a}) = \sum_{i \in \mathcal{N}_C} \mathbb{F}_i(s' \mid s, a_i)$, and

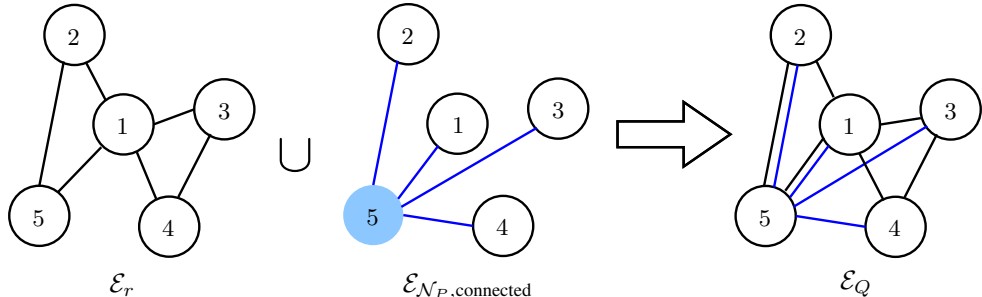

Figure 1: Relationship between $\mathcal{E}_r$, $\mathcal{E}_{\mathcal{N}_P,\text{connected}}$, and $\mathcal{E}_Q$. Here, we define $\mathcal{E}_{\mathcal{N}_P,\text{connected}} := \{(i,j) \mid i \in \mathcal{N}_P \text{ or } j \in \mathcal{N}_P, i \neq j\}$. $r_i(s, a_i, \cdot)$ is decomposable with respect to $\mathcal{E}_{r,i}$ for all $i \in \mathcal{N}$, and $\mathbb{P}(s' \mid s, \cdot)$ is decomposable with respect to $\mathcal{N}_P$ (See Remark 1). The transition dynamics $\mathbb{P}$ is expressed as the ensemble of controllers in the set $\mathcal{N}_P = \{5\}$ while the $\mathcal{N}_C$ in this case is $\{1, 5\}$.

$r_i(s, \boldsymbol{a}) = \sum_{j \in \mathcal{E}_{Q,i}} r_{i,j}(s, a_i, a_j)$ for some $\{\mathbb{F}_i\}_{i \in \mathcal{N}_C}$ and $\{r_{i,j}\}_{(i,j) \in \mathcal{E}_Q}$, then the $Q_{i,j}^V$ given in Definition 1 can be represented as

$$Q_{i,j}^V(s, a_i, a_j) = r_{i,j}(s, a_i, a_j) + \sum_{s' \in \mathcal{S}} \gamma \left( \mathbf{1}(j \in \mathcal{N}_C)\mathbb{F}_j(s' \mid s, a_j) + \mathbf{1}(i \in \mathcal{N}_C)\lambda_{i,j}(s)\mathbb{F}_i(s' \mid s, a_i) \right) V(s')$$

for any non-negative $(\lambda_{i,j}(s))_{(i,j) \in \mathcal{E}_Q}$ such that $\sum_{j \in \mathcal{E}_{Q,i}} \lambda_{i,j}(s) = 1$, for all $i \in \mathcal{N}$ and $s \in \mathcal{S}$. We call it the *canonical* decomposition of $\{Q_i^V\}_{i \in \mathcal{N}}$ when $Q_{i,j}^V$ can be represented as above with $\lambda_{i,j}(s) = 1/|\mathcal{E}_{Q,i}|$ for $j \in \mathcal{E}_{Q,i}$. The case with $\mathcal{N}_C = \emptyset$ is deferred to Appendix B.

We introduce this canonical decomposition since the representation of $Q_{i,j}^V$ is in general not unique, and we may use this canonical form to simplify the algorithm design later.

### 3.2 Examples of multi-player (zero-sum) NMGs

We now provide several examples of (multi-player) MGs with networked separable interactions here and in Section B.5.

**Example 1 (Markov fashion games).** Fashion games are an intriguing class of games [25, 26, 27] that plays a vital role not only in Economics theory but also in practice. A fashion game is a networked extension of the Matching Pennies game, in which each player has the action space $\mathcal{A}_i = \{-1, +1\}$, which means *light* and *dark* color fashions, respectively, for example. There are two types of players: *conformists* ($\mathfrak{c}$), who prefer to conform to their neighbors' fashion (action), and *rebels* ($\mathfrak{r}$), who prefer to oppose their neighbors' fashion (action). Such interactions with the neighbors are exactly captured by polymatrix games. We denote the interaction network between players as $\mathcal{G} = (\mathcal{N}, \mathcal{E})$.

Such a game naturally involves the following state transition dynamics: we introduce the state $s \in \mathcal{S} = \mathbb{Z}$ by setting $s_0 = 0$ and $s_{t+1} \sim s_t + \text{Unif}((a_{t,c})_{c \in \mathcal{C}})$, which indicates the *fashion trend* where $\mathcal{C} \subseteq \mathcal{N}$ is the set of influencers. The fashion trend favors either light or dark colors if $s \geq 0$ or $s < 0$, respectively. We can think of dynamics as the impact of the influencers on the fashion trend at time $t$. For each $(s, \boldsymbol{a})$, the reward function for player $i$, depending on whether she is a conformist or a rebel, are defined as $r_{\mathfrak{c},i}(s, \boldsymbol{a}) = \sum_{j \in \mathcal{E}_i} r_{\mathfrak{c},i,j}(s, a_i, a_j) = \sum_{j \in \mathcal{E}_i} (\frac{1}{|\mathcal{E}_i|} \mathbf{1}(\text{sgn}(s) = a_i) + \mathbf{1}(a_i = a_j))$ and $r_{\mathfrak{r},i}(s, \boldsymbol{a}) = \sum_{j \in \mathcal{E}_i} r_{\mathfrak{r},i,j}(s, a_i, a_j) = \sum_{j \in \mathcal{E}_i} (\frac{1}{|\mathcal{E}_i|} \mathbf{1}(\text{sgn}(s) \neq a_i) + \mathbf{1}(a_i \neq a_j))$, respectively. This is an NMG as defined in Definition 1. Moreover, if the conformists and rebels constitute a bipartite graph, i.e., the neighbors of a conformist are all rebels and vice versa, it becomes a multi-player constant-sum MG with networked separable interactions, and we can subtract the constant offset to make it a zero-sum NMG.

### 3.3 Relationship between CCE and NE in zero-sum NMGs

A well-known property for zero-sum NGs is that marginalizing a CCE leads to a NE, which makes it computationally tractable to find the NE [23, 24]. We now provide below a counterpart in the Markov game setting, and provide a more detailed statement of the result in Appendix B.

**Proposition 3.** Given an $\epsilon$-approximate Markov CCE of an infinite-horizon $\gamma$-discounted zero-sum NMG, marginalizing it at each state results in an $\frac{(n+1)}{(1-\gamma)}\epsilon$-approximate Markov NE of the zero-sum NMG. The same argument also holds for the finite-horizon episodic setting with $(1 - \gamma)^{-1}$ being replaced by $H$.

This result holds for both stationary and non-stationary $\epsilon$-approximate Markov CCEs. We defer the proof of Proposition 3 to Appendix B. This proposition suggests that if we can have some algorithms to find an approximate Markov CCE for a zero-sum NMG, we can obtain an approximate Markov NE by marginalizing the approximate CCE at each state. We also emphasize that the *Markovian* property of the equilibrium policies is important for the result to hold. As a result, the learning algorithm in [30], which learns an approximate Markov *non-stationary* CCE with polynomial time and samples, may thus be used to find an approximate Markov *non-stationary* NE in zero-sum NMGs. However, as the focus of [30] was the more challenging setting of *model-free learning*, the complexity therein has a high dependence on the problem parameters, and the algorithm can only find non-perfect equilibria. When it comes to (perfect) equilibrium computation, one may exploit the multi-player zero-sum structure of zero-sum NMGs, and develop more natural and faster algorithms to find a Markov non-stationary NE. Moreover, when it comes to *stationary* equilibrium computation, even Markov CCE is *not* tractable in general-sum cases [30, 29]. Hereafter, we will focus on approaching zero-sum NMGs from these perspectives.

## 4 Hardness for Stationary CCE Computation

Given the results in Section 3.3, it seems tempting and sufficient to compute the Markov CCE of the zero-sum NMG. Indeed, computing CCE (and thus NE) in zero-sum polymatrix games is known to be tractable [23, 24]. It is thus natural to ask: *Is finding Markov CCE computationally tractable?* Next, we answer the question with different answers for finding stationary CCE (in infinite-horizon $\gamma$-discounted setting) and non-stationary CCE (in finite-horizon episodic setting), respectively.

For *two-player* infinite-horizon $\gamma$-discounted *zero-sum* MGs, significant progress in computing/learning the (Markov) stationary NE has been made recently [35, 36, 37, 38, 39, 40, 41, 42]. On the other hand, for *multi-player general-sum* MGs, recent results in [30, 29] showed that computing (Markov) stationary CCE can be PPAD-hard and thus believed to be computationally intractable. We next show that this hardness persists in most non-degenerate cases even if one enforces the zero-sum and networked interaction structures in the multi-player case. We state the formal result as follows, whose detailed proof is available in Section C.

**Theorem 1.** There is a constant $\epsilon > 0$ for which computing an $\epsilon$-approximate Markov perfect stationary CCE in infinite-horizon $\frac{1}{2}$-discounted zero-sum NMGs, whose underlying network structure contains either a triangle or a 3-path subgraph, is PPAD-hard. Moreover, given the PCP for PPAD conjecture [43], there is a constant $\epsilon > 0$ such that computing even an $\epsilon$-approximate Markov non-perfect stationary CCE in such zero-sum NMGs is PPAD-hard.

*Proof Sketch of Theorem 1.* Due to space constraints, we focus on the case with three players, and the underlying network structure has a triangle subgraph. Proof for the 3-path case is similar and can be found in Section C. We will show that for *any* general-sum two-player *turn-based* MG (**A**), the problem of computing its Markov stationary CCE, which is inherently a PPAD-hard problem [30], can be reduced to computing the Markov stationary CCE of a three-player zero-sum MG with a triangle structure networked separable interactions (**B**). Consider an MG (**A**) with two players, players 1 and 2, and reward functions $r_1(s, a_1, a_2)$ and $r_2(s, a_2, a_1)$, where $a_i$ is the action of the $i$-th player and $r_i$ is the reward function of the $i$-th player. The transition dynamics is given by $\mathbb{P}(s' \,|\, s, a_1, a_2)$. In even rounds, player 2's action space is limited to Noop2, and in odd rounds, player 1's action space is limited to Noop1, where Noop is an abbreviation of "no-operation", i.e., the player does not affect the transition dynamics or the reward in that round. We denote player 1's action space in even rounds as $\mathcal{A}_{1,\text{even}}$ and player 2's action space in odd rounds as $\mathcal{A}_{2,\text{odd}}$, respectively.

Now, we construct a three-player zero-sum NMG. with a triangle network structure. We set the reward function as $\widetilde{r}_i(s, \boldsymbol{a}) = \sum_{j \neq i} \widetilde{r}_{i,j}(s, a_i, a_j)$ and $\widetilde{r}_{i,j}(s, a_i, a_j) = -\widetilde{r}_{j,i}(s, a_j, a_i)$. The reward functions are designed so that $\widetilde{r}_{i,j} = -\widetilde{r}_{j,i}$ for all $i, j$, $\widetilde{r}_{1,2} + \widetilde{r}_{1,3} = r_1$, and $\widetilde{r}_{2,1} + \widetilde{r}_{2,3} = r_2$, where $r_1, r_2$ are the reward functions in game (**A**), by introducing a dummy player, player 3. In even rounds, player 2's action space is limited to Noop2, and in odd rounds, player 1's action space is limited to Noop1. Player 3's action space is always limited to Noop3 in all rounds. The transition dynamics is defined as $\widetilde{\mathbb{P}}(s' \,|\, s, a_1, a_2, a_3) = \mathbb{P}(s' \,|\, s, a_1, a_2)$, since $a_3$ is always chosen from Noop3. In other words, player 3's action does not affect the rewards of the other two players, nor the transition dynamics, and players 1 and 2 will receive the reward as in the two-player turn-based MG. Also, note that due to the turn-based structure of the game (**A**), the transition dynamics satisfy the decomposable condition in our Proposition 1, and it is thus a zero-sum NMG. In fact, turn-based dynamics can be represented as an ensemble of single controller dynamics, as we have discussed in Section 3.

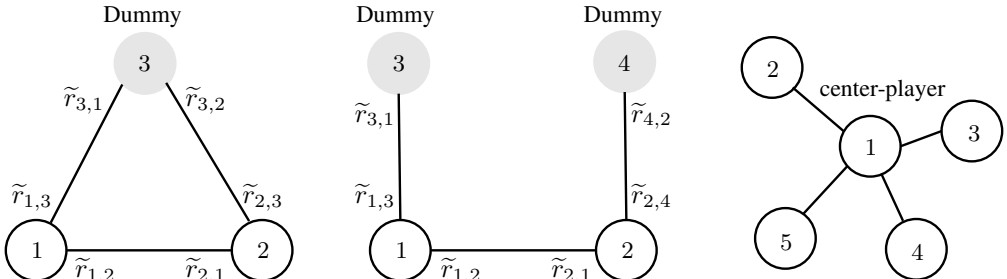

Figure 2: (Left, Middle): PPAD-hardness reduction visualization of $\mathcal{E}_Q$. (Right): A star-shaped zero-sum NMG.

Note that the new game **(B)** is still a turn-based game, and thus the Markov stationary CCE is the same as the Markov stationary NE. Also, note that by construction, the equilibrium policies of players 1 and 2 at the Markov stationary CCE of the game **(B)** constitute a Markov stationary CCE of the game **(A)**. If the underlying network is more general than a triangle, but contains a triangle subgraph, we can specify the reward and transition dynamics of these three players as above, and specify all other players to be dummy players, whose reward functions are all zero, and do not affect the reward functions of these three players, nor the transition dynamics. This completes the proof. □

Figure 2 briefly explains how we may reduce the equilibrium computation problem of **(A)** to that of **(B)**. In fact, a connected graph that does not contain a subgraph of a triangle or a 3-path has to be a *star-shaped* network (Proposition 7), which is proved in Section C. Hence, by Theorem 1, we know that in the infinite-horizon discounted setting, finding Markov stationary NE/CE/CCE is a computationally hard problem unless the underlying network is star-shaped. This may also imply that *learning* Markov stationary NE in zero-sum NMGs, e.g., using natural dynamics like fictitious play to reach the NE, can be challenging, unless in the star-shaped case. In turn, one may hope fictitious-play dynamics to converge for star-shaped zero-sum NMGs. We instantiate this idea next in Section 5. Furthermore, in light of Theorem 1, we will shift gear to computing Markov *non-stationary* NE by utilizing the structure of networked separable interactions, as to be detailed in Section 6.

## 5   Fictitious-Play Property

In this section, we study the fictitious-play property of multi-player zero-sum games with networked separable interactions, for both the matrix and Markov game settings. Following the convention in [8], we refer to the games in which fictitious-play dynamics converge to the NE as the games that have the *fictitious-play property*. We defer the matrix game case results to Section D, where we have also established convergence of the well-known variant of FP, smooth FP [7], in zero-sum NGs.

Echoing the computational intractability of computing CCE of zero-sum NMG unless the underlying network structure is star-shaped in the infinite-horizon discounted setting (c.f. Theorem 1), we now consider the FP property in such games. Note that by Proposition 1, $\mathcal{E}_Q$ is a star-shape if and only if the reward structure is a star shape and $\mathcal{N}_C = \{1\}$, where player 1 is the center of the star (Figure 2), or there are only two players in zero-sum NMG. There is already existing literature for the latter case [15, 16], so we focus on the former case, which is a single-controller case where player 1 controls the transition dynamics, i.e., $\mathbb{P}(s' \mid s, \boldsymbol{a}) = \mathbb{P}_1(s' \mid s, a_1)$ for some $\mathbb{P}_1$. We now introduce the fictitious-play dynamics for such zero-sum NMGs.

Each player $i$ first initializes her beliefs of other players' policies as uniform distributions, and also initializes her belief of the $Q$-value estimates with arbitrary values. Then, at iteration $k$, player $i$ takes the *best-response* action based on her belief of other players' policies $(\widehat{\pi}_{-i}^{(k)}(s^{(k)}))$, and their $Q$ beliefs $\widehat{Q}_i^{(k)}(s^{(k)}, \boldsymbol{a})$:

$$a_i^{(k)} \in \underset{a_i \in \mathcal{A}_i}{\operatorname{argmax}} \ \widehat{Q}_i^{(k)}(s^{(k)}, e_{a_i}, \widehat{\pi}_{-i}^{(k)}(s^{(k)})).$$

Then, player $i$ implements the action $a_i^{(k)}$, observes other players' actions $a_{-i}^{(k)}$, and updates her beliefs as follows: for each player $i \in \mathcal{N}$, she updates her belief of the opponents' policies as

$$\widehat{\pi}_{-i}^{(k+1)}(s) = \widehat{\pi}_{-i}^{(k)}(s) + \mathbf{1}(s = s^{(k)})\alpha^{N(s)}(e_{a_{-i}^{(k)}} - \widehat{\pi}_{-i}^{(k)}(s))$$

for all $s \in \mathcal{S}$, with stepsize $\alpha^{N(s)} \geq 0$ where $N(s)$ is the visitation count for the state $s$; then if $i = 1$, this player 1 updates the belief of $Q_{1,j}$ for all $j \in \mathcal{N}/\{1\}$ and her own $\widehat{Q}_1(s, \boldsymbol{a})$ for all $s \in \mathcal{S}$ as

$$\widehat{Q}_{1,j}^{(k+1)}(s,a_1,a_j) = \widehat{Q}_{1,j}^{(k)}(s,a_1,a_j)$$
$$+ \mathbf{1}(s = s^{(k)})\beta^{N(s)}\Big(r_{1,j}(s,a_1,a_j) + \gamma \sum_{s'\in\mathcal{S}} \frac{\mathbb{P}_1(s'\mid s,a_1)}{n-1}\cdot \widehat{V}_1^{(k)}(s') - \widehat{Q}_{1,j}^{(k)}(s,a_1,a_i)\Big),$$

which is based on the canonical decomposition given in Proposition 2, where $\widehat{V}_1^{(k)}(s) = \max_{a_1\in\mathcal{A}_1}\widehat{Q}_1^{(k)}(s,e_{a_1},\widehat{\pi}_{-1}^{(k)}(s))$, and $\beta^{N(s)} \geq 0$ is the stepsize. The agent then updates $\widehat{Q}_1^{(k+1)}(s,\boldsymbol{a}) = \sum_{j\in\mathcal{N}/\{1\}}\widehat{Q}_{1,j}^{(k+1)}(s,a_1,a_j)$, for all $s\in\mathcal{S}, \boldsymbol{a}\in\mathcal{A}$. Otherwise, if $i\neq 1$, then player $i$ updates the belief of her $\widehat{Q}_{i,1}(s,\boldsymbol{a})$ for all $s\in\mathcal{S}, \boldsymbol{a}\in\mathcal{A}$ as

$$\widehat{Q}_{i,1}^{(k+1)}(s,a_i,a_1) = \widehat{Q}_{i,1}^{(k)}(s,a_i,a_1)$$
$$+ \mathbf{1}(s = s^{(k)})\beta^{N(s)}\Big(r_{i,1}(s,a_i,a_1) + \gamma \sum_{s'\in\mathcal{S}} \mathbb{P}_1(s'\mid s,a_1)\cdot \widehat{V}_i^{(k)}(s') - \widehat{Q}_{i,1}^{(k)}(s,a_i,a_1)\Big),$$

where $\widehat{V}_i^{(k)}(s) = \max_{a_i\in\mathcal{A}_i}\widehat{Q}_i^{(k)}(s,e_{a_i},\widehat{\pi}_{-i}^{(k)}(s))$, and we let $\widehat{Q}_i^{(k+1)}(s,\boldsymbol{a}) = \widehat{Q}_{i,1}^{(k+1)}(s,a_i,a_1)$ for these $i\neq 1$. The overall dynamics are summarized in Algorithm 4, which resembles the FP dynamics for two-player zero-sum [16] and identical-interest [19] MGs. Now we are ready to present the convergence guarantees.

**Assumption 1.** The sequences of step sizes $\big\{\alpha^k \in (0,1]\big\}_{k\geq 0}$ and $\big\{\beta^k \in (0,1]\big\}_{k\geq 0}$ satisfy the following conditions: (1) $\sum_{k=0}^{\infty}\alpha^k = \infty$, $\sum_{k=0}^{\infty}\beta^k = \infty$, and $\lim_{k\to\infty}\alpha^k = \lim_{k\to\infty}\beta^k = 0$; (2) $\lim_{k\to\infty}\frac{\beta^k}{\alpha^k} = 0$, indicating that the rate at which the beliefs about $Q$-functions are updated is slower than the rate at which the beliefs about policies are updated.

**Theorem 2.** Suppose Assumption 1 holds and Algorithm 4 visits every state infinitely often with probability 1. Then, for a star-shaped multi-player zero-sum NMG, the belief $(\widehat{\pi}^{(k)})_{k\geq 0}$ converges to a Markov stationary NE and the belief $(\widehat{Q}^{(k)})_{k\geq 0}$ converges to the corresponding NE value of the zero-sum NMG with probability 1, as $k\to\infty$.

We defer the proof to Section D.2 due to space constraints. Note that to illustrate the idea, we only present the result for the *model-based* case, i.e., when the transition dynamics $\mathbb{P}$ is known. With this result, it is direct to extend to the model-free and learning case, where $\mathbb{P}$ is not known [16, 19, 17], still using the tool of stochastic approximation [44]. See Section D for more details.

**Remark 4** (Challenges for analyzing general cases). One might ask why we had to focus on a star-shaped structure. First, for general networked structures, even in the matrix-game case, it is known that the NE *values* of a zero-sum NG may not be unique [24]. Hence, suppose one performs *Nash-value iteration*, i.e., solving for the NE of the stage game and conducting backward induction, this value iteration process does not converge in general as the number of backward steps increases, since the solution at each stage is not even unique, and there may not exist a unique fixed point. This is in stark contrast to the max and max min operators in the value iteration updates for single-player and two-player zero-sum cases, respectively. By exploiting a star-shaped structure, we managed to reformulate a *minimax* optimization problem when solving each stage game, which makes the corresponding value iteration operator *contracting*, and thus iterating it infinitely converges to the unique fixed point. Second, suppose there exists some other network structure (other than star-shaped ones) that also leads to a contracting value iteration operator, then for a fixed constant $\gamma$, the fixed point (which corresponds to the Markov stationary CCE/NE of the zero-sum NMG) becomes unique and can be computed efficiently, which contradicts our hardness result in Theorem 1. Indeed, it was the exclusion of a star-shaped structure in Theorem 1 that inspired us to consider this structure in proving the convergence of FP dynamics. That being said, we note that having a contracting value iteration operator is only a *sufficient* condition for the FP dynamics to converge. It would be interesting to explore other structures that enjoy the FP property for reasons beyond this contraction property. We leave this as an immediate future work.

Next, we present another positive result in light of the hardness in Theorem 1, regarding the computation of *non-stationary* equilibria in multi-player zero-sum NMGs.

# 6 Non-Stationary NE Computation

We now focus on computing an (approximate) Markov *non-stationary* equilibrium in zero-sum NMGs. In particular, we show that when relaxing the stationarity requirement, not only CCE, but NE, can be computed efficiently. Before introducing our algorithm, we first recall the folklore result that approximating Markov non-stationary NE in *infinite-horizon* discounted settings can be achieved by finding approximate Markov NE in *finite-horizon* settings, with a large enough horizon length (c.f. Proposition 10). Hence, we will focus on the finite-horizon setting from now on.

Before delving into the details of our algorithm, we introduce the notation $\boldsymbol{Q}_{h,i}(s)$ and $\boldsymbol{Q}_h(s)$ for $h \in [H], i \in \mathcal{N}, s \in \mathcal{S}$ as follows:

$$\boldsymbol{Q}_{h,i}(s) := (Q_{h,i,1}(s), \ldots, Q_{h,i,i-1}(s), \boldsymbol{0}, Q_{h,i,i+1}(s) \ldots, Q_{h,i,n}(s)) \in \mathbb{R}^{|\mathcal{A}_i| \times \sum_{i \in \mathcal{N}} |\mathcal{A}_i|}$$

$$\boldsymbol{Q}_h(s) := ((\boldsymbol{Q}_{h,1}(s))^\intercal, (\boldsymbol{Q}_{h,2}(s))^\intercal, \ldots, (\boldsymbol{Q}_{h,n}(s))^\intercal)^\intercal \in \mathbb{R}^{\sum_{i \in \mathcal{N}} |\mathcal{A}_i| \times \sum_{i \in \mathcal{N}} |\mathcal{A}_i|}.$$

Here, $Q_{h,i,j}$ represents an estimate of the equilibrium value function with canonical decomposition (Proposition 2). Hereafter, we similarly define the notation of $\boldsymbol{Q}_{h,i}^\pi$ and $\boldsymbol{Q}_h^\pi$. Our algorithm is based on value iteration, and iterates three main steps from $h = H$ to 1 as follows: (1) $Q$-value computation: compute $Q_{h,i,j}$, which estimates the equilibrium $Q$-value with a canonical decomposition form; in particular, when $\mathcal{N}_C \neq \emptyset$, $Q_{h,i,j}$ is updated for all $s \in \mathcal{S}, (i,j) \in \mathcal{E}_Q, a_i \in \mathcal{A}_i$, and $a_j \in \mathcal{A}_j$:

$$Q_{h,i,j}(s, a_i, a_j) = r_{h,i,j}(s, a_i, a_j) + \sum_{s' \in \mathcal{S}} \left( \frac{1}{|\mathcal{E}_{Q,i}|} \mathbf{1}(i \in \mathcal{N}_C) \mathbb{F}_{h,i}(s' \mid s, a_i) + \mathbf{1}(j \in \mathcal{N}_C) \mathbb{F}_{h,j}(s' \mid s, a_j) \right) V_{h+1,i}(s'),$$

(2) Policy update: update $\pi_h(s)$ with an NE-ORACLE: finding (approximate)-NE of some zero-sum NG $(\mathcal{G}, \mathcal{A}, (Q_{h,i,j}(s))_{(i,j) \in \mathcal{E}_Q})$ for all $s \in \mathcal{S}$, and (3) Value function update: compute $V_{h,i}$, which estimates the equilibrium value function as follows for all $s \in \mathcal{S}, i \in \mathcal{N}$: $V_{h,i}(s) = \pi_{h,i}^\intercal(s) \boldsymbol{Q}_{h,i}(s) \pi_h(s)$. The overall procedure is summarized in Algorithm 6.

**NE-ORACLE and iteration complexity.** The NE-ORACLE in Algorithm 6 can be instantiated by several different algorithms that can find an NE in a zero-sum NG. Depending on the algorithms, the convergence guarantees can be either in terms of average-iterate, best-iterate, or last-iterate. Note that for algorithms with average-iterate convergence, one may additionally need to *marginalize* the output joint policy, i.e., the approximate CCE, and combine them as a *product* policy that is an approximate NE (Proposition 6). For those with best-/last-iterate convergence, by contrast, the best-/last-iterate is already in product form, and one can directly output it as an approximate NE. Moreover, last-iterate convergence is known to be a more favorable metric than the average-iterate one in learning in games [45, 46, 47, 48, 49], which is able to characterize the *day-to-day* behavior of the iterates and implies the stability of the update rule. Hence, one may prefer to have last-iterate convergence for solving zero-sum N(M)Gs. To this end, two algorithmic ideas may be useful: adding regularization to the payoff matrix [39, 42, 50, 51, 52], and/or using the idea of optimism [47, 36, 53]. Recent results [54, 51] have instantiated the ideas of *optimism-only* and *optimism + regularization*, respectively, for best-/last-iterate convergence in zero-sum polymatrix games. We additionally established results for the idea of *regularization-only* in obtaining last-iterate convergence in these games. Specifically, we propose to study the vanilla Multiplicative Weight Update (MWU) algorithm [55] in the regularized zero-sum NG, as tabulated in Algorithm 9. We have also introduced a variant with diminishing regularization, and summarize the update rule in Algorithm 10.

Given the results above, aggregating $\epsilon$-approximate NE for the zero-sum NGs $(\mathcal{G}, \mathcal{A}, (Q_{h,i,j}(s))_{(i,j) \in \mathcal{E}_Q})$ for all $h \in [H], i \in \mathcal{N}, s \in \mathcal{S}$ provides an $H\epsilon$-approximate NE for the corresponding zero-sum NMG. We have the following formal result.

**Proposition 4.** Suppose that for all $h \in [H], i \in \mathcal{N}, s \in \mathcal{S}$, NE-ORACLE$(\mathcal{G}, \mathcal{A}, (Q_{h,i,j}(s))_{(i,j) \in \mathcal{E}_Q})$ provides an $\epsilon_{h,s}$-approximate NE for the zero-sum NG $(\mathcal{G}, \mathcal{A}, (Q_{h,i,j}(s))_{(i,j) \in \mathcal{E}_Q})$ in Algorithm 6. Then, the output policy $\pi$ in Algorithm 6 is an $(\sum_{h \in [H]} \max_{s \in \mathcal{S}} \epsilon_{h,s})$-approximate NE for the corresponding zero-sum NMG $(\mathcal{G} = (\mathcal{N}, \mathcal{E}_Q), \mathcal{S}, \mathcal{A}, H, (\mathbb{P}_h)_{h \in [H]}, (r_{h,i,j}(s))_{(i,j) \in \mathcal{E}_Q, s \in \mathcal{S}})$.

The proof of Proposition 4 is deferred to Section F. In light of Proposition 4 and Table 1, we obtain Table 2, which summarizes the iteration complexities required to find an $\epsilon$-NE for zero-sum NMGs, with different NE-ORACLE subroutines. Note that the iteration complexities are all polynomial in $H, n, |\mathcal{S}|$, and inherit the order of dependencies on $\epsilon$ from Table 1 for the matrix-game case. In particular, Algorithm 6 with the OMWU in [51] yields the fast rate of $\widetilde{\mathcal{O}}(1/\epsilon)$ for the last iterate.

## Acknowledgement

The authors would like to thank the anonymous reviewers of NeurIPS for their helpful feedback. C.P. acknowledges support from the Xianhong Wu Fellowship, the Korea Foundation for Advanced Studies, and the Siebel Scholarship. K.Z. acknowledges support from the Northrop Grumman – Maryland Seed Grant Program. A.O. acknowledges support from the MIT-Air Force AI Innovation Accelerator Grant and the MIT-DSTA grant 031017-00016.

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
