# Supplementary Materials for
# "Multi-Player Zero-Sum Markov Games with
# Networked Separable Interactions"

In Appendix A, we provide a detailed literature review. In Appendix B, we provide deferred proofs for the results of the zero-sum NMG formulation in Section 3. In Section C, we provide deferred proofs for the PPAD-hardness of computing Markov stationary CCE in zero-sum NMGs, in Section 4. In Section D, we provide deferred proofs for the fictitious-play property results, in Section 5. In Section E, we provide a brief background on stochastic approximation. In Section F, we provide deferred proofs for the results regarding Markov non-stationary NE computation in Section 6. Finally, in Section G, we provide numerical experiments to validate our algorithms.

## A    Related Work

**Tabular Markov game.**    Markov games (MG), which are also referred to as stochastic games, were initially introduced by [13] and have since garnered significant attention within the multi-agent RL literature [56, 57]. Early research, such as [58, 59, 60, 61], established asymptotic convergence of various Q-learning-based dynamics in solving MGs. In contrast, recent studies have mainly focused on developing more sample-efficient methods for learning equilibria in two-player zero-sum Markov games, as demonstrated by [62, 63, 64, 65, 66, 37, 67, 68].

Substantial work has also been conducted on learning correlated equilibrium and coarse correlated equilibrium in Markov games, including model-based [66, 69] and model-free approaches [70, 71, 72, 73]. A recently developed algorithm by [30] is able to learn Markov non-stationary CCE while overcoming the curse of multi-agents, whose sample complexity has recently been improved in [74, 75]. Other studies within the full-information feedback setting have focused on proving convergence to CE/CCE and sublinear individual regret [76].

**Complexity of equilibrium computation.**    Computational challenges can occur for Nash equilibrium-finding in even matrix/normal-form games in general. Computing such equilibria has been proven to be PPAD-complete even for three/two-player general-sum normal-form games [20, 21], which is believed to be computationally hard [32, 77]. Nevertheless, linear programming enables the computation of Nash equilibria in *two-player zero-sum* games and *zero-sum polymatrix* games [24]. Alternative solution concepts including (coarse) correlated equilibria are also more favorable than NE when it comes to computational complexity, as they can also be efficiently computed [22, 3]. More recently, [30, 29] have shown that for infinite-horizon discounted Markov games, computing even the coarse correlated equilibrium that is Markov stationary can be PPAD-hard, which is in stark contrast to the stateless normal-form game case. For a recent overview of the computational complexity for equilibrium computation, we refer to [78].

**Games with network structure.**    Network Games [79] and Graphical Games [80] have been extensively studied in the literature to model the networked interactions among agents. [80] introduced treeNash, an algorithm for computing NE in tree-structured graphical games. The algorithm by [81] can find correlated equilibrium in graphical games. Polymatrix games, wherein edges represent two-player games, constitute a particularly intriguing type of network games. [28] introduced the concept of *separable* zero-sum games, where a player's payoff is the sum of their payoffs from pairwise interactions with other players, and provided equilibrium-finding algorithms. [82] demonstrated that graphical games with edges representing zero-sum games (also called *pairwise zero-sum polymatrix* games) can be reduced to two-person zero-sum games, streamlining the NE computation for this case. [23] established that separable zero-sum multiplayer games can be transformed into pairwise constant-sum polymatrix games. [24] revealed properties of NE in separable zero-sum games, such as non-unique NE payoffs and the reduction of NE computation to CCE computation by marginalizing the equilibria.

More recently, researchers have proposed several NE-finding methods that do not depend on linear programming (LP). [83] employed a continuous-time version of Q-learning to approximate NE in

weighted zero-sum polymatrix games, [54] utilized optimistic mirror descent to find NE in constant-sum polymatrix games, and [51] applied optimistic multiplicative weight updates to find NE in zero-sum polymatrix games.

In the setting with state transitions, the networked structure has also been exploited recently in multi-agent RL [84, 85, 86, 87, 88, 89], where either the communication or interaction, in terms of reward or transition, were assumed to have some networked structure. However, most of these results were focused on the *cooperative* setting (or more generally the *potential* game setting). We instead focus on a multi-player while *non-cooperative*, specifically, *zero-sum*, setting.

In the extensive form games literature, [90] proved that optimistic gradient ascent provides $\mathcal{O}(1/T)$ convergence rate to NE in the network zero-sum extensive form games.

**Entropy regularization.** Entropy regularization is a common approach used in reinforcement learning to foster exploration and enable faster convergence. Recently, both empirical evidence and provable convergence rate guarantees for entropy-regularized MDPs have been established [91, 92, 93, 94, 95, 96]. In addition to its applications in single-agent RL, entropy regularization has been investigated in game-theoretic settings, including two-player zero-sum matrix games [42], multi-player zero-sum games [83, 51], potential games [50], and extensive-form games [97, 98].

**Fictitious play.** Fictitious play is a classical learning dynamics in game theory introduced by [5], in which players develop a belief in their opponent's policy and use a greedy approach to the belief they hold about the opponent's policy. (Stochastic) fictitious-play property ((S-)FPP) is a property of a game that ensures the convergence of (stochastic) fictitious play to a Nash equilibrium of the game. In the case of static games, (S-)FPP holds for two-player zero-sum games [6], 2x$n$ games [9, 10], $n$-player potential games [8], zero-sum polymatrix games [12]. However, FPP normally does not hold for 3x3 games [99]. For stochastic games [13], (S-)FPP holds for zero-sum and identical payoff games [16, 14, 17, 15, 100]. Recently, [100] proved that any stochastic game with turn-based controllers on state transitions has S-FPP, as long as the stage payoffs have S-FPP. For a more detailed overview of fictitious play in stochastic/Markov games, we refer to [101].

**Comparison with independent work [102].** While preparing our work, we noticed an independent preprint [102], which also studied the polymatrix zero-sum structure in Markov games. Encouragingly, they also showed the collapse of Markov CCE to Markov NE and thus their computational tractability. However, there are several key differences that may be summarized as follows. First, the model in [102] is *defined* as a combination of zero-sum polymatrix *reward functions* and *switching-controller* dynamics, under which the desired property of equilibria collapse holds; in contrast, we define the model based on the payoffs of the *auxiliary games* at each state, which, by our Proposition 1, is *equivalent* to the reward being zero-sum polymatrix and the dynamics being *ensemble* (c.f. Remark 3). Our ensemble dynamics covers the switching controller case, and our model is more general in this sense. Second, our proof for equilibria collapse is different from that in [102], which is based on characterizing the solution to some nonlinear program. We instead directly exploit the property of ensemble transition dynamics in marginalizing the joint policies, and its effect on dynamic programming in finding the equilibria. Third, in terms of equilibrium computation, we investigate a series of value-iteration-based algorithms, based on both existing and our new algorithms for solving zero-sum polymatrix games, with finite-iteration last-iterate convergence guarantees, including an $\widetilde{\mathcal{O}}(1/\epsilon)$ rate result. In comparison, [102] uses existing algorithms for *learning* Markov CCE due to equilibria collapse, i.e., [30]. Finally, we have additionally provided hardness results for *stationary* equilibria computation in infinite-horizon discounted settings, fictitious-play dynamics with convergence guarantees, as well as several examples of our model.

# B  Omitted Details in Section 3

## B.1  Omitted proof for Proposition 1 and Proposition 2

**Proposition 1.** For a given graph $\mathcal{G} = (\mathcal{N}, \mathcal{E}_Q)$, an MG $(\mathcal{N}, \mathcal{S}, \mathcal{A}, \mathbb{P}, (r_i)_{i \in \mathcal{N}}, \gamma)$ with more than two players is an NMG with respect to $\mathcal{G}$ if and only if: (1) $r_i(s, a_i, \cdot)$ **is decomposable with respect to** $\mathcal{E}_{Q,i}$ for each $i \in \mathcal{N}, s \in \mathcal{S}, a_i \in \mathcal{A}_i$, i.e., $r_i(s, \boldsymbol{a}) = \sum_{j \in \mathcal{E}_{Q,i}} r_{i,j}(s, a_i, a_j)$ for a set of functions $\{r_{i,j}(s, a_i, \cdot)\}_{j \in \mathcal{E}_{Q,i}}$, and (2) **the transition dynamics** $\mathbb{P}(s' \mid s, \cdot)$ **is decomposable with respect to**

the set $\mathcal{N}_C$ of this $\mathcal{G}$, i.e., $\mathbb{P}(s' \mid s, \boldsymbol{a}) = \sum_{i \in \mathcal{N}_C} \mathbb{F}_i(s' \mid s, a_i)$ for a set of functions $\{\mathbb{F}_i(s' \mid s, \cdot)\}_{i \in \mathcal{N}_C}$ if $\mathcal{N}_C \neq \emptyset$, or $\mathbb{P}(s' \mid s, \boldsymbol{a}) = \mathbb{F}_o(s' \mid s)$ for some constant function (of $\boldsymbol{a}$) $\mathbb{F}_o(s' \mid s)$ if $\mathcal{N}_C = \emptyset$. Moreover, an MG qualifies as a zero-sum NMG if and only if it satisfies an additional condition: the NG, characterized by $(\mathcal{G}, \mathcal{A}, (r_{i,j}(s))_{(i,j) \in \mathcal{E}_Q})$, is a zero-sum NG for all $s \in \mathcal{S}$. In the case of two players, every (zero-sum) Markov game becomes a (zero-sum) NMG.

*Proof.* Firstly, in the case with $\mathcal{N}_C \neq \emptyset$, we prove that if an MG satisfies the decomposability of the reward function $r_i(s, a_i, \cdot)$ and the transition dynamics $\mathbb{P}(s' \mid s, \cdot)$, then $Q_{i,j}^V$ can be decomposed as follows:

$$Q_i^V(s, \boldsymbol{a}) = r_i(s, \boldsymbol{a}) + \gamma \mathbb{E}_{s' \sim \mathbb{P}(\cdot \mid s, \boldsymbol{a})} V(s') = \sum_{j \in \mathcal{E}_{Q,i}} r_{i,j}(s, a_i, a_j) + \gamma \sum_{j \in \mathcal{N}_C} \sum_{s' \in \mathcal{S}} \mathbb{F}_j(s' \mid s, a_j) V(s')$$

$$= \sum_{j \in \mathcal{E}_{Q,i}} \left( r_{i,j}(s, a_i, a_j) + \gamma \sum_{s' \in \mathcal{S}} (\lambda_{i,j}(s) \mathbf{1}(i \in \mathcal{N}_C) \mathbb{F}_i(s' \mid s, a_i) + \mathbf{1}(j \in \mathcal{N}_C) \mathbb{F}_j(s' \mid s, a_j)) V(s') \right)$$

$$=: \sum_{j \in \mathcal{E}_{Q,i}} Q_{i,j}^V(s, a_i, a_j),$$

for any non-negative $(\lambda_{i,j}(s))_{(i,j) \in \mathcal{E}_Q}$ such that $\sum_{j \in \mathcal{E}_{Q,i}} \lambda_{i,j}(s) = 1$, since by definition $\mathcal{N}_C \subseteq \mathcal{E}_{Q,i}$ for every $i \in \mathcal{N}$.

In the case when $\mathcal{N}_C = \emptyset$, we prove that if the MG satisfies decomposability of $r_i(s, a_i, \cdot)$ and $\mathbb{P}(\cdot \mid s, \boldsymbol{a}) = \mathbb{F}_o(\cdot \mid s)$, then $Q_{i,j}^V$ can be decomposed as follows:

$$Q_i^V(s, \boldsymbol{a}) = r_i(s, \boldsymbol{a}) + \gamma \mathbb{E}_{s' \sim \mathbb{P}(\cdot \mid s, \boldsymbol{a})} V(s') = \sum_{j \in \mathcal{E}_{Q,i}} \left( r_{i,j}(s, a_i, a_j) + \gamma \sum_{s' \in \mathcal{S}} \lambda_{i,j}(s) \mathbb{F}_o(s' \mid s) V(s') \right)$$

$$=: \sum_{j \in \mathcal{E}_{Q,i}} Q_{i,j}^V(s, a_i, a_j),$$

for any non-negative $(\lambda_{i,j}(s))_{(i,j) \in \mathcal{E}_Q}$ such that $\sum_{j \in \mathcal{E}_{Q,i}} \lambda_{i,j}(s) = 1$.

Next, we prove the necessary conditions for an MG to be an NMG. By definition, we have

$$Q_i^V(s, \boldsymbol{a}) = \sum_{j \in \mathcal{E}_{Q,i}} Q_{i,j}^V(s, a_i, a_j) = r_i(s, \boldsymbol{a}) + \gamma \langle \mathbb{P}(\cdot \mid s, \boldsymbol{a}), V(\cdot) \rangle$$

for any $V$, which indicates that

$$\sum_{j \in \mathcal{E}_{Q,i}} (Q_{i,j}^V(s, a_i, a_j) - Q_{i,j}^{V'}(s, a_i, a_j)) = \gamma \langle \mathbb{P}(\cdot \mid s, \boldsymbol{a}), V(\cdot) - V'(\cdot) \rangle, \qquad (2)$$

for any $V, V'$ and any $(s, \boldsymbol{a})$.

For every $s \in \mathcal{S}$, define $B_s : \mathcal{S} \to \mathbb{R}$ such that $B_s(s') = \mathbf{1}(s = s')$. We define $\mathbb{G}_{i,j}(s' \mid s, a_i, a_j) := 1/\gamma \sum_{j \in \mathcal{E}_{Q,i}} (Q_{i,j}^{B_{s'}}(s, a_i, a_j) - Q_{i,j}^{\mathbf{0}}(s, a_i, a_j))$, then by plugging in $V = B_{s'}$ and $V' = \mathbf{0}$, we can derive $\mathbb{P}(\cdot \mid s, \boldsymbol{a}) = \sum_{j \in \mathcal{E}_{Q,i}} \mathbb{G}_{i,j}(\cdot \mid s, a_i, a_j)$ for every $i$ from Equation (2). Alternatively, we can take a functional derivative of Equation (2) with respect to $V - V'$, which means that there exist some functions $\{\mathbb{G}_{i,j}\}_{j \in \mathcal{E}_{Q,i}}$ such that $\mathbb{P}(\cdot \mid s, \boldsymbol{a}) = \sum_{j \in \mathcal{E}_{Q,i}} \mathbb{G}_{i,j}(\cdot \mid s, a_i, a_j)$ for every $i$. Note that the decomposability of $\mathbb{P}$ with respect to $\mathcal{E}_{Q,i}$ above has to hold for all $i \in \mathcal{N}$. Therefore, when $\mathcal{N}_C \neq \emptyset$, if $j \notin \mathcal{N}_C$, then there exists some $i \in \mathcal{N}$ such that $(i, j) \notin \mathcal{E}_Q$. In this case, $\mathbb{P}$ is not dependent on this $j$. So $\mathbb{P}$ should be a function of the players in $\mathcal{N}_C$, which indicates that $\mathbb{P}(\cdot \mid s, \boldsymbol{a}) = \sum_{j \in \mathcal{N}_C, j \neq i} \mathbb{F}_{i,j}(\cdot \mid s, a_i, a_j)$ for every $i$ unless $\mathcal{N}_C = \emptyset$. If $\mathcal{N}_C = \emptyset$, then it directly concludes that $\mathbb{P}(\cdot \mid s, \boldsymbol{a}) = \mathbb{F}_o(\cdot \mid s)$ for some $\mathbb{F}_o$, since by the argument above, it should not depend on any player $j$. Next, we focus on the case when $\mathcal{N}_C \neq \emptyset$, and there are more than two players.

Specifically, in this case, if there exist some $k_1 \neq k_2$ and $k_1 \notin \mathcal{N}_C$, such that

$$\mathbb{P}(\cdot \mid s, \boldsymbol{a}) = \sum_{i \in \mathcal{N}_C, i \neq k_1} \mathbb{F}_{k_1, i}(\cdot \mid s, a_{k_1}, a_i) = \sum_{i \in \mathcal{N}_C, i \neq k_2} \mathbb{F}_{k_2, i}(\cdot \mid s, a_{k_2}, a_i),$$

then we choose a fixed $a_{k_1}$ but changing $a_{k_2}$ arbitrarily. This preserves the equality, indicating that

$$\sum_{i\in\mathcal{N}_C, i\neq k_1} \mathbb{F}_{k_1,i}(\cdot|s, a_{k_1,\text{fix}}, a_i) = \sum_{i\in\mathcal{N}_C, i\neq k_2} \mathbb{F}_{k_2,i}(\cdot|s, a_{k_2}, a_i) \tag{3}$$

for any $a_{k_2}$. Since $k_1 \notin \mathcal{N}_C$, we can have the left-hand side of (3) written as $\sum_{i\in\mathcal{N}_C} \mathbb{F}_i(\cdot|s, a_i)$ by setting $\mathbb{F}_i(\cdot|s, a_i) := \mathbb{F}_{k_1,i}(\cdot|s, a_{k_1,\text{fix}}, a_i)$. Meanwhile, the right-hand side of (3) is also $\mathbb{P}(\cdot|s, \boldsymbol{a})$ by definition, which concludes that $\mathbb{P}(\cdot|s, \boldsymbol{a}) = \sum_{i\in\mathcal{N}_C} \mathbb{F}_i(\cdot|s, a_i)$, and proves the theorem. In other words, as long as at least one $k_1$ does not belong to $\mathcal{N}_C$, we can conclude the theorem.

If no such a $k_1 \notin \mathcal{N}_C$ exists, then it means that all players are in $\mathcal{N}_C$. In this case, for any $k_1 \neq k_2$, for a fixed $a_{k_1,\text{fix}}$, we have

$$\sum_{i\in\mathcal{N}_C/\{k_1\}} \mathbb{F}_{k_1,i}(\cdot|s, a_{k_1,\text{fix}}, a_i) = \sum_{i\in\mathcal{N}_C/\{k_1,k_2\}} \mathbb{F}_{k_2,i}(\cdot|s, a_{k_2}, a_i) + \mathbb{F}_{k_2,k_1}(\cdot|s, a_{k_2}, a_{k_1,\text{fix}}). \tag{4}$$

Therefore, if we define $\mathbb{G}_i(\cdot|s, a_i) := \mathbb{F}_{k_1,i}(\cdot|s, a_{k_1,\text{fix}}, a_i)$ for $i \in \mathcal{N}_C/\{k_1, k_2\}$, and $\mathbb{G}_{k_2}(\cdot|s, a_{k_2}) := \mathbb{F}_{k_1,k_2}(\cdot|s, a_{k_1,\text{fix}}, a_{k_2}) - \mathbb{F}_{k_2,k_1}(\cdot|s, a_{k_2}, a_{k_1,\text{fix}})$, then we have

$$\sum_{i\in\mathcal{N}_C/\{k_1,k_2\}} \mathbb{F}_{k_2,i}(\cdot|s, a_{k_2}, a_i) = \sum_{i\in\mathcal{N}_C/\{k_1\}} \mathbb{G}_i(\cdot|s, a_i). \tag{5}$$

Let $k_3 \in \mathcal{N}_C$ such that $k_3 \neq k_1, k_2$. By definition of $\mathbb{F}_{i,j}$, we have

$$\mathbb{P}(\cdot|s, \boldsymbol{a}) = \sum_{i\in\mathcal{N}_C/\{k_3\}} \mathbb{F}_{k_3,i}(\cdot|s, a_{k_3}, a_i) = \sum_{i\in\mathcal{N}_C/\{k_2\}} \mathbb{F}_{k_2,i}(\cdot|s, a_{k_2}, a_i).$$

Plugging (5), we have

$$\mathbb{P}(\cdot|s, \boldsymbol{a}) = \sum_{i\in\mathcal{N}_C/\{k_3\}} \mathbb{F}_{k_3,i}(\cdot|s, a_{k_3}, a_i) = \sum_{i\in\mathcal{N}_C/\{k_1\}} \mathbb{G}_i(\cdot|s, a_i) + \mathbb{F}_{k_2,k_1}(\cdot|s, a_{k_2}, a_{k_1}) \tag{6}$$

for any $\boldsymbol{a} \in \mathcal{A}$. If we now fix $a_{k_3}$ as $a_{k_3,\text{fix}}$, then from (6) we know that

$$\sum_{i\in\mathcal{N}_C/\{k_3\}} \mathbb{F}_{k_3,i}(\cdot|s, a_{k_3,\text{fix}}, a_i) = \sum_{i\in\mathcal{N}_C/\{k_1,k_3\}} \mathbb{G}_i(\cdot|s, a_i) + \mathbb{G}_{k_3}(\cdot|s, a_{k_3,\text{fix}}) + \mathbb{F}_{k_2,k_1}(\cdot|s, a_{k_2}, a_{k_1}).$$

$$\tag{7}$$

Plugging (7) to (6), we have

$$\mathbb{P}(\cdot|s, \boldsymbol{a}) = \sum_{i\in\mathcal{N}_C/\{k_1,k_3\}} \mathbb{G}_i(\cdot|s, a_i) + \mathbb{F}_{k_2,k_1}(\cdot|s, a_{k_2}, a_{k_1}) + \mathbb{G}_{k_3}(\cdot|s, a_{k_3})$$

$$= \sum_{i\in\mathcal{N}_C/\{k_3\}} \mathbb{F}_{k_3,i}(\cdot|s, a_{k_3,\text{fix}}, a_i) - \mathbb{G}_{k_3}(\cdot|s, a_{k_3,\text{fix}}) + \mathbb{G}_{k_3}(\cdot|s, a_{k_3}) =: \sum_{i\in\mathcal{N}_C} \mathbb{F}_i(\cdot|s, a_i)$$

where $\mathbb{F}_i(\cdot|s, a_i) := \mathbb{F}_{k_3,i}(\cdot|s, a_{k_3,\text{fix}}, a_i)$ for $i \in \mathcal{N}_C/\{k_3\}$ and $\mathbb{F}_{k_3}(\cdot|s, a_{k_3}) := -\mathbb{G}_{k_3}(\cdot|s, a_{k_3,\text{fix}}) + \mathbb{G}_{k_3}(\cdot|s, a_{k_3})$, which concludes the decomposability of the transition dynamics. Finally, note that we can ensure the non-negativity of $\mathbb{F}_i$, since we can iterate the following procedure:

---

**Algorithm 1** Procedure for constructing non-negative $\{\mathbb{F}_i\}_{i\in\mathcal{N}_C}$

---

  **while** there exists $s, s' \in \mathcal{S}$ and $i \in \mathcal{N}_C$ such that $\min_{a_i\in\mathcal{A}_i} \mathbb{F}_i(s' \mid s, a_i) < 0$ **do**
    Set $s, s' \in \mathcal{S}$ and $i \in \mathcal{N}_C$ that satisfying $\min_{a_i\in\mathcal{A}_i} \mathbb{F}_i(s' \mid s, a_i) < 0$
    Sort $\mathcal{N}_C$ according to the descending order of $b_k := \min_{a_k\in\mathcal{A}_k} \mathbb{F}_k(s' \mid s, a_k)$, and denote it as $\{j_1, j_2, \ldots, j_{|\mathcal{N}_C|}\}$
    Define tmp1 $= 0, t = 1$
    **while** tmp1 $< -\min_{a_i\in\mathcal{A}_i} \mathbb{F}_i(s' \mid s, a_i)$ **do**
      Define tmp2 $= -\min_{a_i\in\mathcal{A}_i} \mathbb{F}_i(s' \mid s, a_i) -$ tmp1 $> 0$
      Define tmp3 $= \min(\min_{a_{j_t}\in\mathcal{A}_{j_t}} \mathbb{F}_{j_t}(s' \mid s, a_{j_t}), \text{tmp2}) > 0$;
      Update $\mathbb{F}_{j_t}(s'|s, a_{j_t}) \leftarrow \mathbb{F}_{j_t}(s' \mid s, a_{j_t}) -$ tmp3 for all $a_{j_t} \in \mathcal{A}_{j_t}$
      Update $\mathbb{F}_i(s' \mid s, a_i) \leftarrow \mathbb{F}_i(s' \mid s, a_i) +$ tmp3 for all $a_i \in \mathcal{A}_i$
      Update tmp1 $\leftarrow$ tmp1 + tmp3 and $t \leftarrow t + 1$
    **end while**
  **end while**

---

In this context, the third and fourth lines of the inner-while loop ensure that $\mathbb{P}(s'|s,\boldsymbol{a}) = \sum_{j \in \mathcal{N}_C} \mathbb{F}_j(s' \mid s, a_j)$ always holds for $s, s' \in \mathcal{S}$, $\boldsymbol{a} \in \mathcal{A}$. Furthermore, tmp3 remains greater than 0 within the inner-while loop. This is because if $\min_{a_{j_t} \in \mathcal{A}_{j_t}} \mathbb{F}_{j_t}(s' \mid s, a_{j_t}) \leq 0$, then it implies that $\min_{a_{j_k} \in \mathcal{A}_{j_k}} \mathbb{F}_{j_k}(s' \mid s, a_{j_k}) = 0$ for all $k \in [t]$. Also, $\min_{a_{j_k} \in \mathcal{A}_{j_k}} \mathbb{F}_{j_k}(s' \mid s, a_{j_k}) \leq \min_{a_{j_t} \in \mathcal{A}_{j_t}} \mathbb{F}_{j_t}(s' \mid s, a_{j_t}) \leq 0$ for every $k \in [|\mathcal{N}_C|]$. Meanwhile, since we assumed $\min_{a_i \in \mathcal{A}_i} \mathbb{F}_i(s' \mid s, a_i) < 0$, we can define a vector $\widetilde{a}$ with elements $\widetilde{a}_j \in \arg\min_{a_j \in \mathcal{A}_j} \mathbb{F}_j(s' \mid s, a_j)$. This implies that $\mathbb{P}(s' \mid s, \widetilde{a}) = \sum_{i \in \mathcal{N}_C} \min_{a_i \in \mathcal{A}_i} \mathbb{F}_i(s' \mid s, a_i) < 0$, which contradicts the condition $\mathbb{P}(s' \mid s, \widetilde{a}) \geq 0$.

Therefore, our procedure ensures that the number of pairs $(s, s') \in \mathcal{S} \times \mathcal{S}$ and indexes $i \in \mathcal{N}_C$ for which $\min_{a_i \in \mathcal{A}_i} \mathbb{F}_i(s' \mid s, a_i) < 0$ is consistently reduced.

If there are only two players, then both of them belong to $\mathcal{N}_C$. However, one cannot decompose the transition dynamics as above, as there is no such a $k_3 \neq k_1, k_2$ to construct the aforementioned formula. Indeed, any two-player (zero-sum) MG satisfies our definition of zero-sum NMGs in Definition 1.

For the reward decomposition, if $\mathcal{N}_C \neq \emptyset$, we have

$$r_i(s, \boldsymbol{a}) = \sum_{j \in \mathcal{E}_{Q,i}} \left( Q_{i,j}^V(s, a_i, a_j) - \gamma \langle \mathbf{1}(j \in \mathcal{N}_C) \mathbb{F}_j(\cdot \mid s, a_j), V(\cdot) \rangle \right)$$

and if $\mathcal{N}_C = \emptyset$, we have

$$r_i(s, \boldsymbol{a}) = \sum_{j \in \mathcal{E}_{Q,i}} \left( Q_{i,j}^V(s, a_i, a_j) - \gamma \langle \frac{1}{|\mathcal{E}_{Q,i}|} \mathbb{F}_o(\cdot \mid s), V(\cdot) \rangle \right).$$

Hence, $r_i(s, a_i, \cdot)$ can be represented as $r_i(s, \boldsymbol{a}) = \sum_{j \in \mathcal{E}_{Q,i}} r_{i,j}(s, a_i, a_j)$ for some functions $(r_{i,j})_{(i,j) \in \mathcal{E}_Q}$. The same procedure as Algorithm 1 provides that we can ensure the non-negativity of $r_{i,j}$, so that $r_i(s, a_i, \cdot)$ is decomposable with respect to $\mathcal{E}_{Q,i}$. In fact, adding any large-enough constant to $r_{i,j}$ does not change the solution to the problem, while ensuring the non-negativity of $r_{i,j}$. $\qquad\square$

**Proposition 1 - Finite-horizon version.** For a given graph $\mathcal{G} = (\mathcal{N}, \mathcal{E}_Q)$, an MG $(\mathcal{N}, \mathcal{S}, \mathcal{A}, (\mathbb{P}_h)_{h \in [H]}, (r_{h,i})_{i \in \mathcal{N}, h \in [H]})$ with more than two players is an NMG with $\mathcal{G}$ if and only if: (1) $r_{h,i}(s, a_i, \cdot)$ **is decomposable with respect to** $\mathcal{E}_{Q,i}$ for each $i \in \mathcal{N}$, $s \in \mathcal{S}$, $a_i \in \mathcal{A}_i$, $h \in [H]$, i.e., $r_{h,i}(s, \boldsymbol{a}) = \sum_{j \in \mathcal{E}_{Q,i}} r_{h,i,j}(s, a_i, a_j)$ for a set of functions $\{r_{h,i,j}(s, a_i, \cdot)\}_{j \in \mathcal{E}_{Q,i}}$ and (2) **the transition dynamics** $\mathbb{P}_h(s' \mid s, \cdot)$ **is decomposable with respect to** $\mathcal{N}_C$ corresponding to this $\mathcal{G}$ for all $h \in [H]$, i.e., $\mathbb{P}_h(s' \mid s, \boldsymbol{a}) = \sum_{i \in \mathcal{N}_C} \mathbb{F}_{h,i}(s' \mid s, a_i)$ for a set of functions $\{\mathbb{F}_{h,i}(s' \mid s, \cdot)\}_{i \in \mathcal{N}_C}$ if $\mathcal{N}_C \neq \emptyset$, or $\mathbb{P}_h(s' \mid s, \boldsymbol{a}) = \mathbb{F}_{h,o}(s' \mid s)$ for some constant function (of $\boldsymbol{a}$) $\mathbb{F}_{h,o}(s' \mid s)$ if $\mathcal{N}_C = \emptyset$. Moreover, an MG qualifies as a zero-sum NMG if and only if it satisfies an additional condition: the NG, characterized by $(\mathcal{G}, \mathcal{A}, (r_{h,i,j}(s))_{(i,j) \in \mathcal{E}_Q})$, must be a zero-sum NG for all $s \in \mathcal{S}, h \in [H]$. In the case of two players, every (zero-sum) Markov game becomes a (zero-sum) NMG.

**Proposition 2 (Decomposition of $(Q_i^V)_{i \in \mathcal{N}}$).** For an infinite-horizon $\gamma$-discounted NMG with $\mathcal{G} = (\mathcal{N}, \mathcal{E}_Q)$ such that $\mathcal{N}_C \neq \emptyset$, if we know that $\mathbb{P}(s' \mid s, \boldsymbol{a}) = \sum_{i \in \mathcal{N}_C} \mathbb{F}_i(s' \mid s, a_i)$, and $r_i(s, \boldsymbol{a}) = \sum_{j \in \mathcal{E}_{Q,i}} r_{i,j}(s, a_i, a_j)$ for some $\{\mathbb{F}_i\}_{i \in \mathcal{N}_C}$ and $\{r_{i,j}\}_{(i,j) \in \mathcal{E}_Q}$, then the $Q_{i,j}^V$ given in Definition 1 can be represented as

$$Q_{i,j}^V(s, a_i, a_j) = r_{i,j}(s, a_i, a_j) + \sum_{s' \in \mathcal{S}} \gamma \left( \mathbf{1}(j \in \mathcal{N}_C) \mathbb{F}_j(s' \mid s, a_j) + \mathbf{1}(i \in \mathcal{N}_C) \lambda_{i,j}(s) \mathbb{F}_i(s' \mid s, a_i) \right) V(s')$$

for any non-negative $(\lambda_{i,j}(s))_{(i,j) \in \mathcal{E}_Q}$ such that $\sum_{j \in \mathcal{E}_{Q,i}} \lambda_{i,j}(s) = 1$ for all $i \in \mathcal{N}$ and $s \in \mathcal{S}$. For an infinite-horizon $\gamma$-discounted NMG with $\mathcal{G} = (\mathcal{N}, \mathcal{E}_Q)$ such that $\mathcal{N}_C = \emptyset$, if we know that $\mathbb{P}(s' \mid s, \boldsymbol{a}) = \mathbb{F}_o(s' \mid s)$, and $r_i(s, \boldsymbol{a}) = \sum_{j \in \mathcal{E}_{Q,i}} r_{i,j}(s, a_i, a_j)$ for some $\mathbb{F}_o$ and $\{r_{i,j}\}_{(i,j) \in \mathcal{E}_Q}$, then the $Q_{i,j}^V$ given in Definition 1 can be represented as

$$Q_{i,j}^V(s, a_i, a_j) = r_{i,j}(s, a_i, a_j) + \sum_{s' \in \mathcal{S}} \gamma \left( \lambda_{i,j}(s) \mathbb{F}_o(s' \mid s) \right) V(s')$$

for any non-negative $(\lambda_{i,j}(s))_{(i,j)\in\mathcal{E}_Q}$ such that $\sum_{j\in\mathcal{E}_{Q,i}}\lambda_{i,j}(s) = 1$ for all $i \in \mathcal{N}$ and $s \in \mathcal{S}$. We call it the *canonical* decomposition of $\{Q_i^V\}_{i\in\mathcal{N}}$ when $Q_{i,j}^V$ can be represented as above with $\lambda_{i,j}(s) = 1/|\mathcal{E}_{Q,i}|$ for $j \in \mathcal{E}_{Q,i}$.

Proposition 2 naturally follows from the proof of Proposition 1 (a).

**Proposition 2 - Finite-horizon version** (Decomposition of $(Q_{h,i}^V)_{i\in\mathcal{N}}$). For a finite-horizon NMG with $\mathcal{G} = (\mathcal{N}, \mathcal{E}_Q)$ with $\mathcal{N}_C \neq \emptyset$, we know $\mathbb{P}_h(s' \mid s, \boldsymbol{a}) = \sum_{i\in\mathcal{N}_C}\mathbb{F}_{h,i}(s' \mid s, a_i)$, and $r_{h,i}(s, \boldsymbol{a}) = \sum_{j\in\mathcal{E}_{Q,i}} r_{h,i,j}(s, a_i, a_j)$ for some $\{\mathbb{F}_{h,i}\}_{i\in\mathcal{N}_C}$ and $\{r_{h,i,j}\}_{(i,j)\in\mathcal{E}_Q}$, then the $Q_{h,i,j}^V$ given in Definition 1 can be represented as

$$Q_{h,i,j}^V(s, a_i, a_j) = r_{h,i,j}(s, a_i, a_j)$$
$$+ \sum_{s'\in\mathcal{S}} \left(\mathbf{1}(j \in \mathcal{N}_C)\mathbb{F}_{h,j}(s' \mid s, a_j) + \mathbf{1}(i \in \mathcal{N}_C)\lambda_{h,i,j}(s)\mathbb{F}_{h,i}(s' \mid s, a_i)\right) V_{h+1}(s')$$

for any non-negative $(\lambda_{h,i,j}(s))_{(i,j)\in\mathcal{E}_Q}$ such that $\sum_{j\in\mathcal{E}_{Q,i}}\lambda_{h,i,j}(s) = 1$ for all $i \in \mathcal{N}$, $s \in \mathcal{S}$, and $h \in [H]$. For a finite-horizon NMG with $\mathcal{G} = (\mathcal{N}, \mathcal{E}_Q)$ such that $\mathcal{N}_C = \emptyset$, $\mathbb{P}_h(s' \mid s, \boldsymbol{a}) = \mathbb{F}_{h,o}(s' \mid s)$, and $r_{h,i}(s, \boldsymbol{a}) = \sum_{j\in\mathcal{E}_{Q,i}} r_{h,i,j}(s, a_i, a_j)$ for some $\mathbb{F}_{h,o}$ and $\{r_{h,i,j}\}_{(i,j)\in\mathcal{E}_Q}$, the $Q_{h,i,j}^V$ given in Definition 1 can be represented as

$$Q_{h,i,j}^V(s, a_i, a_j) = r_{h,i,j}(s, a_i, a_j) + \sum_{s'\in\mathcal{S}} \left(\lambda_{h,i,j}(s)\mathbb{F}_{h,o}(s' \mid s)\right) V_{h+1}(s')$$

for any non-negative $(\lambda_{h,i,j}(s))_{(i,j)\in\mathcal{E}_Q}$ such that $\sum_{j\in\mathcal{E}_{Q,i}}\lambda_{h,i,j}(s) = 1$ for all $i \in \mathcal{N}$, $s \in \mathcal{S}$, and $h \in [H]$. We call it the *canonical* decomposition of $Q$-value functions when $Q_{h,i,j}^V$ can be represented as above with $\lambda_{h,i,j}(s) = 1/|\mathcal{E}_{Q,i}|$ for $j \in \mathcal{E}_{Q,i}$.

## B.2   An alternative definition of NMGs

**Definition 3** (An alternative definition of NMGs). An infinite-horizon $\gamma$-discounted MG is called a *Multi-player MG with Networked separable interactions (NMG)* characterized by a tuple

$$(\mathcal{G} = (\mathcal{N}, \mathcal{E}_Q), \mathcal{S}, \mathcal{A}, \mathbb{P}, (r_i)_{i\in\mathcal{N}}, \gamma)$$

if for any *policy* $\pi$, there exist a set of functions $(Q_{i,j}^\pi)_{(i,j)\in\mathcal{E}_Q}$ and an undirected connected graph $\mathcal{G} = (\mathcal{N}, \mathcal{E}_Q)$ such that $Q_i^\pi(s, \boldsymbol{a}) = \sum_{j\in\mathcal{E}_{Q,i}} Q_{i,j}^\pi(s, a_i, a_j)$ holds for every $i \in \mathcal{N}$, $s \in \mathcal{S}$, $\boldsymbol{a} \in \mathcal{A}$. A finite-horizon MG is called a *Multi-player MG with Networked separable interactions* if for any *policy* $\pi$, there exist a set of functions $(Q_{h,i,j}^\pi)_{(i,j)\in\mathcal{E}_Q, h\in[H]}$ such that $Q_{h,i}^\pi(s, \boldsymbol{a}) = \sum_{j\in\mathcal{E}_{Q,i}} Q_{h,i,j}^\pi(s, a_i, a_j)$ holds for every $i \in \mathcal{N}$, $s \in \mathcal{S}$, $\boldsymbol{a} \in \mathcal{A}$, and $h \in [H]$.

The "if" condition of Proposition 1 also holds for this definition. However, the proof for the "only if" condition of Proposition 1 uses the functional derivative argument. We cannot use the functional derivative here directly, since $\boldsymbol{V}^\pi := (V^\pi(s))_{s\in\mathcal{S}}$ cannot represent all vectors in $[0, R/(1-\gamma)]^{|\mathcal{S}|}$. To be specific, we have

$$Q_i^\pi(s, \boldsymbol{a}) = \sum_{j\in\mathcal{E}_{Q,i}} Q_{i,j}^\pi(s, a_i, a_j) = r_i(s, \boldsymbol{a}) + \gamma\langle\mathbb{P}(\cdot \mid s, \boldsymbol{a}), V_i^\pi(\cdot)\rangle$$

for any policies $\pi, \pi'$, which indicates

$$\sum_{j\in\mathcal{E}_{Q,i}} (Q_{i,j}^\pi(s, a_i, a_j) - Q_{i,j}^{\pi'}(s, a_i, a_j)) = \gamma\langle\mathbb{P}(\cdot \mid s, \boldsymbol{a}), V_i^\pi(\cdot) - V_i^{\pi'}(\cdot)\rangle. \qquad (8)$$

If we can find a set of policies $(\pi^{(k)})_{k\in[|\mathcal{S}|]}$ such that the vectors in $\{\boldsymbol{V}_i^{\pi^{(k)}} - \boldsymbol{V}_i^{\pi'}\}_{k\in[|\mathcal{S}|]}$ are independent for some fixed $\pi'$, then we can concatenate the vectors $\{\boldsymbol{V}_i^{\pi^{(k)}} - \boldsymbol{V}_i^{\pi'}\}_{k\in[|\mathcal{S}|]}$ together as a matrix of size $|\mathcal{S}| \times |\mathcal{S}|$ that is full-rank, and solve for $\mathbb{P}(\cdot \mid s, \boldsymbol{a})$ by solving the linear equations (8), for some fixed $(s, \boldsymbol{a})$. This way, we can show that $\mathbb{P}(\cdot \mid s, \boldsymbol{a}) = \sum_{j\in\mathcal{E}_{Q,i}} \mathbb{F}_{i,j}(\cdot \mid s, a_i, a_j)$, and the rest of the proof follows from that of Proposition 1. However, such a set of policies $(\pi^{(k)})_{k\in[|\mathcal{S}|]}$ (and $\pi'$) may not exist in some degenerate cases, as to be detailed below.

**Definition 4** (Degenerate MG with respect to player $i$)**.** We call an MG *degenerate* with respect to player $i \in \mathcal{N}$ if there exists some $s \in \mathcal{S}$ such that for any $\pi$, $Q_i^\pi(s, \boldsymbol{a})$ is a constant function of $\boldsymbol{a} \in \mathcal{A}$.

**Definition 5** (Non-degenerate MG)**.** We call an MG *non-degenerate* if an MG is not degenerate with respect to any player $i \in \mathcal{N}$.

Now, we are ready to state the counterpart of Proposition 1 in these non-degenerate cases. The proof for the "if" direction is exactly the same as that of Proposition 1. We focus on the proof of the "only if" statement.

**Proposition 1 - An alternative definition version.** For a given graph $\mathcal{G} = (\mathcal{N}, \mathcal{E}_Q)$, a *non-degenerate* MG $(\mathcal{N}, \mathcal{S}, \mathcal{A}, \mathbb{P}, (r_i)_{i \in \mathcal{N}}, \gamma)$ (in the sense of Definition 5) with more than two players is an NMG with $\mathcal{G}$ if and only if: (1) $r_i(s, a_i, \cdot)$ **is decomposable with respect to** $\mathcal{E}_{Q,i}$ for each $i \in \mathcal{N}, s \in \mathcal{S}, a_i \in \mathcal{A}_i$, i.e., $r_i(s, \boldsymbol{a}) = \sum_{j \in \mathcal{E}_{Q,i}} r_{i,j}(s, a_i, a_j)$ for a set of functions $\{r_{i,j}(s, a_i, \cdot)\}_{j \in \mathcal{E}_{Q,i}}$, and (2) **the transition dynamics** $\mathbb{P}(s' \mid s, \cdot)$ **is decomposable with respect to** the $\mathcal{N}_C$ corresponding to this $\mathcal{G}$, i.e., $\mathbb{P}(s' \mid s, \boldsymbol{a}) = \sum_{i \in \mathcal{N}_C} \mathbb{F}_i(s' \mid s, a_i)$ for a set of functions $\{\mathbb{F}_i(s' \mid s, \cdot)\}_{i \in \mathcal{N}_C}$ if $\mathcal{N}_C \neq \emptyset$, or $\mathbb{P}(s' \mid s, \boldsymbol{a}) = \mathbb{F}_o(s' \mid s)$ for some constant function (of $\boldsymbol{a}$) $\mathbb{F}_o(s' \mid s)$ if $\mathcal{N}_C = \emptyset$.

*Proof.* Proofs of the claims are deferred to Section B.3 to preserve the flow of the argument.

**Claim 1.** For player $i \in \mathcal{N}$, if there exists a policy $\pi$ such that for every $s \in \mathcal{S}$, there exist $\boldsymbol{a}_{s,1}, \boldsymbol{a}_{s,2}$ that make $Q_i^\pi(s, \boldsymbol{a}_{s,1}) \neq Q_i^\pi(s, \boldsymbol{a}_{s,2})$, then we can construct $|\mathcal{S}|$ number of policies $(\pi^{(k)})_{k \in [|\mathcal{S}|]}$ such that $\{\boldsymbol{V}_i^{\pi^{(k)}} - \boldsymbol{V}_i^\pi\}_{k \in [|\mathcal{S}|]}$ are independent for any fixed $\pi$.

Therefore, for player $i \in \mathcal{N}$, if the condition of Claim 1 holds, then we can guarantee that there exist some functions $\{\mathbb{G}_{i,j}\}_{j \in \mathcal{E}_{Q,i}}$ such that $\mathbb{P}(\cdot|s, \boldsymbol{a}) = \sum_{j \in \mathcal{E}_{Q,i}} \mathbb{G}_{i,j}(\cdot|s, a_i, a_j)$, by the argument after Equation (8). If we assume that the condition of Claim 1 holds for every player $i \in \mathcal{N}$, then there exist some functions $\{\mathbb{G}_{i,j}\}_{(i,j) \in \mathcal{E}_Q}$ such that $\mathbb{P}(\cdot|s, \boldsymbol{a}) = \sum_{j \in \mathcal{E}_{Q,i}} \mathbb{G}_{i,j}(\cdot|s, a_i, a_j)$ for every $i \in \mathcal{N}$. Then, we can prove the decomposability of $\mathbb{P}(s' \mid s, \cdot)$ with respect to $\mathcal{N}_C$ with the same steps as Equations (3) - (7).

Hence, by Claim 1, we only need to prove that if an MG is non-degenerate with respect to player $i$, then there exists a policy $\pi$ such that there exist $\boldsymbol{a}_{s,1}, \boldsymbol{a}_{s,2}$ that make $Q_i^\pi(s, \boldsymbol{a}_{s,1}) \neq Q_i^\pi(s, \boldsymbol{a}_{s,2})$ for every $s \in \mathcal{S}$. If $|\mathcal{A}| = 1$, then the transition dynamics are already decomposed, so we do not need to consider this case.

**Claim 2.** Assume $|\mathcal{A}| \geq 2$. For player $i \in \mathcal{N}$, assume that for any policy $\pi$, there exists a state $s_\pi \in \mathcal{S}$ such that $Q_i^\pi(s_\pi, \boldsymbol{a})$ is a constant function of $\boldsymbol{a}$. Then, there exists a state $s \in \mathcal{S}$ such that uniformly for any policy $\pi$, $Q_i^\pi(s, \boldsymbol{a})$ is a constant function of $\boldsymbol{a}$.

Claim 2 shows that if the assumption of Claim 1 does not hold for player $i \in \mathcal{N}$, then

$$\mathcal{S}_{\text{const},i} := \{s \mid \text{For any } \pi, Q_i^\pi(s, \boldsymbol{a}) \text{ is a constant function of } \boldsymbol{a}\}$$

is not an empty set. By Definition 4, if an MG is not degenerate with respect to player $i$, then $\mathcal{S}_{\text{const},i}$ is empty, and thus the conditions of Claim 1 always hold for player $i$. Therefore, if we assume the non-degeneracy of MG (Definition 5), by Claim 1 and the arguments immediately following it, $\mathbb{P}(s' \mid s, \cdot)$ is decomposable with respect to $\mathcal{N}_C$. □

**Remark 5** (Degenerate MG with respect to player $i$)**.** The degeneracy of MGs as defined above can indeed be rare. To illustrate, consider a seemingly degenerate scenario where for all $s \in \mathcal{S}$, $r_i(s, \boldsymbol{a})$ remains a constant function with respect to $\boldsymbol{a}$; even under such circumstances, it is possible for the MG to be non-degenerate with respect to player $i$. For example, assume that $\mathcal{N} = [2], \gamma = \frac{1}{2}$, $\mathcal{S} = \{-1, 1\}$ $\mathcal{A}_i = \{-1, 1\}$, $\mathbb{P}(s'|s, a_1, a_2) = \mathbf{1}(s' = sa_1a_2), r_1(s, \boldsymbol{a}) = r_2(s, \boldsymbol{a}) = (1 + s)$. Let $\pi$ satisfy $\pi(\boldsymbol{a} \mid 1) = \mathbf{1}(a_1 = 1, a_2 = -1), \pi(\boldsymbol{a} \mid -1) = \mathbf{1}(a_1 = -1, a_2 = -1)$. Then, we have $V_1^\pi(1) = 2 + \frac{1}{2}V_1^\pi(-1)$ and $V_1^\pi(-1) = \frac{1}{2}V_1^\pi(-1)$, which further means $V_1^\pi(-1) = 0$ and $V_1^\pi(1) = 2$, i.e., $V_1^\pi(s) = 1 + s$ always holds. As a result, $Q_1^\pi(1, \boldsymbol{a}) = 1 + s + \frac{1}{2}(sa_1a_2 + 1)$, which is not a constant function, and the game is thus non-degenerate with respect to agent $i$. Moreover, suppose that the policy, transition dynamics, and reward functions are randomly chosen. The measure of the event that the existence of $s \in \mathcal{S}$ such that $Q_i^\pi(s, \boldsymbol{a})$ is constant for all possible actions $\boldsymbol{a}$ under this randomly chosen policy, transition dynamics, and reward function is 0. This is primarily because

$Q_i^\pi(s, \cdot) : \mathcal{A} \to \mathbb{R}$ must lie on a particular hyperplane in the overall value function space, which takes measure 0. Oftentimes, different actions will transition to different states and yield different rewards, thereby generating almost unique $Q_i^\pi(s, \boldsymbol{a})$ values.

**Remark 6.** If we have more than two players that satisfy the condition of Claim 1, i.e., the MG is non-degenerate with respect to more than two players (instead of being non-degenerate with respect to all players), then we can still guarantee the decomposability of $\mathbb{P}(s' \mid s, \cdot)$ with respect to a set that is not necessarily the same as $\mathcal{N}_C$. As a byproduct of the decomposability of $\mathbb{P}(s' \mid s, \cdot)$, we can guarantee the decomposability of $r_i(s, a_i, \cdot)$, too.

### B.3 Deferred proof of the claims in Appendix B.2

We define $\Pi := \begin{bmatrix} \pi(s_1) & 0 & \cdots & 0 \\ 0 & \pi(s_2) & \cdots & 0 \\ \vdots & \vdots & \ddots & 0 \\ 0 & 0 & \cdots & \pi(s_{|\mathcal{S}|}) \end{bmatrix} \in \mathbb{R}^{|\mathcal{S}||\mathcal{A}| \times |\mathcal{S}|}$, where $\pi(s)$ is a column vector

for the policy at state $s$, $\boldsymbol{P} := (\mathbb{P}(s' \mid s, \boldsymbol{a}))_{((s,\boldsymbol{a}),s')}$, $\boldsymbol{R} := \begin{bmatrix} \boldsymbol{r}(s_1)^\mathsf{T} & 0 & \cdots & 0 \\ 0 & \boldsymbol{r}(s_2)^\mathsf{T} & \cdots & 0 \\ \vdots & \vdots & \ddots & 0 \\ 0 & 0 & \cdots & \boldsymbol{r}(s_{|\mathcal{S}|})^\mathsf{T} \end{bmatrix} \in$

$\mathbb{R}^{|\mathcal{S}| \times |\mathcal{S}||\mathcal{A}|}$, then we have $\boldsymbol{V}_i^\pi = (I - \gamma \Pi^\mathsf{T} \boldsymbol{P})^{-1} \boldsymbol{R} \Pi \boldsymbol{1} \in \mathbb{R}^{|\mathcal{S}|}$ where $\boldsymbol{1} \in \mathbb{R}^{|\mathcal{S}|}$ is $(1, \ldots, 1)^\mathsf{T}$.

*Proof of Claim 1.* If we differentiate $\boldsymbol{V}_i^\pi$ with respect to $\pi$ along the direction $\Delta_{(s,\boldsymbol{a}_1,\boldsymbol{a}_2)} := e_{\boldsymbol{a}_1,s} - e_{\boldsymbol{a}_2,s} \in \mathbb{R}^{|\mathcal{A}||\mathcal{S}|}$ for some $\boldsymbol{a}_1, \boldsymbol{a}_2 \in \mathcal{A}$, i.e., the direction that increases (or decreases) $\pi(\boldsymbol{a}_1 \mid s)$ and decreases (or increases) $\pi(\boldsymbol{a}_2 \mid s)$, respectively, then by computation, we can have the following directional derivative:

$$\nabla_{\Delta_{(s,\boldsymbol{a}_1,\boldsymbol{a}_2)}} \boldsymbol{V}_i^\pi$$

$$= (I - \gamma \Pi^\mathsf{T} \boldsymbol{P})^{-1} \left( \sum_{s' \in \mathcal{S}} (\mathbb{P}(s' \mid s, \boldsymbol{a}_1) V_i^\pi(s') - \mathbb{P}(s' \mid s, \boldsymbol{a}_2) V_i^\pi(s')) + r_i(s, \boldsymbol{a}_1) - r_i(s, \boldsymbol{a}_2) \right) e_s$$

$$= (Q_i^\pi(s, \boldsymbol{a}_1) - Q_i^\pi(s, \boldsymbol{a}_2))(I - \gamma \Pi^\mathsf{T} \boldsymbol{P})^{-1} e_s. \tag{9}$$

Therefore, if there exists a policy $\pi$ such that for every $s$, there exist $\boldsymbol{a}_{s,1}, \boldsymbol{a}_{s,2}$ that make $Q_i^\pi(s, \boldsymbol{a}_{s,1}) \neq Q_i^\pi(s, \boldsymbol{a}_{s,2})$, then deviating from $\pi$ along the $\Delta_{(s,\boldsymbol{a}_{s,1},\boldsymbol{a}_{s,2})}$ direction for every $s$ provides $\{\boldsymbol{V}_i^{\pi^{(k)}} - \boldsymbol{V}_i^\pi\}_{k \in [|\mathcal{S}|]}$ vectors that are independent of each other, since $(I - \gamma \Pi^\mathsf{T} \boldsymbol{P})^{-1}$ is an invertible matrix for any $\Pi$. □

*Proof of Claim 2.* Note that $\pi \in \Delta(\mathcal{A})^{|\mathcal{S}|}$. For all $s \in \mathcal{S}$, define $\Pi_s$ as

$$\Pi_s := \{\pi \mid Q_i^\pi(s, \boldsymbol{a}) \text{ is a constant function of } \boldsymbol{a}\},$$

where we omit the dependence on $i$ as we focus on the discussion on a specific $i$ here. For any $\pi$, there exists a state $s_\pi$ such that $Q_i^\pi(s_\pi, \boldsymbol{a})$ is a constant function of $\boldsymbol{a}$, which means that $\pi \in \Pi_{s_\pi}$ and thus $\Pi_{s_\pi} \neq \emptyset$, which further yields $\sum_{s \in \mathcal{S}}$ (measure of $(\Pi_s)$) $\geq$ (measure of the whole space of $(\Delta(\mathcal{A})^{|\mathcal{S}|})$) $> 0$. By the pigeonhole principle, we know that there exists some $s \in \mathcal{S}$ such that (measure of $(\Pi_s)$) $> 0$ since $\mathcal{S}$ is a finite set.

Note that for the above $s$ such that (measure of $(\Pi_s)$) $> 0$, for any pair of $\boldsymbol{a}_1, \boldsymbol{a}_2 \in \mathcal{A}$, $Q_i^\pi(s, \boldsymbol{a}_1) - Q_i^\pi(s, \boldsymbol{a}_2)$ can be represented by the ratio of polynomials of $\pi$ while it has a non-zero measure set of solution, we can conclude that $Q_i^\pi(s, \boldsymbol{a}_1) - Q_i^\pi(s, \boldsymbol{a}_2) = 0$ for every $\pi \in \Delta(\mathcal{A})^{|\mathcal{S}|}$, $\boldsymbol{a}_1, \boldsymbol{a}_2 \in \mathcal{A}$ for $s$ such that (measure of $(\Pi_s)$) $> 0$. Therefore, for the $s$ such that (measure of $(\Pi_s)$) $> 0$, we have that for every $\pi$, $Q_i^\pi(s, \boldsymbol{a})$ is a constant function of $\boldsymbol{a}$. □

### B.4 Counterexample for Alternative Definition 3

We now show via a counterexample that the alternative definition given in Definition 3 may not even preserve the networked separable structure of the reward functions in general.

Consider a Markov game with the following specifications: $\mathcal{N} = [3]$, $\gamma = \frac{1}{2}$, $\mathcal{S} = \{s_1, s_2, s_3\}$, $\mathcal{A}_i = \{0, 1\}$, $\mathbb{P}(s_3 \mid s_3, \boldsymbol{a}) = \mathbb{P}(s_2 \mid s_2, \boldsymbol{a}) = 1$ for any $\boldsymbol{a} \in \mathcal{A}$, $\mathbb{P}(s_2 \mid s_1, \boldsymbol{a}) = \frac{1}{2} + \frac{1}{2} \cdot (-1)^{a_1 a_2 a_3}$, $\mathbb{P}(s_3 \mid s_1, \boldsymbol{a}) = \frac{1}{2} - \frac{1}{2} \cdot (-1)^{a_1 a_2 a_3}$, $r_1(s_3, \boldsymbol{a}) = 1$ and $r_1(s_2, \boldsymbol{a}) = 2$ for any $\boldsymbol{a} \in \mathcal{A}$, and $r_1(s_1, \boldsymbol{a}) = \frac{1}{2} - \frac{1}{2} \cdot (-1)^{a_1 a_2 a_3}$. Then, we have by definition that for any $\boldsymbol{a} \in \mathcal{A}$, $Q_1^\pi(s_2, \boldsymbol{a}) = V_1^\pi(s_2) = \frac{1}{1-\gamma} r_1(s_2, \boldsymbol{a}) = 4$, $Q_1^\pi(s_3, \boldsymbol{a}) = V_1^\pi(s_3) = \frac{1}{1-\gamma} r_1(s_3, \boldsymbol{a}) = 2$, and

$$Q_1^\pi(s_1, \boldsymbol{a}) = r_1(s_1, \boldsymbol{a}) + \gamma \mathbb{P}(s_3 \mid s_1, \boldsymbol{a}) V_1^\pi(s_3) + \gamma \mathbb{P}(s_2 \mid s_1, \boldsymbol{a}) V_1^\pi(s_2)$$
$$= r_1(s_1, \boldsymbol{a}) + \frac{1}{2} \cdot (\frac{1}{2} - \frac{1}{2} \cdot (-1)^{a_1 a_2 a_3}) \cdot 2 + \frac{1}{2} \cdot (\frac{1}{2} + \frac{1}{2} \cdot (-1)^{a_1 a_2 a_3}) \cdot 4 = 2.$$

Hence, $Q_1^\pi(s, a_1, \cdot)$ is decomposable with respect to $\{2, 3\}$, however, the reward $r_1(s, a_1, \cdot)$ is not, due to $r_1(s_1, \boldsymbol{a})$. Note that this counterexample exactly exhibits the importance of the ergodicity of the Markov chain in removing the degeneracy.

### B.5 Examples of (zero-sum) NMGs

**Example 2 (Markov security games).** Security games as described in [103, 24] is a primary example of zero-sum NGs/polymatrix games, which features two types of players: *attackers* who work as a group ($\mathfrak{a}$), and *users* ($\mathfrak{u}$). Let $\mathfrak{U}$ denote the set of all users. We construct a star-shaped network (c.f. Figure 2) with the attacker group including $n_\mathfrak{a}$ number of attackers sitting at the center, connected to each user. There is an IP address set $[C]$. We define the action spaces for each user $\mathfrak{u}_i$ and the attacker group as $\mathcal{A}_{\mathfrak{u}_i} = [C]$ and $\mathcal{A}_\mathfrak{a} = \{T \mid T \subseteq [C], |T| = n_\mathfrak{a}\}$, respectively. Each user selects one IP address, while the attacker group selects a subset $I \subseteq [C]$. For each user whose IP address is attacked, the attacker group gains one unit of payoff, and the attacked user loses one unit. Conversely, if a user's IP address is not attacked, the user earns one unit of payoff, and the attacker loses one unit.

We naturally extend the security games to *Markov security games* as follows: we define state $s \in \mathcal{S} = \mathbb{R}^C$ by setting $s_0 = \mathbf{0}$ and $s_{t+1} \sim s_t + \text{Unif}((e_{a_{\mathfrak{u}_i, t}})_{\mathfrak{u}_i \in \mathfrak{U}})$, representing the vector of *security level* for the IP addresses. Specifically, a vaccine program can improve the security level of each IP address if it has been attacked previously. We define $X \in \mathbb{R}^C$ as a vector such that each of its components, $X_c$, corresponds to a unique user's IP address, indexed by $c$. Each $X_c$ is defined by the random variable as $X_c \sim 2\text{Bern}(1 - 1/(s_{t,c} + 1)) - 1$, indicating the outcome of a potential attack on IP address $c \in [C]$. Here, $s_{t,c}$ denotes the security level of each IP address $c$ at a given time $t$, i.e., the $c$-th component of $s_t$. The success probability of an attack on an IP address is inversely proportional to its security level, represented by $1/(s_{t,c} + 1)$. Therefore, higher security levels make an attack less likely to succeed. The term $2\text{Bern}(1 - 1/(s_{t,c} + 1)) - 1$ describes a Bernoulli distribution, typically taking values 0 or 1, that has been scaled and shifted to take values $-1$ or 1 instead. Here, $-1$ represents an unsuccessful attack, while 1 denotes a successful attack on the IP address $c$. Therefore, each $X_c$ provides a probabilistic view of the failure of an attack on each IP address, given its security level. For each $(s, \boldsymbol{a})$, the reward functions for the users and the attacker group are defined as $r_{\mathfrak{u}_i}(s, \boldsymbol{a}, I) = r_{\mathfrak{u}_i, \mathfrak{a}}(s, a_{\mathfrak{u}_i}, I) = \mathbf{1}(a_{\mathfrak{u}_i} \in I) X_{a_{\mathfrak{u}_i}} + \mathbf{1}(a_{\mathfrak{u}_i} \notin I)$ and $r_\mathfrak{a}(s, \boldsymbol{a}, I) = \sum_{\mathfrak{u}_i \in \mathfrak{U}} r_{\mathfrak{a}, \mathfrak{u}_i}(s, a_{\mathfrak{u}_i}, I) - \mathbf{1}(a_{\mathfrak{u}_i} \notin I)$ where $r_{\mathfrak{a}, \mathfrak{u}_i}(s, a_{\mathfrak{u}_i}, I) = -\mathbf{1}(a_{\mathfrak{u}_i} \in I) X_{a_{\mathfrak{u}_i}}$. The reward function of users can be interpreted as follows: if the user's action $a_{\mathfrak{u}_i}$ is in the set of attacked IP addresses $I$ and the attack failed (i.e., $X_{a_{\mathfrak{u}_i}} = 1$), then the user receives a reward equal to 1. Otherwise, if the user's action is not in $I$, the user also receives a reward of 1, likely representing a successful defense or evasion of an attack. Since the reward is always zero-sum, this game is a zero-sum NMG with networked separable interactions.

**Example 3 (Global economy)** . Macroeconomic dynamics may also be modeled through either zero-sum NMGs or NMGs. Trading between nations has been analyzed in game theory [104, 105]. We consider nations as players, each nation has an action space, $\mathcal{A}_i = \mathbb{R}$, and the actions decide their expenditure levels. We define the *state* of the global economy, $s \in \mathbb{R}$, such that $s_0 = 0$ and $s_{t+1} \sim s_t + \text{Unif}((a_{c,t})_{c \in \mathcal{C}}) + Z_t$. Here, $Z_t$ is a random variable representing the unpredictable nature of global events (e.g., COVID-19), and $\mathcal{C}$ represents the set of *powerful* nations, which models the fact that powerful nations' politics or military spending have a relatively significant impact on global economy [106, 107]. The aggregated (or ensemble) effect of the powerful nations on the economy is modeled by the term $\text{Unif}((a_{c,t})_{c \in \mathcal{C}})$.

During the global financial crisis in 2008-2009, many nations implemented significant fiscal stimulus measures to counteract the downturn [108, 109]. Conversely, in good economic conditions, the estimated government spending multipliers were less than one, suggesting that the increased government spending in such situations might not have the intended positive effects on the economy [110]. Such a state-dependence on reward functions may be modeled as follows. First, we consider the reward being decomposable with respect to nations, as it can be interpreted as (1) the expenditure of each nation is related to the amount of payment spent on trading, and (2) we focus on the case with *bilateral* trading, where the surplus from trading can be decomposed by the surplus from the *pairwise* trading with other nations. Second, as mentioned above, the relationship between government spending and the global economy can be seen as *countercyclical* [110], which we use the formula $s(a_j - a_i)$ to model explicitly, for nation $i$. Specifically, $s > 0$ denotes a good economic condition, in which all the nations may choose to decrease the expenditure level (the $-a_i$ term). Hence, the reward function for nation $i$ can be written as $r_i(s, \boldsymbol{a}) = \sum_{j \in \mathcal{N}} r_{i,j}(s, a_i, a_j) = \texttt{Const} + \sum_{j \in \mathcal{N}} s(a_j - a_i)$, where the positive constant $\texttt{Const}$ represents the net benefit out of the tradings. Hence, the game shares the characteristics of being a constant-sum NMG. Moreover, other alternative forms of the reward functions may exist to reflect the countercyclical phenomenon, and may not necessarily satisfy the zero-sum (constant-sum) property, but the game would still qualify as an NMG.

## B.6 Reviewing existing results for zero-sum NGs

In a zero-sum NG, i.e., a zero-sum polymatrix game, for each player $i \in \mathcal{N}$, the reward $r_i(a_i, \pi_{-i})$ of using a pure strategy $a_i \in \mathcal{A}_i$ is a linear function of $\pi_{-i}$. The following linear program, which involves variables $\pi \in \prod_{i \in \mathcal{N}} \Delta(\mathcal{A}_i)$ and $\boldsymbol{v} = (v_i)_{i \in \mathcal{N}}$, aims to minimize the sum of the variables $v_i$:

$$\begin{aligned}
\min_{\boldsymbol{v}, \pi} \quad & \sum_{i \in \mathcal{N}} v_i \\
\text{subject to} \quad & v_i \geq r_i(e_{a_i}, \pi_{-i}), \quad \text{for all } i \in \mathcal{N}, a \in \mathcal{A}_i, \\
& \pi \in \prod_{i \in \mathcal{N}} \Delta(\mathcal{A}_i).
\end{aligned}$$

Reference [24] states that if $(\pi^\star, \boldsymbol{v}^\star)$ is an optimal solution to the above linear program, then $\pi^\star$ is an NE of the zero-sum NG, and the optimal value of the above linear program is 0. Conversely if $\pi^\star$ is an NE, then there exists a $\boldsymbol{v}^\star$ such that $(\pi^\star, \boldsymbol{v}^\star)$ is an optimal solution to the above linear program and $\boldsymbol{v}^\star$ is the expected reward vector under $\pi^\star$. By observation, we additionally have the following proposition as an extension:

**Proposition 5.** If $(\pi^\star, \boldsymbol{v}^\star)$ is an $\epsilon$-optimal solution to the above linear program, then $\pi^\star$ is an $\epsilon$-NE and $r_i(\pi^\star) \leq v_i^\star \leq r_i(\pi^\star) + \epsilon$ for all $i \in \mathcal{N}$. Conversely, if $\pi^\star$ is an $\epsilon$-NE, then there exists a $\boldsymbol{v}^\star$ such that $(\pi^\star, \boldsymbol{v}^\star)$ is an $n\epsilon$-optimal solution to the above linear program.

*Proof.* If $(\pi^\star, \boldsymbol{v}^\star)$ is an $\epsilon$-optimal solution to the above linear program (whose optimal solution is exactly 0), we have

$$\epsilon \geq \sum_{i \in \mathcal{N}} v_i^\star \underset{(i)}{=} \sum_{i \in \mathcal{N}} (v_i^\star - r_i(\pi^\star)) \underset{(ii)}{\geq} \sum_{i \in \mathcal{N}} \left( \max_{\mu_i \in \Delta(\mathcal{A}_i)} r_i(\mu_i, \pi_{-i}^\star) - r_i(\pi^\star) \right)$$

which proves that $\pi^\star$ is an $\epsilon$-NE: here $(i)$ holds since the sum of reward over players is zero, and $(ii)$ holds due to the constraint of the given linear program. Moreover, we can also observe that $r_i(\pi^\star) \leq v_i^\star \leq r_i(\pi^\star) + \epsilon$ holds, since $0 \leq \max_{\mu_i \in \Delta(\mathcal{A}_i)} r_i(\mu_i, \pi_{-i}^\star) - r_i(\pi^\star) \leq v_i^\star - r_i(\pi^\star) \leq \epsilon$ for each $i \in \mathcal{N}$.

Conversely, suppose that $\pi^\star$ is an $\epsilon$-NE. Then, defining $v_i^\star := \max_{\mu_i \in \Delta(\mathcal{A}_i)} r_i(\mu_i, \pi_{-i}^\star)$ satisfies the constraints of the given linear program. In addition, we have

$$\sum_{i \in \mathcal{N}} v_i^\star = \sum_{i \in \mathcal{N}} (v_i^\star - r_i(\pi^\star)) = \sum_{i \in \mathcal{N}} \left( \max_{\mu_i \in \Delta(\mathcal{A}_i)} r_i(\mu_i, \pi_{-i}^\star) - r_i(\pi^\star) \right) \leq n\epsilon,$$

which concludes the theorem. $\qquad \square$

**Proposition 6.** Suppose $\pi^\star$ is an $\epsilon$-approximate CCE of the zero-sum NG. Then the product of its marginalized policy $\widehat{\pi}^\star$ is an $n\epsilon$-approximate NE of the zero-sum NG. Moreover, it holds that $r_i(\pi^\star) \geq r_i(\widehat{\pi}^\star) \geq r_i(\pi^\star) - n\epsilon$ for every $i \in \mathcal{N}$.

*Proof.* A similar method with [24]'s Theorem 2 can provide proof of Proposition 6. To be specific, define $v_i^\star := r_i(\pi^\star)$, then we have $(\widehat{\pi}^\star, \boldsymbol{v}^\star)$ is an $n\epsilon$-optimal solution. By Proposition 5, we can conclude that $\widehat{\pi}^\star$ is an $n\epsilon$-approximate NE and $r_i(\pi^\star) \geq r_i(\widehat{\pi}^\star) \geq r_i(\pi^\star) - n\epsilon$ for all $i \in \mathcal{N}$. $\quad\square$

We note that Proposition 5 and Proposition 6 are not in [24], but it plays an important role in proving Proposition 3.

## B.7 Omitted proof of Proposition 3

Before introducing the proof of Proposition 3, we provide the relationship between approximate Markov stationary CCE and approximate auxiliary-game CCE in Markov games.

**Claim 3.** For an infinite-horizon $\gamma$-discounted MG, an $\epsilon$-approximate Markov stationary CCE $\pi$ of this MG makes $\pi(s)$ an $\epsilon$-approximate CCE of the auxiliary game at each state $s \in \mathcal{S}$, where the auxiliary game payoff matrix at each $s \in \mathcal{S}$ is defined as $(r_i(s, \boldsymbol{a}) + \gamma\langle\mathbb{P}(\cdot|s, \boldsymbol{a}), V_i^\pi(\cdot)\rangle)_{\boldsymbol{a}\in\mathcal{A}}$ for player $i \in \mathcal{N}$.

*Proof.* By definition of $\epsilon$-approximate Markov (perfect) CCE, we have for all $s \in \mathcal{S}$ and all $i \in \mathcal{N}$ that

$$V_i^\pi(s) \leq \max_{\mu_i \in \Delta(\mathcal{A}_i)^{|\mathcal{S}|}} V_i^{\mu_i, \pi_{-i}}(s) \leq V_i^\pi(s) + \epsilon. \tag{10}$$

Then, by one-step of Bellman equation for $\max_{\mu_i \in \Delta(\mathcal{A}_i)^{|\mathcal{S}|}} V_i^{\mu_i, \pi_{-i}}(s)$, we also know that

$$\max_{\mu_i \in \Delta(\mathcal{A}_i)^{|\mathcal{S}|}} V_i^{\mu_i, \pi_{-i}}(s) = \max_{\nu \in \Delta(\mathcal{A}_i)} \mathbb{E}_{\nu, \pi_{-i}(s)}\left[r_i(s, \boldsymbol{a}) + \gamma\langle\mathbb{P}(\cdot \,|\, s, \boldsymbol{a}), \max_{\mu_i \in \Delta(\mathcal{A}_i)^{|\mathcal{S}|}} V_i^{\mu_i, \pi_{-i}}(\cdot)\rangle\right]. \tag{11}$$

Moreover, we have by the left inequality of Equation (10) that

$$\max_{\nu \in \Delta(\mathcal{A}_i)} \mathbb{E}_{\nu, \pi_{-i}(s)}\left[r_i(s, \boldsymbol{a}) + \gamma\langle\mathbb{P}(\cdot \,|\, s, \boldsymbol{a}), \max_{\mu_i \in \Delta(\mathcal{A}_i)^{|\mathcal{S}|}} V_i^{\mu_i, \pi_{-i}}(\cdot)\rangle\right]$$
$$\geq \max_{\nu \in \Delta(\mathcal{A}_i)} \mathbb{E}_{\nu, \pi_{-i}(s)}\left[r_i(s, \boldsymbol{a}) + \gamma\langle\mathbb{P}(\cdot \,|\, s, \boldsymbol{a}), V_i^\pi(\cdot)\rangle\right]. \tag{12}$$

Also, we have by one-step Bellman consistency equation for $V_i^\pi(s)$ that

$$V_i^\pi(s) = \mathbb{E}_{\pi(s)}\left[r_i(s, \boldsymbol{a}) + \gamma\langle\mathbb{P}(\cdot \,|\, s, \boldsymbol{a}), V_i^\pi(\cdot)\rangle\right]. \tag{13}$$

Combining Equations (10), (12) and (13), we have

$$\max_{\nu \in \Delta(\mathcal{A}_i)} \mathbb{E}_{\nu, \pi_{-i}(s)}\left[r_i(s, \boldsymbol{a}) + \gamma\langle\mathbb{P}(\cdot \,|\, s, \boldsymbol{a}), V_i^\pi(\cdot)\rangle\right] \leq \mathbb{E}_{\pi(s)}\left[r_i(s, \boldsymbol{a}) + \gamma\langle\mathbb{P}(\cdot \,|\, s, \boldsymbol{a}), V_i^\pi(\cdot)\rangle\right] + \epsilon,$$

for all $i \in \mathcal{N}$ and $s \in \mathcal{S}$, which proves that $\pi(s)$ is an $\epsilon$-approximate CCE of the auxiliary game, where the game payoff matrix at each state $s \in \mathcal{S}$ is $(r_i(s, \boldsymbol{a}) + \gamma\langle\mathbb{P}(\cdot|s, \boldsymbol{a}), V_i^\pi(\cdot)\rangle)_{\boldsymbol{a}\in\mathcal{A}}$ for player $i \in \mathcal{N}$. $\quad\square$

**Claim 4.** For an $H$-horizon MG and $h \in [H]$, an $\epsilon$-approximate Markov CCE $\pi = \{\pi_h\}_{h\in[H]}$ of this MG makes $\pi_h(s)$ an $\epsilon$-approximate CCE of the auxiliary game at each state $s \in \mathcal{S}$ and each $h \in [H]$, where the auxiliary game payoff matrix at $(s, h)$ is defined as $(r_{h,i}(s, \boldsymbol{a}) + \langle\mathbb{P}_h(\cdot|s, \boldsymbol{a}), V_{h+1,i}^\pi(\cdot)\rangle)_{\boldsymbol{a}\in\mathcal{A}}$ for player $i \in \mathcal{N}$, where $V_{H+1,i}^{\widetilde{\pi}}(s) = 0$ for any policy $\widetilde{\pi}$, and for all $s \in \mathcal{S}$ and $i \in \mathcal{N}$.

*Proof.* By definition of $\epsilon$-approximate Markov (perfect) CCE, we have that for all $s \in \mathcal{S}$, $h \in [H]$, and $i \in \mathcal{N}$

$$V_{h,i}^\pi(s) \leq \max_{\mu_i \in \Delta(\mathcal{A}_i)^{|\mathcal{S}|\times H}} V_{h,i}^{\mu_i, \pi_{-i}}(s) \leq V_{h,i}^\pi(s) + \epsilon. \tag{14}$$

Then, by one-step of Bellman equation for $\max_{\mu_i \in \Delta(\mathcal{A}_i)^{|\mathcal{S}| \times H}} V_{h,i}^{\mu_i, \pi_{-i}}(s)$, we also know that

$$\max_{\mu_i \in \Delta(\mathcal{A}_i)^{|\mathcal{S}| \times H}} V_{h,i}^{\mu_i, \pi_{-i}}(s)$$

$$= \max_{\nu \in \Delta(\mathcal{A}_i)} \mathbb{E}_{\nu, \pi_{h,-i}(s)} \left[ r_{h,i}(s, \boldsymbol{a}) + \langle \mathbb{P}_h(\cdot \mid s, \boldsymbol{a}), \max_{\mu_i \in \Delta(\mathcal{A}_i)^{|\mathcal{S}| \times H}} V_{h+1,i}^{\mu_i, \pi_{-i}}(\cdot) \rangle \right]. \tag{15}$$

Moreover, we have by the left inequality of Equation (14) and $V_{H+1,i}^{\widetilde{\pi}}(s) = 0$ for any $\widetilde{\pi}$ that

$$\max_{\nu \in \Delta(\mathcal{A}_i)} \mathbb{E}_{\nu, \pi_{h,-i}(s)} \left[ r_{h,i}(s, \boldsymbol{a}) + \langle \mathbb{P}_h(\cdot \mid s, \boldsymbol{a}), \max_{\mu_i \in \Delta(\mathcal{A}_i)^{|\mathcal{S}| \times (H-1)}} V_{h+1,i}^{\mu_i, \pi_{-i}}(\cdot) \rangle \right] \tag{16}$$

$$\geq \max_{\nu \in \Delta(\mathcal{A}_i)} \mathbb{E}_{\nu, \pi_{h,-i}(s)} \left[ r_{h,i}(s, \boldsymbol{a}) + \langle \mathbb{P}_h(\cdot \mid s, \boldsymbol{a}), V_{h+1,i}^{\pi}(\cdot) \rangle \right].$$

Also, we have by one-step Bellman equation for $V_{h,i}^{\pi}(s)$ that

$$V_{h,i}^{\pi}(s) = \mathbb{E}_{\pi_h(s)} \left[ r_{h,i}(s, \boldsymbol{a}) + \langle \mathbb{P}_h(\cdot \mid s, \boldsymbol{a}), V_{h+1,i}^{\pi}(\cdot) \rangle \right]. \tag{17}$$

Combining Equations (14), (15) and (17), we can conclude the theorem. $\square$

**Proposition 3.** Given an $\epsilon$-approximate Markov CCE of an infinite-horizon $\gamma$-discounted zero-sum NMG, marginalizing it at each state results in an $\frac{(n+1)}{(1-\gamma)}\epsilon$-approximate Markov NE of the zero-sum NMG. The same argument also holds for the finite-horizon episodic setting with $(1-\gamma)^{-1}$ being replaced by $H$.

*Proof.* For an arbitrary joint Markov policy $\mu$, we define $d_{s',\mu}(s) := (1 - \gamma)\mathbb{E}_{s_1=s',\mu}[\sum_{t=1}^{\infty} \gamma^{t-1}\mathbf{1}(s_t = s)]$ which is the discounted visitation measure of states when we follow policy $\mu$ and start from state $s'$. Here, $s_t$ denotes the state at timestep $t$. We use $\pi$ to denote an $\epsilon$-approximate Markov CCE, and $\widehat{\pi}$ to denote the product policy of the per-state marginalized policies of $\pi$ for all agents $i \in \mathcal{N}$. We define $\pi_i$ as the per-state marginalized policy for player $i$ from $\pi$. With this notation, we have $\widehat{\pi} := \pi_1 \times \cdots \times \pi_n$. Note that for zero-sum NMGs, by Proposition 1 we know that if $\mathcal{N}_C \neq \emptyset$, then $\mathbb{P}(s' \mid s, \boldsymbol{a}) = \sum_{i \in \mathcal{N}_C} \mathbb{F}_i(s' \mid s, a_i)$ or $\mathbb{P}_h(s' \mid s, \boldsymbol{a}) = \sum_{i \in \mathcal{N}_C} \mathbb{F}_{h,i}(s' \mid s, a_i)$, for infinite- and finite-horizon cases, respectively; and if $\mathcal{N}_C = \emptyset$, we have $\mathbb{P}(s' \mid s, \boldsymbol{a}) = \mathbb{F}_o(s' \mid s)$ or $\mathbb{P}_h(s' \mid s, \boldsymbol{a}) = \mathbb{F}_{h,o}(s' \mid s)$ for the two cases.

In the infinite-horizon case, by Claim 3, we know that $\pi(s)$ also serves as an $\epsilon$-approximate CCE for an auxiliary game with a payoff matrix $(r_i(s, \boldsymbol{a}) + \gamma \langle \mathbb{P}(\cdot \mid s, \boldsymbol{a}), V_i^{\pi}(\cdot) \rangle)_{\boldsymbol{a} \in \mathcal{A}}$ for player $i \in \mathcal{N}$. Since this auxiliary game is a zero-sum NG by the definition of zero-sum NMG, Proposition 6 implies that the policy $\widehat{\pi}(s)$ is an $n\epsilon$-approximate NE of the auxiliary game with the same payoff matrix, and the following inequality is valid for all $i \in \mathcal{N}$ and $s \in \mathcal{S}$:

$$V_i^{\pi}(s) = r_i(s, \pi) + \gamma \sum_{s' \in \mathcal{S}} \sum_{\boldsymbol{a} \in \mathcal{A}} \mathbb{P}(s' \mid s, \boldsymbol{a}) \pi(\boldsymbol{a} \mid s) V_i^{\pi}(s')$$

$$\leq r_i(s, \widehat{\pi}) + \gamma \sum_{s' \in \mathcal{S}} \sum_{\boldsymbol{a} \in \mathcal{A}} \mathbb{P}(s' \mid s, \boldsymbol{a}) \widehat{\pi}(\boldsymbol{a} \mid s) V_i^{\pi}(s') + n\epsilon. \tag{18}$$

Applying the inequality $V_i^{\pi}(s') \leq r_i(s', \widehat{\pi}) + \gamma \sum_{\widetilde{s} \in \mathcal{S}} \sum_{\boldsymbol{a} \in \mathcal{A}} \mathbb{P}(\widetilde{s} \mid s', \boldsymbol{a}) \widehat{\pi}(\boldsymbol{a} \mid s') V_i^{\pi}(\widetilde{s}) + n\epsilon$ into the final expression of Equation (18), and applying it recursively, we have that for every $i \in \mathcal{N}$ and $s \in \mathcal{S}$:

$$V_i^{\pi}(s) \leq V_i^{\widehat{\pi}}(s) + n\epsilon/(1-\gamma). \tag{19}$$

Moreover, we have that for any $\mu \in \Delta(\mathcal{A}_i)^{|\mathcal{S}|}$

$$V_i^{\widehat{\pi}}(s) \geq V_i^{\pi}(s) - n\epsilon/(1-\gamma) \geq V_i^{\mu, \pi_{-i}}(s) - (n+1)\epsilon/(1-\gamma)$$

$$= \mathbb{E}_{\boldsymbol{a} \sim \mu(s') \times \pi_{-i}(s'), s' \sim d_{s,\mu,\pi_{-i}}}[r_i(s', \boldsymbol{a})] - (n+1)\epsilon/(1-\gamma)$$

$$\underset{(i)}{=} \mathbb{E}_{\boldsymbol{a} \sim \mu(s') \times \widehat{\pi}_{-i}(s'), s' \sim d_{s,\mu,\pi_{-i}}}[r_i(s', \boldsymbol{a})] - (n+1)\epsilon/(1-\gamma)$$

$$\underset{(ii)}{=} \mathbb{E}_{\boldsymbol{a} \sim \mu(s') \times \widehat{\pi}_{-i}(s'), s' \sim d_{s,\mu,\widehat{\pi}_{-i}}}[r_i(s', \boldsymbol{a})] - (n+1)\epsilon/(1-\gamma)$$

$$= V_i^{\mu, \widehat{\pi}_{-i}}(s) - (n+1)\epsilon/(1-\gamma),$$

where the second inequality follows from $\pi$ being an $\epsilon$-approximate Markov CCE, and $(i)$ holds since for arbitrary $(\nu_i, \nu_{-i}) \in \Delta(\mathcal{A}_i)^{|\mathcal{S}|} \times \Delta(\mathcal{A}_{-i})^{|\mathcal{S}|}$ and for any $\nu_\mathcal{S} \in \Delta(\mathcal{S})$,

$$\mathbb{E}_{\boldsymbol{a} \sim \nu_i(s') \times \nu_{-i}(s'), s' \sim \nu_\mathcal{S}}[r_i(s', \boldsymbol{a})]$$

$$= \sum_{\boldsymbol{a} \in \mathcal{A}} \sum_{s' \in \mathcal{S}} \nu_\mathcal{S}(s') r_i(s', \boldsymbol{a}) \nu_i(a_i \mid s') \nu_{-i}(\boldsymbol{a}_{-i} \mid s')$$

$$= \sum_{j \in \mathcal{E}_{Q,i}} \sum_{\boldsymbol{a} \in \mathcal{A}} \sum_{s' \in \mathcal{S}} \nu_\mathcal{S}(s') r_{i,j}(s', a_i, a_j) \nu_i(a_i \mid s') \nu_{-i}(\boldsymbol{a}_{-i} \mid s')$$

$$= \sum_{j \in \mathcal{E}_{Q,i}} \sum_{a_i \in \mathcal{A}_i} \sum_{a_j \in \mathcal{A}_j} \sum_{s' \in \mathcal{S}} \nu_\mathcal{S}(s') r_{i,j}(s', a_i, a_j) \nu_i(a_i \mid s') \nu_j(a_j \mid s')$$

$$= \mathbb{E}_{\boldsymbol{a} \sim \nu_i(s') \times \widehat{\nu}_{-i}(s'), s' \sim \nu_\mathcal{S}}[r_i(s', \boldsymbol{a})],$$

where $\widehat{\nu} := \nu_1 \times \cdots \times \nu_n$ is the product policy of the per-state marginalized policies of $\nu$, and $(ii)$ holds due to the following fact: if $\mathcal{N}_C \neq \emptyset$

$$\mathbb{P}_\pi(s' \mid s) := \sum_{\boldsymbol{a} \in \mathcal{A}} \mathbb{P}(s' \mid s, \boldsymbol{a}) \pi(\boldsymbol{a} \mid s) = \sum_{\boldsymbol{a} \in \mathcal{A}} \sum_{i \in \mathcal{N}_C} \mathbb{F}_i(s' \mid s, a_i) \pi(\boldsymbol{a} \mid s)$$

$$= \sum_{i \in \mathcal{N}_C} \sum_{\boldsymbol{a} \in \mathcal{A}} \mathbb{F}_i(s' \mid s, a_i) \pi(\boldsymbol{a} \mid s) = \sum_{i \in \mathcal{N}_C} \sum_{a_i \in \mathcal{A}_i} \mathbb{F}_i(s' \mid s, a_i) \pi_i(a_i \mid s) =: \mathbb{P}_{\widehat{\pi}}(s' \mid s),$$

$$(20)$$

or if $\mathcal{N}_C = \emptyset$

$$\mathbb{P}_\pi(s' \mid s) = \mathbb{F}_o(s' \mid s) = \mathbb{P}_{\widehat{\pi}}(s' \mid s). \tag{21}$$

In other words, the marginalized policy's state visitation measure $d_{s,\widehat{\pi}}$ is the same as the original policy's state visitation measure $d_{s,\pi}$. Therefore, marginalizing $\epsilon$-approximate Markov CCE provides $(n+1)\epsilon/(1-\gamma)$-approximate Markov NE.

Moreover, a similar argument holds for the finite-horizon episodic setting. In the $H$-horizon case, since $\pi$ is an $\epsilon$-approximate Markov CCE, by Claim 4 $\pi_h(s)$ also serves as an $\epsilon$-approximate CCE for an auxiliary game with a payoff matrix defined by $(r_{h,i}(s, \boldsymbol{a}) + \langle \mathbb{P}_h(\cdot \mid s, \boldsymbol{a}), V^\pi_{h+1,i}(\cdot) \rangle)_{\boldsymbol{a} \in \mathcal{A}}$ for player $i \in \mathcal{N}$. Since this auxiliary game is a zero-sum NG by the definition of zero-sum NMG, Proposition 6 implies that the policy $\widehat{\pi}_h(s)$ is an $n\epsilon$-approximate NE of the auxiliary-game with the same payoff matrix, and the following inequality is valid for all $i \in \mathcal{N}$ and $s \in \mathcal{S}$:

$$V^\pi_{h,i}(s) = r_{h,i}(s, \pi) + \sum_{s' \in \mathcal{S}} \sum_{\boldsymbol{a} \in \mathcal{A}} \mathbb{P}_h(s' \mid s, \boldsymbol{a}) \pi_h(\boldsymbol{a} \mid s) V^\pi_{h+1,i}(s')$$

$$\leq r_{h,i}(s, \widehat{\pi}) + \sum_{s' \in \mathcal{S}} \sum_{\boldsymbol{a} \in \mathcal{A}} \mathbb{P}_h(s' \mid s, \boldsymbol{a}) \widehat{\pi}_h(\boldsymbol{a} \mid s) V^\pi_{h+1,i}(s') + n\epsilon. \tag{22}$$

Applying the inequality

$$V^\pi_{h+1,i}(s') \leq r_{h+1,i}(s', \widehat{\pi}) + \sum_{\widetilde{s} \in \mathcal{S}} \sum_{\boldsymbol{a} \in \mathcal{A}} \mathbb{P}_{h+1}(\widetilde{s} \mid s', \boldsymbol{a}) \widehat{\pi}_{h+1}(\boldsymbol{a} \mid s') V^\pi_{h+2,i}(\widetilde{s}) + n\epsilon$$

into (22) continually, iterating this procedure from $h+1$ to $H$, yields that for every $i \in \mathcal{N}$ and $s \in \mathcal{S}$:

$$V^\pi_{h,i}(s) \leq V^{\widehat{\pi}}_{h,i}(s) + hn\epsilon.$$

Moreover, we have that for any $\mu \in \Delta(\mathcal{A}_i)^{|\mathcal{S}| \times H}$

$$V^{\widehat{\pi}}_{h,i}(s) \geq V^\pi_{h,i}(s) - n\epsilon H \geq V^{\mu, \pi_{-i}}_{h,i}(s) - (n+1)\epsilon H = V^{\mu, \widehat{\pi}_{-i}}_{h,i}(s) - (n+1)\epsilon h,$$

with a similar observation on the visitation measure under $\pi$ and $\widehat{\pi}$, which concludes the proof. $\quad\square$

## C  Omitted Details in Section 4

**Theorem 1.** There is a constant $\epsilon > 0$ for which computing an $\epsilon$-approximate Markov perfect stationary CCE in infinite-horizon $\frac{1}{2}$-discounted zero-sum NMGs, whose underlying network structure contains either a triangle or a 3-path subgraph, is PPAD-hard. Moreover, given the PCP for PPAD conjecture [43], there is a constant $\epsilon > 0$ such that computing even an $\epsilon$-approximate Markov non-perfect stationary CCE in such zero-sum NMGs is PPAD-hard.

*Proof.* We separate the proof for the two cases as follows.

**Case 1.** $\mathcal{E}_Q$ **contains a triangle subgraph.** We will show that for *any* general-sum two-player *turn-based* MG **(A)**, the problem of computing its Markov stationary CCE, which is inherently a PPAD-hard problem [30], can be reduced to computing the Markov stationary CCE of a three-player zero-sum MG with a triangle structure networked separable interactions **(B)**. Consider an MG **(A)** with two players, players 1 and 2, and a reward function $r_1(s, a_1, a_2)$ and $r_2(s, a_2, a_1)$, where $a_i$ is the action of the $i$-th player and $r_i$ is the reward function of the $i$-th player. The transition dynamics is given by $\mathbb{P}(s' \mid s, a_1, a_2)$. In even rounds, player 2's action space is limited to Noop2, and in odd rounds, player 1's action space is limited to Noop1, where Noop is an abbreviation of "no-operation", i.e., the player does not affect the transition dynamics nor reward functions in that round. We denote player 1's action space in even rounds as $\mathcal{A}_{1,\text{even}}$ and player 2's action space in odd rounds as $\mathcal{A}_{2,\text{odd}}$.

Now, we construct a three-player zero-sum NMG. We set the reward function as $\widetilde{r}_i(s, \boldsymbol{a}) = \sum_{j \neq i} \widetilde{r}_{i,j}(s, a_i, a_j)$ and $\widetilde{r}_{i,j}(s, a_i, a_j) = -\widetilde{r}_{j,i}(s, a_j, a_i)$, where the reward functions are designed so that $\widetilde{r}_{i,j} = -\widetilde{r}_{j,i}$ for all $i, j$, $\widetilde{r}_{1,2} + \widetilde{r}_{1,3} = r_1$, and $\widetilde{r}_{2,1} + \widetilde{r}_{2,3} = r_2$, by introducing a dummy player, player 3. Here $r_1, r_2$ are the reward functions in game **(A)**. In even rounds, player 2's action space is limited to Noop2, and in odd rounds, player 1's action space is limited to Noop1. Player 3's action space is always limited to Noop3 in all rounds. The transition dynamics is defined as $\widetilde{\mathbb{P}}(s' \mid s, a_1, a_2, a_3) = \mathbb{P}(s' \mid s, a_1, a_2)$, since $a_3$ is always Noop3. In other words, player 3's action does not affect the rewards of the other two players, nor the transition dynamics, and players 1 and 2 will receive the reward as in the two-player turn-based MG. Also, note that due to the turn-based structure of the game **(A)**, the transition dynamics satisfy the decomposable condition in our Proposition 1, and it is thus a zero-sum NMG. In fact, every turn-based dynamics can be represented as an ensemble of single-controller dynamics, as we have discussed in Section 3. We set the reward function values as follows:

$$\widetilde{r}_{1,3}(s, a_1, \text{Noop3}) = -\widetilde{r}_{3,1}(s, \text{Noop3}, a_1) = r_1(s, a_1, \text{Noop2}) + r_2(s, \text{Noop2}, a_1)$$
$$\widetilde{r}_{1,3}(s, \text{Noop1}, \text{Noop3}) = -\widetilde{r}_{3,1}(s, \text{Noop3}, \text{Noop1}) = 0$$
$$\widetilde{r}_{2,3}(s, a_2, \text{Noop3}) = -\widetilde{r}_{3,2}(s, \text{Noop3}, a_2) = r_1(s, \text{Noop1}, a_2) + r_2(s, a_2, \text{Noop1})$$
$$\widetilde{r}_{2,3}(s, \text{Noop2}, \text{Noop3}) = -\widetilde{r}_{3,2}(s, \text{Noop3}, \text{Noop2}) = 0$$
$$\widetilde{r}_{1,2}(s, a_1, \text{Noop2}) = -\widetilde{r}_{2,1}(s, \text{Noop2}, a_1) = -r_2(s, \text{Noop2}, a_1)$$
$$\widetilde{r}_{1,2}(s, \text{Noop1}, a_2) = -\widetilde{r}_{2,1}(s, a_2, \text{Noop1}) = r_1(s, \text{Noop1}, a_2).$$

Note that the new game **(B)** is still a turn-based game, and thus the Markov stationary CCE is the same as the Markov stationary NE. Also, note that by construction, we know that the equilibrium policies of players 1 and 2 at the Markov stationary CCE of the game **(B)** constitute a Markov stationary CCE of the game **(A)**. If the underlying network is more general and contains a triangle subgraph, we can specify the reward and transition dynamics of these three players as above, and specify all other players to be dummy players, whose reward functions are all zero, and do not affect the reward functions of these three players, nor the transition dynamics.

**Case 2.** $\mathcal{E}_Q$ **contains a 3-path subgraph.** We will show that for *any* general-sum two-player *turn-based* MG **(A)**, the problem of computing its Markov stationary CCE can also be reduced to computing the Markov stationary CCE of a four-player zero-sum MG with 3-path networked separable interactions **(B)**. Consider an MG **(A)** with two players, players 1 and 2, and a reward function $r_1(s, a_1, a_2)$ and $r_2(s, a_2, a_1)$, where $a_i$ is the action of the $i$-th player and $r_i$ is the reward function of the $i$-th player. The transition dynamics is given by $\mathbb{P}(s' \mid s, a_1, a_2)$. In even rounds, player 2's action space is limited to Noop2, and in odd rounds, player 1's action space is limited to Noop1, where Noop is an abbreviation of "no-operation", i.e., the player does not affect the transition dynamics nor the reward functions in that round. We denote player 1's action space in even rounds as $\mathcal{A}_{1,\text{even}}$ and player 2's action space in odd rounds as $\mathcal{A}_{2,\text{odd}}$.

Now, we construct a four-player zero-sum NMG with a 3-path network structure. We set the reward function as $\widetilde{r}_1(s, \boldsymbol{a}) = \widetilde{r}_{1,2}(s, a_1, a_2) + \widetilde{r}_{1,3}(s, a_1, a_3)$ and $\widetilde{r}_2(s, \boldsymbol{a}) = \widetilde{r}_{2,1}(s, a_2, a_1) + \widetilde{r}_{2,4}(s, a_2, a_4)$ and $\widetilde{r}_{i,j}(s, a_i, a_j) = -\widetilde{r}_{j,i}(s, a_j, a_i)$. The reward functions are designed so that $\widetilde{r}_{1,2} + \widetilde{r}_{1,3} = r_1$, and $\widetilde{r}_{2,1} + \widetilde{r}_{2,4} = r_2$, where $r_1, r_2$ are the reward functions in game **(A)**, by introducing dummy players, player 3 and player 4. In even rounds, player 2's action space is limited to Noop2, and in odd rounds, player 1's action space is limited to Noop1. Player 3's action space is always limited to Noop3 in all rounds. Player 4's action space is always limited to Noop4 in all rounds. The transition dynamics is defined as $\widetilde{\mathbb{P}}(s' \mid s, a_1, a_2, a_3, a_4) = \mathbb{P}(s' \mid s, a_1, a_2)$, since $a_3$ is always Noop3 and $a_4$

is always Noop4. In other words, player 3 and player 4's actions do not affect the rewards of the other two players, nor the transition dynamics, and players 1 and 2 will receive the reward as in the two-player turn-based MG. Also, note that due to the turn-based structure of the game **(A)**, the transition dynamics satisfy the decomposable condition in our Proposition 1, and it is thus a zero-sum NMG. In fact, every turn-based dynamics can be represented as an ensemble of single-controller dynamics, as we have discussed in Section 3. We set the reward function values as follows:

$$\widetilde{r}_{1,3}(s, a_1, \mathsf{Noop3}) = -\widetilde{r}_{3,1}(s, \mathsf{Noop3}, a_1) = r_1(s, a_1, \mathsf{Noop2}) + r_2(s, \mathsf{Noop2}, a_1)$$

$$\widetilde{r}_{1,3}(s, \mathsf{Noop1}, \mathsf{Noop3}) = -\widetilde{r}_{3,1}(s, \mathsf{Noop3}, \mathsf{Noop1}) = 0$$

$$\widetilde{r}_{2,4}(s, a_2, \mathsf{Noop4}) = -\widetilde{r}_{4,2}(s, \mathsf{Noop4}, a_2) = r_1(s, \mathsf{Noop1}, a_2) + r_2(s, a_2, \mathsf{Noop1})$$

$$\widetilde{r}_{2,4}(s, \mathsf{Noop2}, \mathsf{Noop4}) = -\widetilde{r}_{4,2}(s, \mathsf{Noop4}, \mathsf{Noop2}) = 0$$

$$\widetilde{r}_{1,2}(s, a_1, \mathsf{Noop2}) = -\widetilde{r}_{2,1}(s, \mathsf{Noop2}, a_1) = -r_2(s, \mathsf{Noop2}, a_1)$$

$$\widetilde{r}_{1,2}(s, \mathsf{Noop1}, a_2) = -\widetilde{r}_{2,1}(s, a_2, \mathsf{Noop1}) = r_1(s, \mathsf{Noop1}, a_2).$$

Note that the new game **(B)** is still a turn-based game, and thus the Markov stationary CCE is the same as the Markov stationary NE. Also, note that by construction, we know that the equilibrium policies of players 1 and 2 at the Markov stationary CCE of the game **(B)** constitute a Markov stationary CCE of the game **(A)**. If the underlying network is more general and contains a 3-path subgraph, we can specify the reward and transition dynamics of these four players in the subgraph as above, and specify all other players to be dummy players, whose reward functions are all zero, and do not affect the reward functions of these three players, nor the transition dynamics. This completes the proof. □

**Proposition 7.** A connected graph that does not contain a subgraph of a triangle or a 3-path must be a star-shaped graph.

*Proof.* If the diameter of a connected graph is exactly 1, then there are only two nodes, which form a star-shaped network. If the diameter of a connected graph is greater than 2, it contradicts the non-existence of a 3-path subgraph. If the diameter of a connected graph is exactly 2, we denote the middle node as $c$, and the leftmost and rightmost nodes as $l$ and $r$. If either $l$ or $r$ has another neighbor other than $c$, it implies the existence of a 3-path subgraph, which contradicts the assumption. Therefore, the additional nodes other than $l, c, r$, if exist, have to be connected to $c$. If two neighbors of $c$ are directly connected, then it contradicts the non-existence of a triangle subgraph. Hence, all nodes except $c$ have to be connected to $c$ while not being connected to each other, which leads to a star-shaped graph. □

# D   Omitted Details in Section 5

We refer to Section E for the existing relevant result regarding stochastic approximation. The proof structure for Section D follows three steps: (1) find the continuous-time dynamics of the fictitious-play learning dynamics, (2) identify a Lyapunov function for the continuous-time version of the fictitious play ($V(\pi)$ or $L(\pi)$), and (3) since the discrete version can be viewed as a perturbed version of the continuous-time dynamics (Theorem 5), the limit point of fictitious play is contained in the level set of a Lyapunov function (Theorem 6). Theorem 5 and Theorem 6 are stated in Section E, and these theorems are restatements of [44]. In this section, with a slight abuse of notation, we interchangeably use $a_i$ to refer to either an action in $\mathcal{A}_i$, or a pure strategy $\pi_i \in \Delta(\mathcal{A}_i)$, where $\pi_i(a_i) = 1$ and $\pi_i(a_i') = 0$ for all $a_i' \neq a_i$.

## D.1   Matrix game case

### D.1.1   Fictitious-play in zero-sum NGs

We first introduce the fictitious-play dynamics for zero-sum NGs with $\mathcal{G} = (\mathcal{N}, \mathcal{E})$, i.e., zero-sum polymatrix games [23, 24], the very same one as in [5, 6]: at iteration, $k$, each player $i$ maintains a belief of the opponents' policies, $(\widehat{\pi}_{-i}^{(k)})$; she then takes action by best responding to the belief, and then updates the belief as:

$$\text{Take action: } a_i^{(k)} \in \operatorname*{argmax}_{a_i \in \mathcal{A}_i} \ r_i(e_{a_i}, \widehat{\pi}_{-i}^{(k)}), \quad \text{Update belief: } \widehat{\pi}_{-i}^{(k+1)} = \widehat{\pi}_{-i}^{(k)} + \alpha^{(k)}(e_{a_{-i}^{(k)}} - \widehat{\pi}_{-i}^{(k)})$$

where $r_i(\pi)$ is the expected payoff under joint policy $\pi$ (see Equation (1)), and $\alpha^{(k)} \geq 0$ is the stepsize. The overall procedure is summarized in Algorithm 2.

---

**Algorithm 2** Fictitious Play in zero-sum NGs ($i$-th player)

---

Choose $\widehat{\pi}_j^{(0)}$ as a uniform distribution for all $j \in \mathcal{N}/\{i\}$
**for** each timestep $k = 0, 1, \dots$ **do**
    Take action $a_i^{(k)} \in \text{argmax}_{a_i \in \mathcal{A}_i} r_i(e_{a_i}, \widehat{\pi}_{-i}^{(k)})$
    Observe other players' action $a_{-i}^{(k)}$
    Update the policy belief as $\widehat{\pi}_{-i}^{(k+1)} = \widehat{\pi}_{-i}^{(k)} + \alpha^{(k)}(e_{a_{-i}^{(k)}} - \widehat{\pi}_{-i}^{(k)})$
**end for**

---

We provide the convergence guarantee of the FP dynamics as follows, showing that zero-sum NGs, i.e., zero-sum polymatrix games [23, 24], possess the fictitious-play property [8].

**Theorem 3.** Assuming that $\sum_{k=0}^{\infty} \alpha^{(k)} \to \infty$ and $\alpha^{(k)} \to 0$ as $k \to \infty$, then the limit points of $(\widehat{\pi}^{(k)})_{k \geq 0}$ are the NE of the zero-sum NG.

*Proof of Theorem 3.* To prove the fictitious-play property, we consider a continuous version of Algorithm 2. Assuming that $\sum_{k=0}^{\infty} \alpha^{(k)} \to \infty$ and $\alpha^{(k)} \to 0$, [44, Proposition 3.27] states that we can characterize the limit set of $(\widehat{\pi}^{(k)})_{k \geq 0}$ by considering the following dynamics:

$$\pi_i + \frac{d\pi_i}{dt} \in \underset{a_i \in \mathcal{A}_i}{\text{argmax}} \ r_i(e_{a_i}, \pi_{-i}). \tag{23}$$

We define a Lyapunov function as

$$V(\pi) = \sum_{i \in \mathcal{N}} \left( \max_{a_i \in \mathcal{A}_i} \ r_i(e_{a_i}, \pi_{-i}) - r_i(\pi) \right). \tag{24}$$

**Claim 5.** $V(\pi(t))$ is a Lyapunov function for (23).

*Proof.* Let $\text{argmax}_{a_i \in \mathcal{A}_i} r_i(e_{a_i}, \pi_{-i})$ in the formula be $a_i^{\star}$, then we have

$$\frac{dV(\pi(t))}{dt} = \sum_{i \in \mathcal{N}} \left( \sum_{j \in \mathcal{E}_i} e_{a_i^{\star}}^{\intercal} r_{i,j} \pi_j' \right) = \sum_{i \in \mathcal{N}} \left( \sum_{j \in \mathcal{E}_i} e_{a_i^{\star}}^{\intercal} r_{i,j} (e_{a_j^{\star}} - \pi_j) \right)$$

$$= \sum_{i \in \mathcal{N}} \left( \sum_{j \in \mathcal{E}_i} -e_{a_i^{\star}}^{\intercal} r_{i,j} \pi_j \right) = -V(\pi(t))$$

where we use $\pi_j'$ to denote $\frac{d\pi_j}{dt}$, and the first equality is derived from the envelope theorem. Since $\max_{a_i} r_i(a_i, \pi_{-i}) \geq r_i(\pi)$, $V$ is guaranteed to be non-negative. We can thus express $V(t) = V(0)e^{-t}$, indicating that it is decreasing with a linear rate in continuous time. $\square$

Consequently, [44, Proposition 3.27] implies

$$\lim_{k \to \infty} \left( \sum_{i \in \mathcal{N}} \left( \max_{a_i \in \mathcal{A}_i} \ r_i(e_{a_i}, \widehat{\pi}_{-i}^{(k)}) - r_i(\widehat{\pi}^{(k)}) \right) \right) = 0$$

which concludes that every limit point of $(\widehat{\pi}^{(k)})_{k \geq 0}$ is an NE. $\square$

Note that the fictitious-play learning dynamics for zero-sum polymatrix games have also been proposed and analyzed in [12], and our result above is a reproduction of it.

### D.1.2 Smooth fictitious play in zero-sum NGs

We can also provide guarantees for the learning dynamics of *smooth* fictitious play (may also be referred to as *stochastic* fictitious play later) [7], with convergence to the quantal response equilibrium (QRE) of the game [111, 112].

**Definition 6.** A policy $\pi_\tau^\star = \left(\pi_{\tau,1}^\star, \cdots, \pi_{\tau,n}^\star\right)$ is a quantal response equilibrium of the game with regularization coefficient $\tau$ if the following condition holds

$$\pi_{\tau,i}^\star(a_i) = \frac{\exp\left([\boldsymbol{r}_i \pi_\tau^\star]_{a_i} / \tau\right)}{\sum_{a_i' \in \mathcal{A}_i} \exp\left([\boldsymbol{r}_i \pi_\tau^\star]_{a_i'} / \tau\right)}$$

for all $i \in \mathcal{N}$ and $a_i \in \mathcal{A}_i$ [111].

A QRE always exists in finite games. Moreover, a QRE has an equivalent notion as finding the Nash equilibrium of the game with entropy-regularized payoffs: i.e., $\pi_\tau^\star$ satisfies that

$$r_{\tau,i}\left(\pi_i', \pi_{\tau,-i}^\star\right) \leq r_{\tau,i}\left(\pi_\tau^\star\right),$$

where

$$r_{\tau,i}(\pi) := r_i(\pi) + \tau\mathcal{H}(\pi_i) - \sum_{j \in \mathcal{E}_{r,i}} \frac{\tau}{|\mathcal{E}_{r,j}|}\mathcal{H}(\pi_j) \tag{25}$$

and $\mathcal{H}(\pi_i) := -\sum_{a_i \in \mathcal{A}_i} \pi_i(a_i)\log(\pi_i(a_i))$ is the Shannon entropy function [113]. Reference [83] provided a novel analysis showing that a unique NE exists for zero-sum NGs with entropy regularization (thus the QRE for the unregularized zero-sum NG).

**Remark 7.** In most existing literature [51, 83], the entropy regularized reward is defined as $r_i(\pi) + \tau\mathcal{H}(\pi_i)$. Indeed, note that

$$\operatorname*{argmax}_{\pi_i \in \Delta(\mathcal{A}_i)} \left(r_i(\pi) + \tau\mathcal{H}(\pi_i)\right) = \operatorname*{argmax}_{\pi_i \in \Delta(\mathcal{A}_i)} \left(r_i(\pi) + \tau\mathcal{H}(\pi_i) - \sum_{j \in \mathcal{E}_{r,i}} \frac{\tau}{|\mathcal{E}_{r,j}|}\mathcal{H}(\pi_j)\right)$$

for any $i \in \mathcal{N}$, so it does not affect the equilibria. Moreover, by defining $r_{\tau,i}(\pi) := r_i(\pi) + \tau\mathcal{H}(\pi_i) - \sum_{j \in \mathcal{E}_{r,i}} \frac{\tau}{|\mathcal{E}_{r,j}|}\mathcal{H}(\pi_j)$, we can have that $\sum_{i \in \mathcal{N}} r_{\tau,i}(\pi) = 0$ holds for any joint product policy $\pi$.

---

**Algorithm 3** Stochastic fictitious play in zero-sum NGs ($i$-th player)

---

Choose $\widehat{\pi}_j^{(0)}$ as a uniform distribution for all $j \in \mathcal{N}/\{i\}$
**for** each timestep $k = 0, 1, \dots$ **do**
    Take action $a_i^{(k)} \sim \operatorname{argmax}_{\mu_i \in \Delta(\mathcal{A}_i)} r_{\tau,i}(\mu_i, \widehat{\pi}_{-i}^{(k)})$
    Observe other players' action $a_{-i}^{(k)}$
    Update the policy belief as $\widehat{\pi}_{-i}^{(k+1)} = \widehat{\pi}_{-i}^{(k)} + \alpha^{(k)}(e_{a_{-i}^{(k)}} - \widehat{\pi}_{-i}^{(k)})$
**end for**

---

In Algorithm 3, players initialize their beliefs for other players ($\widehat{\pi}_{-i}$) as a uniform distribution. They sample from the best-response policy with respect to the entropy-regularized reward, given the beliefs of other players' policies. Subsequently, each player observes other players' actions and updates her beliefs.

**Theorem 4.** Assuming that $\sum_{k=0}^\infty \alpha^{(k)} \to \infty$ and $\lim_{k\to\infty} \alpha^{(k)} \to 0$, $(\widehat{\pi}^{(k)})_{k\geq 0}$ converges to a QRE of the zero-sum NG with probability 1.

*Proof of Theorem 4.* To prove the fictitious-play property, we consider a continuous-time version of the learning dynamics in Algorithm 3. Assuming $\sum_{k=0}^\infty \alpha^{(k)} \to \infty$ and $\alpha^{(k)} \to 0$, [44, Proposition 3.27] states that we can characterize the limit set of $(\widehat{\pi}^{(k)})_{k\geq 0}$ by considering the following dynamics

$$\pi_i + \frac{d\pi_i}{dt} = \operatorname*{argmax}_{\mu_i \in \Delta(\mathcal{A}_i)} r_{\tau,i}(\mu_i, \pi_{-i}). \tag{26}$$

We define a Lyapunov function as

$$V_\tau(\pi) = \sum_{i \in \mathcal{N}} \left( \max_{\mu_i \in \Delta(\mathcal{A}_i)} r_{\tau,i}(\mu_i, \pi_{-i}) - r_{\tau,i}(\pi) \right).$$

**Claim 6.** $V_\tau(\pi(t))$ is a Lyapunov function for (26).

*Proof.* Let the maximizer of $r_{\tau,i}(\mu_i, \pi_{-i})$ in the formula be $\mu_i^\star$, which we know is unique due to the regularization. Thus, we have

$$
\begin{aligned}
\frac{dV_\tau(\pi(t))}{dt} &= \sum_{i \in \mathcal{N}} \left( \left( \sum_{j \in \mathcal{E}_i} \mu_i^{\star\mathsf{T}} r_{i,j} \pi_j' \right) - \left( \tau(\mathcal{H}(\pi_i))' \right) \right) \\
&= \sum_{i \in \mathcal{N}} \left( \left( \sum_{j \in \mathcal{E}_i} \mu_i^{\star\mathsf{T}} r_{i,j} (\mu_j^\star - \pi_j) \right) - \left( \tau(\mathcal{H}(\pi_i))' \right) \right) \\
&= \sum_{i \in \mathcal{N}} \left( \left( \sum_{j \in \mathcal{E}_i} -\mu_i^{\star\mathsf{T}} r_{i,j} \pi_j \right) + \tau(1 + \log \pi_i)^\mathsf{T} \pi_i' \right) \\
&= \sum_{i \in \mathcal{N}} \left( \left( \sum_{j \in \mathcal{E}_i} -\mu_i^{\star\mathsf{T}} r_{i,j} \pi_j \right) + \tau(1 + \log \pi_i)^\mathsf{T} (\mu_i^\star - \pi_i) \right) \\
&= \sum_{i \in \mathcal{N}} \left( \left( \sum_{j \in \mathcal{E}_i} -\mu_i^{\star\mathsf{T}} r_{i,j} \pi_j \right) + \tau(\mathcal{H}(\pi_i)) + \tau(\log \pi_i)^\mathsf{T} (\mu_i^\star) \right) \\
&\leq \sum_{i \in \mathcal{N}} \left( \left( \sum_{j \in \mathcal{E}_i} -\mu_i^{\star\mathsf{T}} r_{i,j} \pi_j \right) + \tau(\mathcal{H}(\pi_i)) - \tau(\mathcal{H}(\mu_i)) \right) = -V_\tau(\pi(t))
\end{aligned}
$$

where the first equality is derived from the envelope theorem and the last inequality is from Gibbs' inequality. Since $\max_{\mu_i \in \Delta(\mathcal{A}_i)} r_{\tau,i}(\mu_i, \pi_{-i}) \geq r_{\tau,i}(\pi)$, $V$ is guaranteed to be non-negative. Therefore, we have $0 \leq V(t) \leq V(0)e^{-t}$, indicating that it is decreasing. □

Consequently, [44, Proposition 3.27] implies that

$$\lim_{k \to \infty} \left( \sum_{i \in \mathcal{N}} \left( \max_{\mu_i \in \Delta(\mathcal{A}_i)} r_{\tau,i}(\mu_i, \widehat{\pi}_{-i}^{(k)}) - r_{\tau,i}(\widehat{\pi}^{(k)}) \right) \right) = 0$$

which concludes that every limit point is a QRE. Since the QRE is unique for zero-sum NGs, we conclude that $(\pi^{(k)})_{k \geq 0}$ converges to the QRE of the zero-sum NG. □

**Remark.** Algorithm 2 converges to an NE, and Algorithm 3 converges to a QRE. Since the QRE is unique in zero-sum NG for a fixed $\tau$, we can identify the converging point in Algorithm 3, while we cannot determine which NE is the converging point in Algorithm 2.

### D.2 Fictitious-play property of infinite-horizon zero-sum NMGs of a star-shape

Before presenting the results, we examine some properties of a star-shaped zero-sum NG (i.e., the polymatrix case). We define player 1 as the center player without loss of generality. First, we can view a star-shaped zero-sum NG as a constant-sum separable star-shaped game, as detailed below.

**Proposition 8.** There exist some $\{c_i\}_{i \in \mathcal{N}/\{1\}}$ with $c_i \in \mathbb{R}$ such that a star-shaped zero-sum NG satisfies the following identities:

$$r_{i,1}^\mathsf{T} + r_{1,i} = c_i \mathbf{1}\mathbf{1}^\mathsf{T} \qquad \text{for every } i \in \mathcal{N}/\{1\}, \qquad \sum_{i \in \mathcal{N}/\{1\}} c_i = 0.$$

---

**Algorithm 4** Fictitious play in zero-sum NMGs of a star-shape ($i$-th player)

---

Choose $\widehat{\pi}_j^{(0)}(s)$ to be a uniform distribution for all $j \in \mathcal{N}/\{i\}$ and $s \in \mathcal{S}$

Choose $\widehat{Q}_i^{(0)}(s, \boldsymbol{a})$ to be an arbitrary value for all $s \in \mathcal{S}$ and $\boldsymbol{a} \in \mathcal{A}$

Choose $N(s) = 0$ for all $s \in \mathcal{S}$

**for** each timestep $k = 0, 1, \ldots$ **do**

    Observe the current state $s^{(k)}$ and update the visitation number as $N(s^{(k)}) = N(s^{(k)}) + 1$

    Take action $a_i^{(k)} \in \mathrm{argmax}_{a_i \in \mathcal{A}_i} \widehat{Q}_i^{(k)}(s^{(k)}, e_{a_i}, \widehat{\pi}_{-i}^{(k)}(s^{(k)}))$

    Update $V_i$-belief for all $i \in \mathcal{N}$ and $s \in \mathcal{S}$ as

$$\widehat{V}_i^{(k)}(s) = \max_{a_i \in \mathcal{A}_i} \widehat{Q}_i^{(k)}(s, e_{a_i}, \widehat{\pi}_{-i}^{(k)}(s)) \tag{27}$$

    Observe other players' action $a_{-i}^{(k)}$

    Update the belief as $\widehat{\pi}_{-i}^{(k+1)}(s) = \widehat{\pi}_{-i}^{(k)}(s) + \mathbf{1}(s = s^{(k)})\alpha^{N(s)}(e_{a_{-i}^{(k)}} - \widehat{\pi}_{-i}^{(k)}(s))$ for all $s \in \mathcal{S}$

    **if** player $i = 1$ **then**

        Update the $Q_{1,j}$-belief for all $j \in \mathcal{N}/\{1\}$, $s \in \mathcal{S}$, and $\boldsymbol{a} \in \mathcal{A}$ as

$$\widehat{Q}_{1,j}^{(k+1)}(s, a_1, a_j) = \widehat{Q}_{1,j}^{(k)}(s, a_1, a_j)$$
$$+ \mathbf{1}(s = s^{(k)})\beta^{N(s)}\left(r_{1,j}(s, a_1, a_j) + \gamma \sum_{s' \in \mathcal{S}} \frac{1}{n-1}\mathbb{P}_1(s' \mid s, a_1)\widehat{V}_1^{(k)}(s') - \widehat{Q}_{1,j}^{(k)}(s, a_1, a_j)\right)$$

        Update the $Q_1$-belief for all $s \in \mathcal{S}$ and $\boldsymbol{a} \in \mathcal{A}$ as

$$\widehat{Q}_1^{(k+1)}(s, \boldsymbol{a}) = \sum_{j \in \mathcal{N}/\{1\}} \widehat{Q}_{1,j}^{(k+1)}(s, a_1, a_j)$$

    **else**

        Update the $Q_i$-belief for all $s \in \mathcal{S}$ and $\boldsymbol{a} \in \mathcal{A}$ as

$$\widehat{Q}_i^{(k+1)}(s, \boldsymbol{a}) = \widehat{Q}_{i,1}^{(k+1)}(s, a_i, a_1) = \widehat{Q}_{i,1}^{(k)}(s, a_i, a_1) + \mathbf{1}(s = s^{(k)})\beta^{N(s)}\Big(r_{i,1}(s, a_i, a_1)$$
$$+ \gamma \sum_{s' \in \mathcal{S}} \mathbb{P}_1(s' \mid s, a_1)\widehat{V}_i^{(k)}(s') - \widehat{Q}_{i,1}^{(k)}(s, a_i, a_1)\Big)$$

    **end if**

    State transitions $s^{(k+1)} \sim \mathbb{P}_1(\cdot \mid s^{(k)}, a_1^{(k)})$

**end for**

---

*Proof.* By the definition of a zero-sum NG, for arbitrary $\pi_1 \in \Delta(\mathcal{A}_1), \{\pi_i\}_{i \in \mathcal{N}/\{1\}} \in \prod_{i \in \mathcal{N}/\{1\}} \Delta(\mathcal{A}_i)$, the following holds:

$$\sum_{i \in \mathcal{N}/\{1\}} \left(\pi_1^\mathsf{T} r_{1,i} + \pi_1^\mathsf{T} r_{i,1}^\mathsf{T}\right)\pi_i = 0 \tag{28}$$

which implies $\pi_1^\mathsf{T}(r_{1,i} + r_{i,1}^\mathsf{T}) = c_i \mathbf{1}^\mathsf{T}$ for some constant $c_i$, since Equation (28) holds for any $\pi_i \in \Delta(\mathcal{A}_i)$. To be specific, $\pi_1^\mathsf{T}(r_{1,i} + r_{i,1}^\mathsf{T})\pi_i$ should be the same when we plugging $\pi_i = e_{a_i}$ for any $a_i \in \mathcal{A}_i$, so that every element of $\pi_1^\mathsf{T}(r_{1,i} + r_{i,1}^\mathsf{T})$ is the same, i.e., there exists some $c_i \in \mathbb{R}$ such that $\pi_1^\mathsf{T}(r_{1,i} + r_{i,1}^\mathsf{T}) = c_i \mathbf{1}^\mathsf{T}$. Plugging to Equation (28), we have $\sum_{i \in \mathcal{N}/\{1\}} c_i = 0$. Moreover, this again implies $r_{1,i} + r_{i,1}^\mathsf{T} = c_i \mathbf{1}\mathbf{1}^\mathsf{T}$, since $\pi_1^\mathsf{T}(r_{1,i} + r_{i,1}^\mathsf{T}) = c_i \mathbf{1}^\mathsf{T}$ always holds for any $\pi_1 \in \Delta(\mathcal{A}_1)$ by a similar argument as above. $\qquad\square$

Second, we define the Nash equilibrium value for the center player in a star-shaped zero-sum NG, which is different from the general zero-sum NG case, where there may not exist a unique Nash value [24].

**Proposition 9.** There exists a unique Nash equilibrium value for the center player 1 in a star-shaped zero-sum NG (i.e., $r_{i,j} = 0$ if $i \neq 1$ and $j \neq 1$).

*Proof.* Player 1 aims to maximize $\sum_{i \in \mathcal{N}/\{1\}} \pi_1^\mathsf{T} r_{1,i} \pi_i$ while player $i \neq 1$ aims to maximize $\pi_1^\mathsf{T} r_{i,1}^\mathsf{T} \pi_i = \pi_1^\mathsf{T}(c_i \mathbf{1}\mathbf{1}^\mathsf{T} - r_{1,i})\pi_i = c_i - \pi_1^\mathsf{T} r_{1,i} \pi_i$, with $c_i$ given in Proposition 8. We can solve these problems simultaneously by the following maxmin problem:

$$\underset{\pi_1 \in \Delta(\mathcal{A}_1)}{\text{maximize}} \quad \underset{(\pi_i)_{i \in \mathcal{N}/\{1\}} \in \prod_{i \in \mathcal{N}/\{1\}} \Delta(\mathcal{A}_i)}{\text{minimize}} \sum_{i \in \mathcal{N}/\{1\}} \pi_1^\mathsf{T} r_{1,i} \pi_i.$$

Since $\Delta(\mathcal{A}_1)$ and $\prod_{i \in \mathcal{N}/\{1\}} \Delta(\mathcal{A}_i)$ are compact and convex sets, we can use the minimax theorem to show that $\text{maximize}_{\pi_1 \in \Delta(\mathcal{A}_1)} \text{minimize}_{(\pi_i)_{i \in \mathcal{N}/\{1\}} \in \prod_{i \in \mathcal{N}/\{1\}} \Delta(\mathcal{A}_i)} \sum_{i \in \mathcal{N}/\{1\}} \pi_1^\mathsf{T} r_{1,i} \pi_i$ is unique. $\square$

Note that there can be multiple Nash equilibrium values for each non-center player, but the *sum* of Nash equilibrium values for non-center players is always unique (see the proof of Proposition 9).

Now, we start to prove that fictitious play dynamics given in Algorithm 4 converges to a Markov stationary NE in infinite-horizon zero-sum NMGs with a star-shaped network.

**Theorem 2.** Suppose Assumption 1 holds and Algorithm 4 visits every state infinitely often with probability 1. Then, for a star-shaped multi-player zero-sum NMG, the belief $(\widehat{\pi}^{(k)})_{k \geq 0}$ converges to a Markov stationary NE and the belief $(\widehat{Q}^{(k)})_{k \geq 0}$ converges to the corresponding NE value of the zero-sum NMG with probability 1, as $k \to \infty$.

*Proof.* To prove the result, we consider a continuous-time version of Algorithm 4. Using standard two-timescale stochastic approximation techniques [44, Proposition 3.27], we can show that the limit set of our FP dynamics can be captured by that of a continuous-time differential inclusion. Before, we define several notation:

$$\boldsymbol{Q}_i(s) := (Q_{i,1}(s), \ldots, Q_{i,i-1}(s), \mathbf{0}, Q_{i,i+1}(s) \ldots, Q_{i,n}(s)) \in \mathbb{R}^{|\mathcal{A}_i| \times \sum_{i \in \mathcal{N}} |\mathcal{A}_i|}$$

$$\boldsymbol{Q}(s) := ((\boldsymbol{Q}_1(s))^\mathsf{T}, (\boldsymbol{Q}_2(s))^\mathsf{T}, \ldots, (\boldsymbol{Q}_n(s))^\mathsf{T})^\mathsf{T} \in \mathbb{R}^{\sum_{i \in \mathcal{N}} |\mathcal{A}_i| \times \sum_{i \in \mathcal{N}} |\mathcal{A}_i|}$$

$$h(\boldsymbol{Q}(s)) := \max_{\mu \in \prod_{i \in \mathcal{N}} \Delta(\mathcal{A}_i)} \left| \left( \sum_{i \in \mathcal{N}/\{1\}} \mu_1^\mathsf{T} Q_{1,i}(s) \mu_i + \sum_{i \in \mathcal{N}/\{1\}} \mu_i^\mathsf{T} Q_{i,1}(s) \mu_1 \right) \right|. \qquad (29)$$

Then, we consider the following differential inclusion for each $s \in \mathcal{S}$:

$$\pi_1(s) + \frac{d\pi_1(s)}{dt} \in \underset{a_1 \in \mathcal{A}_1}{\text{argmax}} \left( \sum_{i \in \mathcal{N}/\{1\}} e_{a_1}^\mathsf{T} Q_{1,i}(s) \pi_i(s) \right),$$

$$\pi_i(s) + \frac{d\pi_i(s)}{dt} \in \underset{a_i \in \mathcal{A}_i}{\text{argmax}} \left( e_{a_i}^\mathsf{T} Q_{i,1}(s) \pi_1(s) \right), \qquad \frac{dQ_{1,i}(s)}{dt} = \frac{dQ_{i,1}(s)}{dt} = \mathbf{0}, \qquad (30)$$

with a Lyapunov function candidate being

$$L_\lambda(\pi, \boldsymbol{Q}, s)$$
$$= \left( \max_{a_1 \in \mathcal{A}_1} \left( \sum_{i \in \mathcal{N}/\{1\}} e_{a_1}^\mathsf{T} Q_{1,i}(s) \pi_i(s) \right) + \sum_{i \in \mathcal{N}/\{1\}} \max_{a_i \in \mathcal{A}_i} \left( e_{a_i}^\mathsf{T} Q_{i,1}(s) \pi_1(s) \right) - \lambda h(\boldsymbol{Q}(s)) \right)_+$$

where $h$ is defined before, and $\lambda$ is chosen as $1 < \lambda < 1/\gamma$. Then, Claim 7 below proves that $L_\lambda(\pi, \boldsymbol{Q}, s)$ is a Lyapunov function for (30).

**Claim 7.** For every $1 < \lambda < 1/\gamma$, $L_\lambda(\pi, \boldsymbol{Q}, s)$ is a Lyapunov function of (30) for the set $\Lambda = \{(\pi, \boldsymbol{Q}) : L_\lambda(\pi, \boldsymbol{Q}, s) = 0\}$.

*Proof.* First, we define $V_\lambda(\pi, \boldsymbol{Q}, s)$ as below:

$$V_\lambda(\pi, \boldsymbol{Q}, s) = \max_{a_1 \in \mathcal{A}_1} \left( \sum_{i \in \mathcal{N}/\{1\}} e_{a_1}^\mathsf{T} Q_{1,i}(s) \pi_i(s) \right) + \sum_{i \in \mathcal{N}/\{1\}} \max_{a_i \in \mathcal{A}_i} \left( e_{a_i}^\mathsf{T} Q_{i,1}(s) \pi_1(s) \right) - \lambda h(\boldsymbol{Q}(s)).$$

Then, we have $L_\lambda(\pi, \boldsymbol{Q}, s) = (V_\lambda(\pi, \boldsymbol{Q}, s))_+$. Moreover, let the maximizer of $\sum_{i \in \mathcal{N}/\{1\}} e_{a_1}^\mathsf{T} Q_{1,i}(s)\pi_i(s)$ be $a_1^\star$ and let the maximizer of $e_{a_i}^\mathsf{T} Q_{i,1}(s)\pi_1(s)$ as $a_i^\star$ for $i \in \mathcal{N}/\{1\}$. Then, we have

$$
\begin{aligned}
\frac{dV_\lambda(\pi, \boldsymbol{Q}, s)}{dt} &= \left( \sum_{i \in \mathcal{N}/\{1\}} e_{a_1^\star}^\mathsf{T} Q_{1,i}(s)\pi_i(s)' \right) + \sum_{i \in \mathcal{N}/\{1\}} \left( e_{a_i^\star}^\mathsf{T} Q_{i,1}(s)\pi_1'(s) \right) \\
&= \left( \sum_{i \in \mathcal{N}/\{1\}} e_{a_1^\star}^\mathsf{T} Q_{1,i}(s)(e_{a_i^\star} - \pi_i(s)) \right) + \sum_{i \in \mathcal{N}/\{1\}} \left( e_{a_i^\star}^\mathsf{T} Q_{i,1}(s)(e_{a_1^\star} - \pi_1(s)) \right) \\
&< -V_\lambda(\pi, \boldsymbol{Q}, s)
\end{aligned}
$$

since $\sum_{i \in \mathcal{N}/\{1\}} e_{a_1^\star}^\mathsf{T} Q_{1,i}(s)e_{a_i^\star} + \sum_{i \in \mathcal{N}/\{1\}} e_{a_i^\star}^\mathsf{T} Q_{i,1}(s)e_{a_1^\star} \le h(\boldsymbol{Q}(s)) < \lambda h(\boldsymbol{Q}(s))$ holds by the definition of $h(\boldsymbol{Q}(s))$. Therefore, $V_\lambda(\pi, \boldsymbol{Q}, s)$ is strictly decreasing with respect to time when $V_\lambda(\pi, \boldsymbol{Q}, s) \ge 0$. To emphasize the time dependence of $V_\lambda$ and $L_\lambda$, we will write $V_\lambda(\pi, \boldsymbol{Q}, s, t)$ and $L_\lambda(\pi, \boldsymbol{Q}, s, t)$.

If $V_\lambda(\pi, \boldsymbol{Q}, s, t) \ge 0$, then $L_\lambda(\pi, \boldsymbol{Q}, s, t) = V_\lambda(\pi, \boldsymbol{Q}, s, t)$ is strictly decreasing if $L_\lambda(\pi, \boldsymbol{Q}, s, t) > 0$. Therefore, we can see that if $L_\lambda(\pi, \boldsymbol{Q}, s, t) > 0$ so that $V_\lambda(\pi, \boldsymbol{Q}, s, t) > 0$, then $L_\lambda(\pi, \boldsymbol{Q}, s, t') < L_\lambda(\pi, \boldsymbol{Q}, s, t)$ for all $t' > t$, i.e., $L_\lambda(\pi, \boldsymbol{Q}, s, t)$ keeps strictly decreasing in this case.

If $V_\lambda(\pi, \boldsymbol{Q}, s, t) < 0$, then $L_\lambda(\pi, \boldsymbol{Q}, s, t) = 0$ always holds. Assume that there exists $t_1 < t_2$ such that $V_\lambda(\pi, \boldsymbol{Q}, s, t_1) < 0$ and $V_\lambda(\pi, \boldsymbol{Q}, s, t_2) > 0$. Due to the continuity of $V_\lambda$, there exists some $t \in (t_1, t_2)$ such that $V_\lambda(\pi, \boldsymbol{Q}, s, t) = 0$. Then, $\frac{dV_\lambda(\pi, \boldsymbol{Q}, s, t)}{dt} < -V_\lambda(\pi, \boldsymbol{Q}, s, t) = 0$, so it is strictly negative, which prevents it from becoming a positive value, so it is a contradiction. Therefore, if $L_\lambda(\pi, \boldsymbol{Q}, s, t) = 0$, then $L_\lambda(\pi, \boldsymbol{Q}, s, t') = 0$ for all $t' > t$ in this case. $\qquad\square$

Therefore, [44, Proposition 3.27] implies

$$
\begin{aligned}
\lim_{k \to \infty} \Bigg( \max_{a_1 \in \mathcal{A}_1} &\left( \sum_{i \in \mathcal{N}/\{1\}} e_{a_1}^\mathsf{T} \widehat{Q}_{1,i}^{(k)}(s)\widehat{\pi}_i^{(k)}(s) \right) \\
&+ \sum_{i \in \mathcal{N}/\{1\}} \max_{a_i \in \mathcal{A}_i} \left( e_{a_i}^\mathsf{T} \widehat{Q}_{i,1}^{(k)}(s)\widehat{\pi}_1^{(k)}(s) \right) - \lambda h(\widehat{Q}^{(k)}(s)) \Bigg)_+ = 0
\end{aligned}
\tag{31}
$$

for every $s \in \mathcal{S}$.

In Claim 8, we will prove that an NG with $(\mathcal{G} = (\mathcal{N}, \mathcal{E}_Q), \mathcal{A} = (\mathcal{A}_i)_{i \in \mathcal{N}}, (\widehat{Q}^{(k)}(s))_{(i,j) \in \mathcal{E}_Q})$ asymptotically becomes a zero-sum NG as $k \to \infty$ for all $s \in \mathcal{S}$. Indeed, we have

$$
h(\widehat{\boldsymbol{Q}}^{(k)}(s)) = \max_{\mu \in \prod_{i \in \mathcal{N}} \Delta(\mathcal{A}_i)} \left| \left( \sum_{i \in \mathcal{N}/\{1\}} \mu_1^\mathsf{T} \widehat{\boldsymbol{Q}}_{1,i}^{(k)}(s)\mu_i + \sum_{i \in \mathcal{N}/\{1\}} \mu_i^\mathsf{T} \widehat{\boldsymbol{Q}}_{i,1}^{(k)}(s)\mu_1 \right) \right|,
$$

we can conclude that for arbitrary policy $\mu \in \prod_{i \in \mathcal{N}} \Delta(\mathcal{A}_i)$, the sum of payoffs in NG with $(\mathcal{G} = (\mathcal{N}, \mathcal{E}_Q), \mathcal{A} = (\mathcal{A}_i)_{i \in \mathcal{N}}, (\widehat{Q}^{(k)}(s))_{(i,j) \in \mathcal{E}_Q})$ goes to 0 as $k \to \infty$. Therefore, $h(\widehat{\boldsymbol{Q}}^{(k)}(s)) \to 0$ implies that the NG is zero-sum NG, where $h$ is defined as Equation (29).

Before stating that $\widehat{Q}^{(k)}$ asymptotically becomes zero-sum NGs, we state a lemma from [16].

**Lemma 1** ([16]). Suppose the sequence of random variables $(y_k)_{k \ge 0}$ with $y_k \in \mathbb{R}^d$ satisfies

$$
\begin{aligned}
y_{k+1}[n] &\le (1 - \beta_{n,k})\, y_k[n] + \beta_{n,k}\, (\gamma \|y_k\|_\infty + \bar{\epsilon}_k + \omega_{n,k}) \\
y_{k+1}[n] &\ge (1 - \beta_{n,k})\, y_k[n] + \beta_{n,k}\, (-\gamma \|y_k\|_\infty + \underline{\epsilon}_k + \omega_{n,k})
\end{aligned}
$$

for all $k \ge 0$, where $y_k[n]$ denotes the $n$-th element in $y_k$, $\gamma \in (0, 1)$, $\sum_{k=0}^\infty \beta_{n,k} = \infty$, $\lim_{k \to \infty} \beta_{n,k} = 0$ for each $n$ with probability 1, the error sequence $(\epsilon_k)_{k \ge 0}$ satisfies $\limsup_{k \to \infty} |\bar{\epsilon}_k| \le c$ and $\limsup_{k \to \infty} |\underline{\epsilon}_k| \le c$ for some $c \ge 0$ with probability 1. Here, $\omega_{n,k}$ is a stochastic approximation term that is zero-mean and has finite variance conditioned on the history. Suppose that $\|y_k\|_\infty$ is bounded for all $k$. Then, we have $\limsup_{k \to \infty} \|y_k\|_\infty \le \frac{c}{1-\gamma}$ with probability 1, provided that either $\omega_{n,k} = 0$ for all $n, k$ or $\sum_{k=0}^\infty \beta_{n,k}^2 < \infty$ for each $n$ with probability 1.

**Claim 8.** $h(\widehat{\boldsymbol{Q}}^{(k)}(s))$ converges to 0 for all $s \in \mathcal{S}$. In other words, an NG with $(\mathcal{G} = (\mathcal{N}, \mathcal{E}_Q), \mathcal{A} = (\mathcal{A}_i)_{i \in \mathcal{N}}, (\widehat{Q}^{(k)}(s))_{(i,j) \in \mathcal{E}_Q})$ asymptotically becomes a zero-sum NG as $k \to \infty$ for all $s \in \mathcal{S}$.

*Proof.* Rewriting Equation (31) with the belief of the value function, we have that for all $s \in \mathcal{S}$

$$\lim_{k \to \infty} \left( \sum_{i \in \mathcal{N}} \widehat{V}_i^{(k)}(s) - \lambda h(\widehat{\boldsymbol{Q}}^{(k)}(s)) \right)_+ = 0. \tag{32}$$

By the definition of $\widehat{V}_i^{(k)}(s)$ and Equation (32), we have

$$-\lambda h(\widehat{\boldsymbol{Q}}^{(k)}(s)) \leq -h(\widehat{\boldsymbol{Q}}^{(k)}(s)) \leq \sum_{i \in \mathcal{N}} \widehat{V}_i^{(k)}(s) \leq \lambda h(\widehat{\boldsymbol{Q}}^{(k)}(s)) + \bar{\epsilon}_k(s)$$

for all $s \in \mathcal{S}$ and $k \geq 0$ for some $(\bar{\epsilon}_k(s))_{k \geq 0}$, where $\bar{\epsilon}_k(s) \to 0$. Moreover, summing over all the $Q$-belief estimates over $i$, we have

$$\sum_{i \in \mathcal{N}} \widehat{Q}_i^{(k+1)}(s, \boldsymbol{a}) = (1 - \bar{\beta}_k(s)) \sum_{i \in \mathcal{N}} \widehat{Q}_i^{(k)}(s, \boldsymbol{a}) + \gamma \bar{\beta}_k(s) \left( \sum_{s' \in S} \mathbb{P}(s' \mid s, \boldsymbol{a}) \sum_{i \in \mathcal{N}} \widehat{V}_i^{(k)}(s') \right)$$

where $\bar{\beta}_k(s) := \mathbf{1}(s = s^{(k)})\beta^{(N(s))}$. Thus, we have

$$\sum_{i \in \mathcal{N}} \widehat{Q}_i^{(k+1)}(s, \boldsymbol{a}) \leq (1 - \bar{\beta}_k(s)) \sum_{i \in \mathcal{N}} \widehat{Q}_i^{(k)}(s, \boldsymbol{a}) + \bar{\beta}_k(s) \left( \bar{\gamma} \max_{s' \in \mathcal{S}} h(\widehat{\boldsymbol{Q}}^{(k)}(s')) + \gamma \bar{\epsilon}^{(k)} \right)$$

$$\sum_{i \in \mathcal{N}} \widehat{Q}_i^{(k+1)}(s, \boldsymbol{a}) \geq (1 - \bar{\beta}_k(s)) \sum_{i \in \mathcal{N}} \widehat{Q}_i^{(k)}(s, \boldsymbol{a}) - \bar{\beta}_k(s) \left( \bar{\gamma} \max_{s' \in \mathcal{S}} h(\widehat{\boldsymbol{Q}}^{(k)}(s')) \right)$$

where $\bar{\gamma} = \gamma\lambda \in (0, 1)$. Since $\max_{s' \in \mathcal{S}} h(\widehat{\boldsymbol{Q}}^{(k)}(s'))$ is the maximal value of $\left| \sum_{i \in \mathcal{N}} \widehat{Q}_i^{(k)}(s, \boldsymbol{a}) \right|$, we can apply Lemma 1 to this situation. Let $\mathcal{Z} := \mathcal{S} \times \mathcal{A}$ be the set of all possible state-action pairs. Then, we can view this problem as $y_k[z] := \sum_{i \in \mathcal{N}} \widehat{Q}_i^{(k)}(s, \boldsymbol{a})$, for all $k \geq 0$. Note that $y_k$ is always bounded by $2Rn/(1 - \gamma)$, since the reward function is bounded by $R$, and, every timestep we update the sum of $Q_i$-value estimates over $i \in \mathcal{N}$ with a convex combination of the previous sum of $Q_i$-value estimates over $i \in \mathcal{N}$ and $\left( \sum_i r_i(s, \boldsymbol{a}) + \gamma \sum_{s' \in \mathcal{S}} \mathbb{P}(s' \mid s, \boldsymbol{a}) V(s') \right) \leq \left( 2Rn + \gamma \sum_{s' \in \mathcal{S}} \mathbb{P}(s' \mid s, \boldsymbol{a}) \max_{\boldsymbol{a}'} Q(s', \boldsymbol{a}') \right) \leq 2Rn/(1 - \gamma)$, so we can recursively show all the sum of $Q_i$-value estimates over $i \in \mathcal{N}$ iterates are bounded. Therefore, Lemma 1 yields that $\max_{s' \in \mathcal{S}} h(\widehat{\boldsymbol{Q}}^{(k)}(s')) = \|y_k\|_\infty \to 0$ as $k \to \infty$. As a byproduct, we also have $\lim_{k \to \infty} \left| \sum_{i \in \mathcal{N}} \widehat{V}_i^{(k)}(s) \right| = 0$, completing the proof. $\qquad \square$

For $\boldsymbol{Q} \in \mathbb{R}^{\sum_{i \in \mathcal{N}} |\mathcal{A}_i| \times \sum_{i \in \mathcal{N}} |\mathcal{A}_i|}$ defined in (29), we denote $\boldsymbol{Q}_1 := (Q_{1,2}, \dots, Q_{1,n})$ and $\boldsymbol{Q}_{-1} := (Q_{2,1}^\mathsf{T}, \dots, Q_{n,1}^\mathsf{T})^\mathsf{T}$. Then, we define $\mathrm{Val}_1$ and $\mathrm{Val}_{-1}$, which are maxmin values with respect to $\boldsymbol{Q}_1$ and $\boldsymbol{Q}_{-1}$, respectively, as follows:

$$\mathrm{Val}_1(\boldsymbol{Q}_1(s)) = \max_{\mu_1 \in \Delta(\mathcal{A}_1)} \min_{\mu_2 \in \Delta(\mathcal{A}_2), \dots, \mu_n \in \Delta(\mathcal{A}_n)} \sum_{i \in \mathcal{N}/\{1\}} \mu_1^\mathsf{T} Q_{1,i}(s) \mu_i$$

$$\mathrm{Val}_{-1}(\boldsymbol{Q}_{-1}(s)) = \max_{\mu_2 \in \Delta(\mathcal{A}_2), \dots, \mu_n \in \Delta(\mathcal{A}_n)} \min_{\mu_1 \in \Delta(\mathcal{A}_1)} \sum_{i \in \mathcal{N}/\{1\}} \mu_1^\mathsf{T} Q_{i,1}^\mathsf{T}(s) \mu_i.$$

Note that the Val operator can be viewed as the maxmin operator in the two-player zero-sum case, and it is indeed the star-shaped topology that enables us to write out a value iteration operator based on it, whose fixed point corresponds to the NE of the game. In general, it is hard to define such a value-iteration operator for other network structures. Also, note that since the maxmin formulas in $\mathrm{Val}_1$ and $\mathrm{Val}_{-1}$ are by definition *non-expansive*, the induced value iteration operator is *contracting* (due to the $\gamma \in (0, 1)$ discount factor), which is key in showing the convergence of our FP dynamics.

**Claim 9.** $|\widehat{V}_1^{(k)}(s) - \mathrm{Val}_1(\widehat{\boldsymbol{Q}}_1^{(k)}(s))|$ and $|\sum_{i \in \mathcal{N}/\{1\}} \widehat{V}_i^{(k)}(s) - \mathrm{Val}_{-1}(\widehat{\boldsymbol{Q}}_{-1}^{(k)}(s))|$ converge to 0 for all $s \in \mathcal{S}$.

*Proof.* The definition of $\widehat{V}_i^{(k)}$ gives

$$\widehat{V}_1^{(k)}(s) = \max_{a_i \in \mathcal{A}_i} \mathbb{E}_{a_{-i} \sim \widehat{\pi}_{-1}^{(k)}} \{\widehat{Q}_1^{(k)}(s, \boldsymbol{a})\} \geq \mathrm{Val}_1(\widehat{\boldsymbol{Q}}_1^{(k)}(s)) \geq \min_{\mu_2,\ldots,\mu_n} \sum_{i \in \mathcal{N}/\{1\}} (\widehat{\pi}_1^{(k)})^{\mathsf{T}} \widehat{Q}_{1,i}^{(k)}(s) \mu_i$$

$$\geq \min_{\mu_2,\ldots,\mu_n} \sum_{i \in \mathcal{N}/\{1\}} (\widehat{\pi}_1^{(k)})^{\mathsf{T}} (-(\widehat{Q}_{i,1}^{(k)}(s))^{\mathsf{T}}) \mu_i + \min_{\mu_2,\ldots,\mu_n} \sum_{i \in \mathcal{N}/\{1\}} (\widehat{\pi}_1^{(k)})^{\mathsf{T}} (\widehat{Q}_{1,i}^{(k)}(s) + (\widehat{Q}_{i,1}^{(k)}(s))^{\mathsf{T}}) \mu_i$$

$$\geq -\max_{\mu_2,\ldots,\mu_n} \sum_{i \in \mathcal{N}/\{1\}} (\widehat{\pi}_1^{(k)})^{\mathsf{T}} (\widehat{Q}_{i,1}^{(k)}(s))^{\mathsf{T}} \mu_i + \min_{\mu_2,\ldots,\mu_n} \sum_{i \in \mathcal{N}/\{1\}} (\widehat{\pi}_1^{(k)})^{\mathsf{T}} (\widehat{Q}_{1,i}^{(k)}(s) + (\widehat{Q}_{i,1}^{(k)}(s))^{\mathsf{T}}) \mu_i$$

$$\geq -\max_{\mu_2,\ldots,\mu_n} \sum_{i \in \mathcal{N}/\{1\}} (\widehat{\pi}_1^{(k)})^{\mathsf{T}} (\widehat{Q}_{i,1}^{(k)}(s))^{\mathsf{T}} \mu_i - h(\widehat{\boldsymbol{Q}}^{(k)}(s)) = -\sum_{i \in \mathcal{N}/\{1\}} \widehat{V}_i^{(k)}(s) - h(\widehat{\boldsymbol{Q}}^{(k)}(s)),$$

where the third inequality is due to the summation of minimization being no greater than the minimization, and the fifth inequality is from the definition of $h$. The above inequality further implies

$$\widehat{V}_1^{(k)}(s) + \sum_{i \in \mathcal{N}/\{1\}} \widehat{V}_i^{(k)}(s) + h(\widehat{\boldsymbol{Q}}^{(k)}(s)) \geq \widehat{V}_1^{(k)}(s) - \mathrm{Val}_1(\widehat{\boldsymbol{Q}}_1^{(k)}(s)) \geq 0.$$

The left-hand side goes to zero when $k \to \infty$, so the lemma is proved for player 1. The other direction can be proved in the same way. $\qquad\square$

Then, we define Shapley's minimax value-iteration operator $\mathcal{T}_1 : \mathbb{R}^{|\mathcal{S}| \times (|\mathcal{A}_1| \times \sum_{i \in \mathcal{N}/\{1\}} |\mathcal{A}_i|)} \to \mathbb{R}^{|\mathcal{S}| \times (|\mathcal{A}_1| \times \sum_{i \in \mathcal{N}/\{1\}} |\mathcal{A}_i|)}$ and $\mathcal{T}_{-1} : \mathbb{R}^{|\mathcal{S}| \times (\sum_{i \in \mathcal{N}/\{1\}} |\mathcal{A}_i| \times |\mathcal{A}_1|)} \to \mathbb{R}^{|\mathcal{S}| \times (\sum_{i \in \mathcal{N}/\{1\}} |\mathcal{A}_i| \times |\mathcal{A}_1|)}$ as in a two-player zero-sum Markov game [13] that

$$(\mathcal{T}_1 \boldsymbol{Q}_1)(s, a_1, a_i) = r_1(s, a_1, a_i) + \gamma \sum_{s' \in \mathcal{S}} \frac{1}{n-1} \mathbb{P}_1(s' \mid s, a_1) \mathrm{Val}_1(\boldsymbol{Q}_1(s'))$$

$$(\mathcal{T}_{-1} \boldsymbol{Q}_{-1})(s, a_i, a_1) = r_i(s, a_i, a_1) + \gamma \sum_{s' \in \mathcal{S}} \frac{1}{n-1} \mathbb{P}_1(s' \mid s, a_1) \mathrm{Val}_{-1}(\boldsymbol{Q}_{-1}(s')).$$

Also, we define several norms:

$\|\cdot\|_{\max} : \mathbb{R}^{m \times n} \to \mathbb{R}$ such that $\|A\|_{\max} = \max_{i \in [m], j \in [n]} |A_{i,j}|$

$\|\cdot\|_{\max,1} : \mathbb{R}^{|\mathcal{A}_1| \times \sum_{i \in \mathcal{N}/\{1\}} |\mathcal{A}_i|} \to \mathbb{R}$ such that $\|X_s\|_{\max,1} := \sum_{i \in \mathcal{N}/\{1\}} \|X_{s,i}\|_{\max}$

$\|\cdot\|_{\max,1,\max} : \mathbb{R}^{|\mathcal{S}| \times (|\mathcal{A}_1| \times \sum_{i \in \mathcal{N}/\{1\}} |\mathcal{A}_i|)} \to \mathbb{R}$ such that $\|X\|_{\max,1} := \left\| (\|X_s\|_{\max,1})_{s \in \mathcal{S}} \right\|_{\max}$.

**Claim 10.** $\mathcal{T}_1$ and $\mathcal{T}_{-1}$ are contracting with respect to the norm $\|\cdot\|_{\max,1,\max}$.

*Proof.* By definition of the Val operator, we have that for any $\boldsymbol{Q}_1 = (Q_{1,2}, \ldots, Q_{1,n}) \in \mathbb{R}^{|\mathcal{A}_1| \times \sum_{j \in \mathcal{N}/\{1\}} |\mathcal{A}_j|}$ and $\boldsymbol{Q}_1' = (Q_{1,2}', \ldots, Q_{1,n}') \in \mathbb{R}^{|\mathcal{A}_1| \times \sum_{j \in \mathcal{N}/\{1\}} |\mathcal{A}_j|}$ and any $i \in \mathcal{N}/\{1\}$, $s \in \mathcal{S}$,

$$\|((\mathcal{T}_1 \boldsymbol{Q}_1)(s, a_1, a_i) - (\mathcal{T}_1 \boldsymbol{Q}_1')(s, a_1, a_i))_i\|_{\max}$$

$$= \frac{\gamma}{n-1} \sum_{s' \in \mathcal{S}} \mathbb{P}_1(s' \mid s, a_1)(\mathrm{Val}_1(\boldsymbol{Q}_1(s')) - \mathrm{Val}_1(\boldsymbol{Q}_1'(s')))$$

$$\leq \frac{\gamma}{n-1} \sum_{s' \in \mathcal{S}} \mathbb{P}_1(s' \mid s, a_1) \max_{a_1} \left| \max_{a_2,\ldots,a_n} \sum_{i \in \mathcal{N}/\{1\}} (Q_{1,i} - Q_{1,i}')(s', a_1, a_i) \right|$$

$$\leq \frac{\gamma}{n-1} \|\boldsymbol{Q}_1 - \boldsymbol{Q}_1'\|_{\max,1,\max},$$

so for any $s \in \mathcal{S}$, $\|((\mathcal{T}_1 \boldsymbol{Q}_1)(s, a_1, a_i) - (\mathcal{T}_1 \boldsymbol{Q}_1')(s, a_1, a_i))_i\|_{\max,1} \leq \gamma \|\boldsymbol{Q}_1 - \boldsymbol{Q}_1'\|_{\max,1,\max}$ holds and therefore $\|(\mathcal{T}_1 \boldsymbol{Q}_1 - \mathcal{T}_1 \boldsymbol{Q}_1')_i\|_{\max,1,\max} \leq \gamma \|\boldsymbol{Q}_1 - \boldsymbol{Q}_1'\|_{\max,1,\max}$.

Similarly, we have that for any $\boldsymbol{Q}_{-1} = (Q_{2,1}^\mathsf{T}, \ldots, Q_{n,1}^\mathsf{T})^\mathsf{T} \in \mathbb{R}^{\sum_{j \in \mathcal{N}/\{1\}} |\mathcal{A}_j| \times |\mathcal{A}_1|}$ and $\boldsymbol{Q}'_{-1} = (Q_{2,1}'^\mathsf{T}, \ldots, Q_{n,1}'^\mathsf{T})^\mathsf{T} \in \mathbb{R}^{\sum_{j \in \mathcal{N}/\{1\}} |\mathcal{A}_j| \times |\mathcal{A}_1|}$

$$\left\| \left( (\mathcal{T}_{-1}\boldsymbol{Q}_{-1})(s, a_i, a_1) - (\mathcal{T}_{-1}\boldsymbol{Q}'_{-1})(s, a_i, a_1) \right)_i \right\|_{\max}$$

$$= \frac{\gamma}{n-1} \sum_{s' \in \mathcal{S}} \mathbb{P}_1(s' \mid s, a_1)(\text{Val}_{-1}(\boldsymbol{Q}_{-1}(s')) - \text{Val}_{-1}(\boldsymbol{Q}'_{-1}(s')))$$

$$\leq \frac{\gamma}{n-1} \sum_{s' \in \mathcal{S}} \mathbb{P}_1(s' \mid s, a_1) \max_{a_2, \ldots, a_n} \left| \max_{a_1} \sum_{i \in \mathcal{N}/\{1\}} (Q_{i,1} - Q'_{i,1})(s', a_i, a_1) \right|$$

$$\leq \frac{\gamma}{n-1} \left\| \boldsymbol{Q}_{-1} - \boldsymbol{Q}'_{-1} \right\|_{\max,1,\max},$$

so for any $s \in \mathcal{S}$, $\left\| \left( (\mathcal{T}_{-1}\boldsymbol{Q}_{-1})(s, a_i, a_1) - (\mathcal{T}_{-1}\boldsymbol{Q}'_{-1})(s, a_i, a_1) \right)_i \right\|_{\max,1} \leq \gamma \left\| \boldsymbol{Q}_{-1} - \boldsymbol{Q}'_{-1} \right\|_{\max,1,\max}$ holds and therefore $\left\| (\mathcal{T}_{-1}\boldsymbol{Q}_{-1} - \mathcal{T}_{-1}\boldsymbol{Q}'_{-1})_i \right\|_{\max,1,\max} \leq \gamma \left\| \boldsymbol{Q}_{-1} - \boldsymbol{Q}'_{-1} \right\|_{\max,1,\max}$. $\qquad\square$

Since the operators $\mathcal{T}_1$ and $\mathcal{T}_{-1}$ are contracting, they have a unique fixed point denoted by $\boldsymbol{Q}_1^\star$ and $\boldsymbol{Q}_{-1}^\star$. Moreover, we can guarantee that $\sum_{i \in \mathcal{N}/\{1\}} Q_{1,i}^\star(s, a_1, a_i) + \sum_{i \in \mathcal{N}/\{1\}} Q_{i,1}^\star(s, a_i, a_1) = 0$ for every $(s, \boldsymbol{a})$ and $i \in \mathcal{N}/\{1\}$ since we can interpret it as a fixed point of zero-sum two-player Markov game and $Q^\star$ are the $Q$ value. The update of beliefs on the $Q$-function can be written as

$$\widehat{Q}_1^{(k+1)}(s, \boldsymbol{a}) = (1 - \bar{\beta}_k(s))\widehat{Q}_1^{(k)}(s, \boldsymbol{a}) + \bar{\beta}_k(s) \left( \sum_{i \in \mathcal{N}/\{1\}} \mathcal{T}_1 \widehat{\boldsymbol{Q}}_1^{(k)}(s, a_1, a_i) + \mathcal{E}_1^{(k)}(s, \boldsymbol{a}) \right)$$

$$\sum_{i \in \mathcal{N}/\{1\}} \widehat{Q}_i^{(k+1)}(s, \boldsymbol{a}) = (1 - \bar{\beta}_k(s)) \sum_{i \in \mathcal{N}/\{1\}} \widehat{Q}_i^{(k)}(s, \boldsymbol{a})$$

$$+ \bar{\beta}_k(s) \left( \sum_{i \in \mathcal{N}/\{1\}} \mathcal{T}_{-1} \widehat{\boldsymbol{Q}}_{-1}^{(k)}(s, a_i, a_1) + \mathcal{E}_{-1}^{(k)}(s, \boldsymbol{a}) \right).$$

Here, $\mathcal{E}_1^{(k)}(s, \boldsymbol{a})$ and $\mathcal{E}_{-1}^{(k)}(s, \boldsymbol{a})$ are defined as

$$\mathcal{E}_1^{(k)}(s, \boldsymbol{a}) = \gamma \sum_{s' \in \mathcal{S}} \mathbb{P}_1(s' \mid s, a_1) \left[ \widehat{V}_1^{(k)}(s') - \text{Val}_1(\widehat{\boldsymbol{Q}}_1^{(k)}(s')) \right]$$

$$\mathcal{E}_{-1}^{(k)}(s, \boldsymbol{a}) = \gamma \sum_{s' \in \mathcal{S}} \mathbb{P}_1(s' \mid s, a_1) \left[ \sum_{i \in \mathcal{N}/\{1\}} \widehat{V}_i^{(k)}(s') - \text{Val}_{-1}(\widehat{\boldsymbol{Q}}_{-1}^{(k)}(s')) \right]$$

where the two values go to 0 by Claim 9.

**Claim 11.** $|\widehat{Q}_1^{(k)}(s, \boldsymbol{a}) - \sum_{i \in \mathcal{N}/\{1\}} Q_1^\star(s, a_1, a_i)|$ and $|\widehat{Q}_{-1}^{(k)}(s, \boldsymbol{a}) - \sum_{i \in \mathcal{N}/\{1\}} Q_{-1}^\star(s, a_i, a_1)|$ converge to 0 as $k \to \infty$ for all $s \in \mathcal{S}$ and $\boldsymbol{a} \in \mathcal{A}$.

*Proof.* Define $\widetilde{Q}_1^{(k)}(s, \boldsymbol{a}) := \widehat{Q}_1^{(k)}(s, \boldsymbol{a}) - \sum_{i \in \mathcal{N}/\{1\}} Q_1^\star(s, a_1, a_i)$ and $\widetilde{Q}_{-1}^{(k)}(s, \boldsymbol{a}) := \widehat{Q}_{-1}^{(k)}(s, \boldsymbol{a}) - \sum_{i \in \mathcal{N}/\{1\}} Q_{-1}^\star(s, a_i, a_1)$. Then, by using $Q_1^\star$ and $Q_{-1}^\star$ as the fixed point, we have that for each $s \in \mathcal{S}$ and $\boldsymbol{a} \in \mathcal{A}$:

$$\widetilde{Q}_1^{(k+1)}(s, \boldsymbol{a}) = (1 - \bar{\beta}_k(s))\widetilde{Q}_1^{(k)}(s, \boldsymbol{a})$$

$$+ \bar{\beta}_k(s) \left( \sum_{i \in \mathcal{N}/\{1\}} \mathcal{T}_1 \widehat{\boldsymbol{Q}}_1^{(k)}(s, a_1, a_i) - \sum_{i \in \mathcal{N}/\{1\}} \mathcal{T}_1 \boldsymbol{Q}_1^\star(s, a_1. a_i) + \mathcal{E}_1^{(k)}(s, \boldsymbol{a}) \right)$$

$$\widetilde{Q}_{-1}^{(k+1)}(s, \boldsymbol{a}) = (1 - \bar{\beta}_k(s))\widetilde{Q}_{-1}^{(k)}(s, \boldsymbol{a})$$

$$+ \bar{\beta}_k(s) \left( \sum_{i \in \mathcal{N}/\{1\}} \mathcal{T}_{-1} \widehat{\boldsymbol{Q}}_{-1}^{(k)}(s, a_i, a_1) - \sum_{i \in \mathcal{N}/\{1\}} \mathcal{T}_{-1} \boldsymbol{Q}_{-1}^\star(s, a_i, a_1) + \mathcal{E}_{-1}^{(k)}(s, \boldsymbol{a}) \right)$$

and Claim 10 implies that

$$\widetilde{Q}_1^{(k+1)}(s, \boldsymbol{a}) \leq (1 - \bar{\beta}_k(s))\widetilde{Q}_1^{(k)}(s, \boldsymbol{a}) + \bar{\beta}_k(s) \left( \gamma \left\| \widetilde{Q}_1 \right\|_{\max,1,\max} + \bar{\epsilon}^{(k)} \right)$$

$$\widetilde{Q}_1^{(k+1)}(s, \boldsymbol{a}) \geq (1 - \bar{\beta}_k(s))\widetilde{Q}_1^{(k)}(s, \boldsymbol{a}) + \bar{\beta}_k(s) \left( -\gamma \left\| \widetilde{Q}_1 \right\|_{\max,1,\max} - \bar{\epsilon}^{(k)} \right)$$

which yields $\widetilde{Q}_1^{(k+1)}(s, \boldsymbol{a}) \to 0$ and also $\widetilde{Q}_{-1}^{(k+1)}(s, \boldsymbol{a}) \to 0$ as $k \to \infty$ by Lemma 1. □

Therefore, we verified that $\widehat{V}_1^{(k)}(s) - \mathrm{Val}_1(Q_1^\star(s)) \to 0$ and $\widehat{V}_{-1}^{(k)}(s) - \mathrm{Val}_{-1}(Q_{-1}^\star(s)) \to 0$ for every $s \in \mathcal{S}$. Therefore, the beliefs on the opponents' policies converge to a (perfect) Nash equilibrium of the underlying zero-sum NMG. □

**Remark 8.** This can be also done with the stochastic fictitious-play dynamics, in a similar way as the argument in Section D.1.2.

**Remark 9** (Stationary equilibrium computation via value iteration)**.** By Claim 10, we know that with a star-shaped topology, we can formulate a contracting value iteration operator, which plays an important role in showing the convergence of fictitious play. In fact, iterating such a contracting operator, which leads to the *value iteration* algorithm, can lead to efficient NE computation in this star-shaped case also, with a fixed constant $\gamma$. This folklore result supplements the hardness results in Theorem 1, where stationary equilibria computation in cases other than the star-shaped ones are computationally intractable. This completes the landscape of stationary equilibria computation in zero-sum NMGs. We provide the value-iteration process in Algorithm 5. One can guarantee that $Q_1(s, \boldsymbol{a}) := \sum_{i \in \mathcal{N}/\{1\}} Q_{1,i}(s, a_1, a_i)$ converges to the $Q_1^\star(s, \boldsymbol{a})$, which corresponds to the Nash equilibrium values of the zero-sum NMG. Also, by solving the maxmin problem in (33), $Q_{1,i}(s)$ provides an approximate NE policy.

---

**Algorithm 5** Value iteration for zero-sum NMGs of a star-shape

---

Initialize $Q_{1,i}(s, a_1, a_i) = 0, Q_{i,1}(s, a_i, a_1) = 0$ for all $s \in \mathcal{S}, \boldsymbol{a} \in \mathcal{A}, i \in \mathcal{N}/\{1\}$ and $V_i(s) = 0$ for all $s \in \mathcal{S}, i \in \mathcal{N}$
**for** each iteration $t = 0, 1, \ldots$ **do**
 Find $\mu$ for each $s \in \mathcal{S}$ such that

$$\mu(s) \in \operatorname*{argmax}_{\mu_1 \in \Delta(\mathcal{A}_1)} \operatorname*{argmin}_{\mu_2 \in \Delta(\mathcal{A}_2),\ldots,\mu_n \in \Delta(\mathcal{A}_n)} \sum_{i \in \mathcal{N}/\{1\}} \mu_1^\intercal Q_{1,i}(s)\mu_i \tag{33}$$

 Update $V_1(s) = \sum_{i \in \mathcal{N}/\{1\}} \mu_1^\intercal(s)Q_{1,i}(s)\mu_i(s)$ for all $s \in \mathcal{S}$
 Update $Q_{1,i}(s, a_1, a_i) = r_{1,i}(s, a_1, a_i) + \gamma \sum_{s' \in \mathcal{S}} \frac{1}{n-1}\mathbb{P}_1(s' \mid s, a_1)V_1(s')$ for all $i \in \mathcal{N}/\{1\}$, $s \in \mathcal{S}, \boldsymbol{a} \in \mathcal{A}$
**end for**

---

# E  Background on Stochastic Approximation and Differential Inclusions

Section E introduces the theorem statement of [44]. Let $F : \mathbb{R}^m \rightrightarrows \mathbb{R}^m$ be a set-valued function. Assume that $F$ satisfies the following properties:

1. $F$ is a closed set-valued map, meaning that its graph $\mathrm{Graph}(F) = \{(x, y) : y \in F(x)\}$ is a closed subset of $\mathbb{R}^m \times \mathbb{R}^m$.

2. $F(x)$ is a non-empty, compact, and convex subset of $\mathbb{R}^m$ for all $x \in \mathbb{R}^m$.

3. There exists a constant $c > 0$ such that for all $x \in \mathbb{R}^m$, we have $\sup_{z \in F(x)} |z| \leq c(1 + |x|)$.

The differential inclusion problem involves finding a solution vector function $\boldsymbol{x} : \mathbb{R} \to \mathbb{R}^m$ that satisfies the initial condition $\boldsymbol{x}(0) = x \in \mathbb{R}^m$ and the following relationship for almost all $t \in \mathbb{R}$:

$$\frac{d\boldsymbol{x}(t)}{dt} \in F(\boldsymbol{x}(t)).$$

**Definition 7** (Perturbed solutions). A perturbed solution to $F$ refers to a continuous function $\boldsymbol{y}$ : $\mathbb{R}_+ = [0, \infty) \to \mathbb{R}^m$ that meets the following requirements:

- $\boldsymbol{y}$ is absolutely continuous.

- There is a locally integrable function $t \mapsto U(t)$ that satisfies:

    - For all $T > 0$, the supremum of $\left| \int_t^{t+v} U(s)ds \right|$ over the interval $0 \leq v \leq T$ converges to zero as $t \to \infty$.
    - For almost every $t > 0$, the expression $\frac{d\boldsymbol{y}(t)}{dt} - U(t)$ belongs to $F^{\delta(t)}(\boldsymbol{y}(t))$, where $\delta : [0, \infty) \to \mathbb{R}$ is a function such that $\delta(t) \to 0$ as $t \to \infty$.

**Definition 8** (Stochastic approximations). A discrete-time process $\{x_n\}_{n \in \mathbb{N}}$ is a stochastic approximation if it satisfies the following relationship:

$$x_{n+1} - x_n - \gamma_{n+1}U_{n+1} \in \gamma_{n+1}F(x_n),$$

where the characteristics $\gamma$ and $U$ meet the following conditions:

- The sequence $(\gamma_n)_{n \geq 1}$ consists of non-negative numbers such that $\sum_{n=1}^{\infty} \gamma_n = \infty$ and $\lim_{n \to \infty} \gamma_n = 0$.

- The elements $U_n \in \mathbb{R}^m$ can be either deterministic or random perturbations.

A continuous-time process can be associated with such a process as follows:

**Definition 9** (Affine interpolated process). Define the following: $\tau_0 = 0$ and $\tau_n = \sum_{i=1}^{n} \gamma_i$ for $n \geq 1$. The continuous-time *affine interpolated process* $\boldsymbol{w} : \mathbb{R}_+ \to \mathbb{R}^m$ is defined as:

$$\boldsymbol{w}(\tau_n + s) := x_n + s \frac{x_{n+1} - x_n}{\tau_{n+1} - \tau_n}, \quad s \in [0, \gamma_{n+1}).$$

We define $\Phi_t(x) = \{\boldsymbol{x}(t) : \boldsymbol{x} \text{ is a solution to } \frac{d\boldsymbol{x}(t)}{dt} \in F(\boldsymbol{x}(t)) \text{ with } \boldsymbol{x}(0) = x\}$.

**Definition 10** (Lyapunov function). Lyapunov function for a set $\mathcal{S}$ is a continuous function $V$ : $\mathbb{R}^m \to \mathbb{R}$ if $V(y) < V(x)$ for all $x \in \mathcal{S} \subseteq \mathbb{R}^m$, $y \in \Phi_t(x), t > 0$, and $V(y) \leq V(x)$ for all $x \in \mathcal{S}$, $y \in \Phi_t(x), t > 0$.

**Theorem 5.** Assume that the following hold:

- For all $T > 0$, the supremum of $\left\| \sum_{i=n}^{k-1} \gamma_{i+1}U_{i+1} \right\|$ for $k = n + 1, \ldots, m(\tau_n + T)$ converges to zero as $n \to \infty$, where

$$m(t) = \sup\{k \geq 0 : t \geq \tau_k\}.$$

- $\sup_n \|x_n\| = M < \infty$.

Then the affine interpolated process (c.f. Definition 9) is a perturbed solution.

**Theorem 6.** Suppose that $V$ is a Lyapunov function for a set $\Lambda$. Assume that $V(\Lambda)$ has an empty interior. For every bounded perturbed solution $\boldsymbol{y}$, define $L(\boldsymbol{y}) = \bigcap_{t \geq 0} \overline{\{\boldsymbol{y}(s) : s \geq t\}}$, then $L(\boldsymbol{y})$ is contained in $\Lambda$ and $V(L(\boldsymbol{y}))$ is constant.

# F  Omitted Details in Section 6

We first recall the folklore result that approximating Markov non-stationary NE in *infinite-horizon* discounted settings can be achieved by finding approximate Markov NE in *finite-horizon* settings, with a large enough horizon length.

**Proposition 10.** A $2\epsilon$-approximate Markov non-stationary NE in an infinite-horizon $\gamma$-discounted MG can be generated by (1) truncating the trajectory at time step $H \geq \frac{\log(R/\epsilon)}{1-\gamma}$ and (2) finding an $\epsilon$-approximate Markov NE in the $H$-horizon MG.

*Proof.* We will execute a policy $\pi$ such that (1) for the first $H$ steps, we follow $\epsilon$-approximate NE in the $H$-truncated MG, and (2) after the $H$ steps, we follow an arbitrary policy. Then, we have

$$V_i^\pi(s) = \sum_{h=1}^\infty \mathbb{E}_\pi[\gamma^{h-1} r_i(s_h, a_h) \mid s_1 = s]$$

$$\geq \sum_{h=1}^H \mathbb{E}_{\mu_i, \pi_{-i}}[\gamma^{h-1} r_i(s_h, a_h) \mid s_1 = s] - \epsilon + \sum_{h=H+1}^\infty \mathbb{E}_\pi[\gamma^{h-1} r_i(s_h, a_h) \mid s_1 = s]$$

$$\geq \sum_{h=1}^H \mathbb{E}_{\mu_i, \pi_{-i}}[\gamma^{h-1} r_i(s_h, a_h) \mid s_1 = s] - \epsilon + \sum_{h=H+1}^\infty \mathbb{E}_{\mu_i, \pi_{-i}}[\gamma^{h-1} r_i(s_h, a_h) \mid s_1 = s] - \frac{R\gamma^{h-1}}{1-\gamma}$$

$$\geq \sum_{h=1}^\infty \mathbb{E}_{\mu_i, \pi_{-i}}[\gamma^{h-1} r_i(s_h, a_h) \mid s_1 = s] - 2\epsilon$$

for an arbitrary policy $\mu_i$. Here, the first inequality comes from the definition of NE in the $H$-truncated MG, the second inequality comes from $0 \leq r_i \leq R$ and the last inequality comes from the definition of $H$. Therefore, the executed policy is a $2\epsilon$-approximate NE. $\qquad\square$

In this section, we will utilize two performance metrics: Matrix-NE-Gap and Matrix-QRE-Gap. For the definition of QRE, we refer to Definition 6 and (25).

**Definition 11.** For an NG $M$ with $(\mathcal{G} = (\mathcal{N}, \mathcal{E}_r), \mathcal{A} = \prod_{i \in \mathcal{N}} \mathcal{A}_i, (r_{i,j})_{(i,j) \in \mathcal{E}_r})$, we define Matrix-NE-Gap and Matrix-QRE-Gap of $M$ for some product policy $\pi$ as follows:

$$\mathsf{Matrix\text{-}NE\text{-}Gap}(M, \pi) = \max_{i \in \mathcal{N}} \max_{\pi_i' \in \Delta(\mathcal{A}_i)} (r_i(\pi_i', \pi_{-i}) - r_i(\pi))$$

$$= \max_{i \in \mathcal{N}} \left[ \max_{\pi_i' \in \Delta(\mathcal{A}_i)} \left( \sum_{j \in \mathcal{E}_{r,i}} \pi_i' r_{i,j} \pi_j \right) - \left( \sum_{j \in \mathcal{E}_{r,i}} \pi_i r_{i,j} \pi_j \right) \right].$$

$$\mathsf{Matrix\text{-}QRE\text{-}Gap}_\tau(M, \pi) = \max_{i \in \mathcal{N}} \max_{\pi_i' \in \Delta(\mathcal{A}_i)} (r_{\tau,i}(\pi_i', \pi_{-i}) - r_{\tau,i}(\pi))$$

$$= \max_{i \in \mathcal{N}} \left[ \max_{\pi_i' \in \Delta(\mathcal{A}_i)} \left( \sum_{j \in \mathcal{E}_{r,i}} \pi_i' r_{i,j} \pi_j + \tau \mathcal{H}(\pi_i') \right) - \left( \sum_{j \in \mathcal{E}_{r,i}} \pi_i r_{i,j} \pi_j + \tau \mathcal{H}(\pi_i) \right) \right].$$

When the underlying graph and the action space for the NG are clear, we also write $\mathsf{Matrix\text{-}NE\text{-}Gap}(M, \pi)$ or $\mathsf{Matrix\text{-}QRE\text{-}Gap}_\tau(M, \pi)$ as $\mathsf{Matrix\text{-}NE\text{-}Gap}(\boldsymbol{r}, \pi)$ or $\mathsf{Matrix\text{-}QRE\text{-}Gap}_\tau(\boldsymbol{r}, \pi)$, respectively.

We now provide the relationship between $\mathsf{Matrix\text{-}NE\text{-}Gap}(\boldsymbol{r}, \pi)$ and $\mathsf{Matrix\text{-}QRE\text{-}Gap}_\tau(\boldsymbol{r}, \pi)$.

**Lemma 2** ([51], Page 6, Equation (8))**.** For an NG $M$ with $(\mathcal{G} = (\mathcal{N}, \mathcal{E}_r), \mathcal{A} = \prod_{i \in \mathcal{N}} \mathcal{A}_i, (r_{i,j})_{(i,j) \in \mathcal{E}_r})$, the following holds:

$$\mathsf{Matrix\text{-}NE\text{-}Gap}(\boldsymbol{r}, \pi) \leq \mathsf{Matrix\text{-}QRE\text{-}Gap}_\tau(\boldsymbol{r}, \pi) + \tau \max_{i \in \mathcal{N}} \log |\mathcal{A}_i|.$$

Thus, setting $\tau = \frac{\epsilon}{2 \max_{i \in \mathcal{N}} \log |\mathcal{A}_i|}$, then an $\epsilon/2$-approximate QRE is also an $\epsilon$-approximate NE. Hence, finding an approximate-QRE for zero-sum NGs is sufficient for finding an approximate-NE, with a small enough $\tau$. Now, we define the NE-Gap for an MNMG.

**Definition 12.** For an MNMG $M$ with $(\mathcal{G} = (\mathcal{N}, \mathcal{E}_Q), \mathcal{S}, \mathcal{A}, H, (\mathbb{P}_h)_{h \in [H]}, (r_{h,i,j}(s))_{(i,j) \in \mathcal{E}_Q, s \in \mathcal{S}})$, NE-Gap for some product policy $\pi$ at timestep $h \in [H]$ is defined as follows:

$$\mathsf{NE\text{-}Gap}_h(M, \pi) = \max_{i \in \mathcal{N}} \max_{s \in \mathcal{S}} \max_{\pi_i' \in \Delta(\mathcal{A}_i)^{|\mathcal{S}| \times H}} \left( V_{h,i}^{\pi_i', \pi_{-i}}(s) - V_{h,i}^\pi(s) \right).$$

Now we summarize the algorithm in Algorithm 6. By Algorithm 6, $Q_{h,i,j}(s, a_i, a_j) = Q_{\tau,h,i,j}^\pi(s, a_i, a_j)$ and $V_{h,i}(s) = V_{\tau,h,i}^\pi(s)$ holds, so that an NG characterized by $(\mathcal{G}, \mathcal{A}, (Q_{h,i,j}(s))_{(i,j) \in \mathcal{E}_Q})$ is always a zero-sum NG for all $s \in \mathcal{S}, h \in [H]$. And by induction, we can show that $|Q_{h,i,j}^\pi(s, a_i, a_j)| \leq (H + 1 - h)R$ for all $\pi, h \in [H], (i,j) \in \mathcal{E}_Q, s \in \mathcal{S}, a_i \in \mathcal{A}_i$, $a_j \in \mathcal{A}_j$ (i.e., $\|\boldsymbol{Q}_h^\pi(s)\|_{\max} \leq HR$ for all $\pi, h \in [H], s \in \mathcal{S}$).

**Algorithm 6** A value-iteration-based algorithm for finding NE in zero-sum NMGs

---

Update $V_{H+1,i}(s) = 0$ for all $s \in \mathcal{S}$ and $i \in \mathcal{N}$
**for** step $h = H, H-1, \ldots, 1$ **do**
  **if** $\mathcal{N}_C \neq \emptyset$ **then**
    Update $Q_{h,i,j}(s, a_i, a_j)$ for all $(i,j) \in \mathcal{E}_Q, s \in \mathcal{S}, a_i \in \mathcal{A}_i, a_j \in \mathcal{A}_j$ as

$$Q_{h,i,j}(s, a_i, a_j) = r_{h,i,j}(s, a_i, a_j)$$
$$+ \sum_{s' \in \mathcal{S}} \left( \frac{1}{|\mathcal{E}_{Q,i}|} \mathbf{1}(i \in \mathcal{N}_C) \mathbb{F}_{h,i}(s' \,|\, s, a_i) + \mathbf{1}(j \in \mathcal{N}_C) \mathbb{F}_{h,j}(s' \,|\, s, a_j) \right) \cdot V_{h+1,i}(s')$$

(34)

  **else if** $\mathcal{N}_C = \emptyset$ **then**
    Update $Q_{h,i,j}(s, a_i, a_j)$ for all $(i,j) \in \mathcal{E}_Q, s \in \mathcal{S}, a_i \in \mathcal{A}_i, a_j \in \mathcal{A}_j$ as

$$Q_{h,i,j}(s, a_i, a_j) = r_{h,i,j}(s, a_i, a_j) + \sum_{s' \in \mathcal{S}} \left( \frac{1}{|\mathcal{E}_{Q,i}|} \mathbf{1}(j \in \mathcal{E}_{Q,i}) \mathbb{F}_{h,o}(s' \,|\, s) \cdot V_{h+1,i}(s') \right)$$

(35)

  **end if**
  Update $\pi_h(s) = \mathsf{NE\text{-}ORACLE}(\mathcal{G}, \mathcal{A}, (Q_{h,i,j}(s))_{(i,j) \in \mathcal{E}_Q})$ for all $s \in \mathcal{S}$
  Update $V_{h,i}(s)$ for all $i \in \mathcal{N}, s \in \mathcal{S}$ as

$$V_{h,i}(s) = \sum_{j \in \mathcal{E}_{Q,i}} \pi_{h,i}^{\mathsf{T}}(s) Q_{h,i,j}(s) \pi_{h,j}(s)$$

(36)

**end for**

---

**Proposition 4.** Suppose that for all $h \in [H], i \in \mathcal{N}, s \in \mathcal{S}$, $\mathsf{NE\text{-}ORACLE}(\mathcal{G}, \mathcal{A}, (Q_{h,i,j}(s))_{(i,j) \in \mathcal{E}_Q})$ provides an $\epsilon_{h,s}$-approximate NE for the zero-sum NG $(\mathcal{G}, \mathcal{A}, (Q_{h,i,j}(s))_{(i,j) \in \mathcal{E}_Q})$ in Algorithm 6. Then, the output policy $\pi$ in Algorithm 6 is an $(\sum_{h \in [H]} \max_{s \in \mathcal{S}} \epsilon_{h,s})$-approximate NE for the corresponding zero-sum NMG $(\mathcal{G} = (\mathcal{N}, \mathcal{E}_Q), \mathcal{S}, \mathcal{A}, H, (\mathbb{P}_h)_{h \in [H]}, (r_{h,i,j}(s))_{(i,j) \in \mathcal{E}_Q, s \in \mathcal{S}})$.

*Proof.* Let $M = (\mathcal{G}, \mathcal{S}, \mathcal{A}, H, (\mathbb{P}_h)_{h \in [H]}, (r_{h,i,j}(s))_{(i,j) \in \mathcal{E}_Q, s \in \mathcal{S}})$. For any $\pi$ and $h \in [H]$, we have

$$\mathsf{NE\text{-}Gap}_h(M, \pi) = \max_{i \in \mathcal{N}} \max_{s \in \mathcal{S}} \max_{\pi_i' \in \Delta(\mathcal{A}_i)^{|\mathcal{S}| \times H}} \left( V_{h,i}^{\pi_i', \pi_{-i}}(s) - V_{h,i}^{\pi_i}(s) \right)$$

$$= \max_{i \in \mathcal{N}} \max_{s \in \mathcal{S}} \max_{\pi_i' \in \Delta(\mathcal{A}_i)^{|\mathcal{S}| \times H}} \left[ V_{h,i}^{\pi_i', \pi_{-i}}(s) - V_{h,i}^{(\pi_{h,i}', \pi_{h,-i}), \pi_{h+1:H}}(s) + V_{h,i}^{(\pi_{h,i}', \pi_{h,-i}), \pi_{h+1:H}}(s) - V_{h,i}^{\pi}(s) \right]$$

$$= \max_{i \in \mathcal{N}} \max_{s \in \mathcal{S}} \max_{\pi_i' \in \Delta(\mathcal{A}_i)^{|\mathcal{S}| \times H}} \left[ \mathbb{P}_{h,(\pi_{h,i}', \pi_{h,-i})}(V_{h+1,i}^{\pi_i', \pi_{-i}} - V_{h+1,i}^{\pi_{h+1:H}})(s) + V_{h,i}^{(\pi_{h,i}', \pi_{h,-i}), \pi_{h+1:H}}(s) - V_{h,i}^{\pi}(s) \right]$$

$$\leq \mathsf{NE\text{-}Gap}_{h+1}(M, \pi) + \max_{i \in \mathcal{N}} \max_{s \in \mathcal{S}} \max_{\pi_{h,i}'(s) \in \Delta(\mathcal{A}_i)} \sum_{i \in \mathcal{N}} \left[ V_{h,i}^{(\pi_{h,i}', \pi_{h,-i}), \pi_{h+1:H}}(s) - V_{h,i}^{\pi}(s) \right]$$

$$= \mathsf{NE\text{-}Gap}_{h+1}(M, \pi) + \max_{s \in \mathcal{S}} \mathsf{Matrix\text{-}NE\text{-}Gap}(\boldsymbol{Q}_h(s), \pi_h) \leq \mathsf{NE\text{-}Gap}_{h+1}(M, \pi) + \max_{s \in \mathcal{S}} \epsilon_{h,s}.$$

Therefore, for any $\pi$ and $h \in [H]$, we have $\mathsf{NE\text{-}Gap}_h(M, \pi) \leq \sum_{h \in [H]} \max_{s \in \mathcal{S}} \epsilon_{h,s}$. $\qquad\square$

### F.1 Several examples of **NE-ORACLE**

**Example 1. Optimism & Regularization: OMWU algorithm [51].** According to [51, Theorem 1], if we apply Algorithm 7 to $(\mathcal{G}, \mathcal{A}, (Q_{h,i,j}(s))_{(i,j) \in \mathcal{E}_Q})$ for each $h \in [H], s \in \mathcal{S}$, the required number of iterations for $\mathsf{Matrix\text{-}NE\text{-}Gap}(\boldsymbol{Q}_h^{\pi}(s), \pi_h) \leq \epsilon/H$ is $\widetilde{\mathcal{O}}\left(H^2 d_{\max}/\epsilon\right)$ where $d_{\max}$ is the maximum degree of underlying graph $\mathcal{G}$. Consequently, the overall iteration complexity is $\widetilde{\mathcal{O}}\left(H^3 d_{\max}|\mathcal{S}|/\epsilon\right)$. Note that these results are in terms of last-iterate convergence.

---

**Algorithm 7** OMWU for zero-sum NGs with $\tau$-entropy regularization [51]

---

Choose $\pi_i^{(0)}, \bar{\pi}_i^{(0)}$ as uniform distributions for all $i \in \mathcal{N}$
Define $\tau = 1/(n \max_{i \in \mathcal{N}} \log |\mathcal{A}_i|)$ and $\eta = 1/(8n \|\boldsymbol{r}\|_\infty)$
**for** timestep $t = 0, 1, \ldots,$ **do**
    Update the policy $\bar{\pi}_i^{(t+1)}$ as $\bar{\pi}_i^{(t+1)}(a_i) \propto \bar{\pi}_i^{(t)}(a_i)^{1-\eta\tau} \exp\left(\eta[\boldsymbol{r}_i\pi^{(t)}]_{a_i}\right)$ for all $i \in \mathcal{N}$ and $a_i \in \mathcal{A}_i$
    Update the policy $\pi_i^{(t+1)}$ as $\pi_i^{(t+1)}(a_i) \propto \bar{\pi}_i^{(t+1)}(a_i)^{1-\eta\tau} \exp\left(\eta[\boldsymbol{r}_i\pi^{(t)}]_{a_i}\right)$ for all $i \in \mathcal{N}$ and $a_i \in \mathcal{A}_i$
**end for**

---

| | Regularization | Regularization-free |
|---|---|---|
| Optimism | OMWU [51]: $\widetilde{\mathcal{O}}(1/\epsilon)$ last-iterate | OMD [54]: $\widetilde{\mathcal{O}}(1/\epsilon^2)$ best-iterate Asymptotic last-iterate 

 [114, 115, 116, 54]: $\widetilde{\mathcal{O}}(1/\epsilon)$ average-iterate + Marginalization |
| Optimism-free | Algorithm 9: $\widetilde{\mathcal{O}}(1/\epsilon^4)$ last-iterate 

 Algorithm 10: $\widetilde{\mathcal{O}}(1/\epsilon^6)$ last-iterate | Any no-regret learning algorithm with $\widetilde{\mathcal{O}}(1/\epsilon^2)$ average-iterate + Marginalization |

Table 1: Iteration complexities for finding an $\epsilon$-NE for a zero-sum NG with $(\mathcal{G} = (\mathcal{N}, \mathcal{E}), \mathcal{A}, (r_{i,j})_{(i,j)\in\mathcal{E}})$ with different NE-ORACLE subroutines. $\widetilde{\mathcal{O}}(\cdot)$ omits polylog terms and polynomial dependencies on $n, \|\boldsymbol{r}\|_{\max}, R$.

**Example 2. Optimism & Regularization-free: OMD algorithm [54].** According to [54, Theorem 3.4], if we apply Algorithm 8 to $(\mathcal{G}, \mathcal{A}, (Q_{h,i,j}(s))_{(i,j)\in\mathcal{E}_Q})$ for each $h \in [H], s \in \mathcal{S}$, the required number of iterations for Matrix-NE-Gap$(\boldsymbol{Q}_h^\pi(s), \pi_h) \leq \epsilon/H$ is $\widetilde{\mathcal{O}}\left(H^3 n/\epsilon^2\right)$. Consequently, the overall iteration complexity is $\widetilde{\mathcal{O}}\left(H^4 n |\mathcal{S}|/\epsilon^2\right)$. Note that these results are in terms of best-iterate convergence.

---

**Algorithm 8** OMD with KL-distance generating function for zero-sum NGs [54]

---

Choose $\pi_i^{(0)}, \bar{\pi}_i^{(0)}$ as uniform distributions for all $i \in \mathcal{N}$
Define $\eta = 1/(4n \|\boldsymbol{r}\|_\infty)$
**for** timestep $t = 0, 1, \ldots,$ **do**
    Update the policy $\pi_i^{(t+1)}$ as $\pi_i^{(t+1)}(a_i) \propto \bar{\pi}_i^{(t)}(a_i) \exp\left(\eta[\boldsymbol{r}_i\pi^{(t)}]_{a_i}\right)$ for all $i \in \mathcal{N}$ and $a_i \in \mathcal{A}_i$
    Update the policy $\bar{\pi}_i^{(t+1)}$ as $\bar{\pi}_i^{(t+1)}(a_i) \propto \bar{\pi}_i^{(t)}(a_i) \exp\left(\eta[\boldsymbol{r}_i\pi^{(t+1)}]_{a_i}\right)$ for all $i \in \mathcal{N}$ and $a_i \in \mathcal{A}_i$
**end for**

---

**Example 3. Optimism-free & Regularization: MWU algorithm.** We provide MWU for zero-sum NGs with regularization in Section F.2. According to Theorem 7 and Theorem 8, if we apply Algorithm 9 or Algorithm 10 to $(\mathcal{G}, \mathcal{A}, (Q_{h,i,j}(s))_{(i,j)\in\mathcal{E}_Q})$ for each $h \in [H], s \in \mathcal{S}$, the required number of iterations for Matrix-NE-Gap$(\boldsymbol{Q}_h^\pi(s), \pi_h) \leq \epsilon/H$ is $\widetilde{\mathcal{O}}\left(H^8 n/\epsilon^4\right)$ or $\widetilde{\mathcal{O}}\left(H^{18} n^3/\epsilon^6\right)$, respectively. Consequently, the overall iteration complexity is $\widetilde{\mathcal{O}}\left(H^9 n |\mathcal{S}|/\epsilon^4\right)$ or $\widetilde{\mathcal{O}}\left(H^{19} n^3 |\mathcal{S}|/\epsilon^6\right)$, respectively. Note that these results are in terms of last-iterate convergence.

| | Regularization | Regularization-free |
|---|---|---|
| Optimism | Algorithm 6 + OMWU [51]: $\widetilde{\mathcal{O}}(1/\epsilon)$ last-iterate | Algorithm 6 + OMD [54]: $\widetilde{\mathcal{O}}(1/\epsilon^2)$ best-iterate 

 Algorithm 6 + [114, 115, 116, 54]: $\widetilde{\mathcal{O}}(1/\epsilon)$ average-iterate + Marginalization |
| Optimism-free | Algorithm 6 + Algorithm 9: $\widetilde{\mathcal{O}}(1/\epsilon^4)$ last-iterate 

 Algorithm 6 + Algorithm 10: $\widetilde{\mathcal{O}}(1/\epsilon^6)$ last-iterate | Algorithm 6 + Any no-regret learning algorithm with $\widetilde{\mathcal{O}}(1/\epsilon^2)$ average-iterate + Marginalization |

Table 2: Iteration complexities for finding an $\epsilon$-NE for a zero-sum NMG with different NE-ORACLE subroutines. $\widetilde{\mathcal{O}}(\cdot)$ omits polylog terms and polynomial dependencies on $n, H, |\mathcal{S}|, R, \|r\|_{\max}$.

## F.2 Analysis of MWU for zero-sum NGs with regularization

### F.2.1 Fixed regularization

First, we provide an algorithm that has a *fixed* coefficient for entropy-regularization (Algorithm 9). Recall that $0 \le r_i \le R$ for some $R > 0$, for all $i \in \mathcal{N}$.

---

**Algorithm 9** MWU for zero-sum NGs with $\tau$-entropy regularization

---

Choose $K = \lceil 2R/\tau + \log(\max_{i \in \mathcal{N}} |\mathcal{A}_i|)) \rceil$
Choose $\pi_i^{(0)}$ as a uniform distribution for all $i \in \mathcal{N}$
**for** timestep $t = 0, 1, \dots$ **do**
  Define $\eta^{(t)} = 1/(\tau(t + K))$
  Update the policy as $\pi_i^{(t+1)}(a_i) \propto (\pi_i^{(t)}(a_i))^{1-\eta^{(t)}\tau} \exp\left(\eta^{(t)}[r_i\pi^{(t)}]_{a_i}\right)$ for all $i \in \mathcal{N}$ and $a_i \in \mathcal{A}_i$
**end for**

---

**Claim 12.** Define $\Omega_i = \left\{ \pi_i \in \Delta(\mathcal{A}_i) \mid \pi_i(a_i) \ge \frac{1}{|\mathcal{A}_i|} \exp\left(-\frac{R}{\tau}\right) \text{ for all } a_i \in \mathcal{A}_i \right\}$ and $g_i^{(t)} = r_i\pi^{(t)} - \tau \log \pi_i^{(t)}$ in Algorithm 9. Then, $\pi_i^{(t+1)} = \operatorname{argmax}_{\pi_i \in \Omega_i} \left( \pi_i^\mathsf{T} g_i^{(t)} - \frac{1}{\eta^{(t)}} \mathrm{KL}(\pi_i, \pi_i^{(t)}) \right)$ holds for all $i \in \mathcal{N}$ and $t \ge 0$.

*Proof.* The equation $\pi_i^{(t+1)}(a_i) \propto (\pi_i^{(t)}(a_i))^{1-\eta^{(t)}\tau} \exp\left(\eta^{(t)}[r_i\pi^{(t)}]_{a_i}\right)$ implies that

$$\pi_i^{(t+1)} = \operatorname*{argmax}_{\pi_i \in \Delta(\mathcal{A}_i)} \left( \pi_i^\mathsf{T} g_i^{(t)} - \frac{1}{\eta^{(t)}} \mathrm{KL}(\pi_i, \pi_i^{(t)}) \right)$$

by a simple algebra. The remaining part to establish is that $\pi_i^{(t+1)} \in \Omega_i$ holds true for all $t \ge 0$ and for every $i \in \mathcal{N}$. To prove the remaining part, we use induction. As $\pi_i^{(0)}$ is chosen to be a uniform

distribution, it is clear that $\pi_i^{(0)} \in \Omega_i$. Under the assumption that $\pi_i^{(t)} \in \Omega_i$, we have

$$\pi_i^{(t+1)}(a_i) = \frac{(\pi_i^{(t)}(a_i))^{1-\eta^{(t)}\tau} \exp\left([\eta^{(t)}\boldsymbol{r}_i\pi^{(t)}]_{a_i}\right)}{\sum_{a_i'\in\mathcal{A}_i}(\pi_i^{(t)}(a_i'))^{1-\eta^{(t)}\tau} \exp\left([\eta^{(t)}\boldsymbol{r}_i\pi^{(t)}]_{a_i'}\right)}$$

$$\underset{(i)}{\geq} \frac{\left(\frac{1}{|\mathcal{A}_i|}\exp\left(-\frac{R}{\tau}\right)\right)^{1-\eta^{(t)}\tau}}{\sum_{a_i'\in\mathcal{A}_i}(\pi_i^{(t)}(a_i'))^{1-\eta^{(t)}\tau} \exp\left([\eta^{(t)}\boldsymbol{r}_i\pi^{(t)}]_{a_i'}\right)}$$

$$\underset{(ii)}{\geq} \frac{\left(\frac{1}{|\mathcal{A}_i|}\exp\left(-\frac{R}{\tau}\right)\right)^{1-\eta^{(t)}\tau}}{\exp(\eta^{(t)}R)\sum_{a_i'\in\mathcal{A}_i}(\pi_i^{(t)}(a_i'))^{1-\eta^{(t)}\tau}} \underset{(iii)}{\geq} \frac{\left(\frac{1}{|\mathcal{A}_i|}\exp\left(-\frac{R}{\tau}\right)\right)^{1-\eta^{(t)}\tau}}{\exp(\eta^{(t)}R)|\mathcal{A}_i|^{\eta^{(t)}\tau}}$$

$$= \frac{1}{|\mathcal{A}_i|}\exp\left(-\frac{R}{\tau}\right)$$

thereby concluding the proof of our claim. In the above, $(i)$ is derived from $\exp\left([\eta^{(t)}\boldsymbol{r}_i\pi^{(t)}]_{a_i}\right) \geq 1$ and the induction hypothesis; $(ii)$ is the result of $\exp\left([\eta^{(t)}\boldsymbol{r}_i\pi^{(t)}]_{a_i}\right) \leq \exp(\eta^{(t)}R)$. Finally, $(iii)$ comes from $\sum_{a_i\in\mathcal{A}_i}(\pi_i^{(t)}(a_i))^{1-\eta^{(t)}\tau} \leq |\mathcal{A}_i|\left(\frac{1}{|\mathcal{A}_i|}\sum_{a_i\in\mathcal{A}_i}\pi_i^{(t)}(a_i)\right)^{1-\eta^{(t)}\tau}$, by Jensen's inequality. $\square$

In the forthcoming analysis of Algorithm 9 and Algorithm 10, our first step bounds $\mathrm{KL}(\pi_\tau^\star, \pi)$. Following this, we employ Proposition 11 below to bound the term $\mathsf{Matrix\text{-}QRE\text{-}Gap}_\tau$. Finally, we leverage Lemma 2 to bound the $\mathsf{Matrix\text{-}NE\text{-}Gap}$.

**Proposition 11.** For any $\tau > 0$, $\pi$, and $\boldsymbol{r}$, the following holds:

$$\mathsf{Matrix\text{-}QRE\text{-}Gap}_\tau(\boldsymbol{r}, \pi) \leq \mathcal{O}\left(\left(\tau\max_{i\in\mathcal{N}}\log|\mathcal{A}_i| + R\right)\sqrt{\mathrm{KL}(\pi_\tau^\star, \pi)}\right).$$

*Proof.* For any $\pi$, $\tau$, and $\boldsymbol{r}$, we have

$\mathsf{Matrix\text{-}QRE\text{-}Gap}_\tau(\boldsymbol{r}, \pi)$

$$= \max_{i\in\mathcal{N}}\left[\max_{\pi_i'\in\Delta(\mathcal{A}_i)}\left(\sum_{j\in\mathcal{E}_{r,i}}\pi_i'r_{i,j}\pi_j + \tau\mathcal{H}(\pi_i')\right) - \left(\sum_{j\in\mathcal{E}_{r,i}}\pi_ir_{i,j}\pi_j + \tau\mathcal{H}(\pi_i)\right)\right]$$

$$= \max_{i\in\mathcal{N}}\left[\max_{\pi_i'\in\Delta(\mathcal{A}_i)}\left(\sum_{j\in\mathcal{E}_{r,i}}\pi_i'r_{i,j}\pi_{\tau,j}^\star + \tau\mathcal{H}(\pi_i') + \sum_{j\in\mathcal{E}_{r,i}}\pi_i'r_{i,j}\pi_j - \sum_{j\in\mathcal{E}_{r,i}}\pi_i'r_{i,j}\pi_{\tau,j}^\star\right)\right.$$
$$\left. - \left(\sum_{j\in\mathcal{E}_{r,i}}\pi_ir_{i,j}\pi_j + \tau\mathcal{H}(\pi_i)\right)\right]$$

$$\underset{(i)}{\leq} \max_{i\in\mathcal{N}}\left[\max_{\pi_i'\in\Delta(\mathcal{A}_i)}\left(\sum_{j\in\mathcal{E}_{r,i}}\pi_i'r_{i,j}\pi_{\tau,j}^\star + \tau\mathcal{H}(\pi_i')\right) + \sum_{j\in\mathcal{E}_{r,i}}R\left\|\pi_j - \pi_{\tau,j}^\star\right\|_1\right.$$
$$\left. - \left(\sum_{j\in\mathcal{E}_{r,i}}\pi_ir_{i,j}\pi_j + \tau\mathcal{H}(\pi_i)\right)\right]$$

$$\underset{(ii)}{\leq} \max_{i\in\mathcal{N}}\left[\max_{\pi_i'\in\Delta(\mathcal{A}_i)}\left(\sum_{j\in\mathcal{E}_{r,i}}\pi_i'r_{i,j}\pi_{\tau,j}^\star + \tau\mathcal{H}(\pi_i')\right) - \left(\sum_{j\in\mathcal{E}_{r,i}}\pi_{\tau,i}^\star r_{i,j}\pi_{\tau,j}^\star + \tau\mathcal{H}(\pi_i)\right)\right.$$
$$\left. + 2\sum_{j\in\mathcal{E}_{r,i}}R\left\|\pi_j - \pi_{\tau,j}^\star\right\|_1 + R\left\|\pi_i - \pi_{\tau,i}^\star\right\|_1\right]$$

$$\underset{(iii)}{\leq} \mathcal{O}\left(R\sqrt{\mathrm{KL}(\pi_\tau^\star, \pi)}\right)$$

$$+ \max_{i \in \mathcal{N}}\left[\max_{\pi_i' \in \Delta(\mathcal{A}_i)}\left(\sum_{j \in \mathcal{E}_{r,i}} \pi_i' r_{i,j} \pi_{\tau,j}^\star + \tau\mathcal{H}(\pi_i')\right) - \left(\sum_{j \in \mathcal{E}_{r,i}} \pi_{\tau,i}^\star r_{i,j} \pi_{\tau,j}^\star + \tau\mathcal{H}(\pi_i)\right)\right]$$

$$\underset{(iv)}{\leq} \mathcal{O}\left(R\sqrt{\mathrm{KL}(\pi_\tau^\star, \pi)}\right)$$

$$+ \max_{i \in \mathcal{N}}\left[\max_{\pi_i' \in \Delta(\mathcal{A}_i)}\left(\sum_{j \in \mathcal{E}_{r,i}} \pi_i' r_{i,j} \pi_{\tau,j}^\star + \tau\mathcal{H}(\pi_i')\right)\right.$$

$$\left. - \left(\sum_{j \in \mathcal{E}_{r,i}} \pi_{\tau,i}^\star r_{i,j} \pi_{\tau,j}^\star + \tau\mathcal{H}(\pi_{\tau,i}^\star)\right) + \sqrt{3}\tau\log|\mathcal{A}_i|\sqrt{\mathrm{KL}(\pi_{\tau,i}^\star, \pi_i)}\right]$$

$$\leq \mathcal{O}\left(\left(\tau\max_{i \in \mathcal{N}}\log|\mathcal{A}_i| + R\right)\sqrt{\mathrm{KL}(\pi_\tau^\star, \pi)}\right),$$

where $(i)$ and $(ii)$ are due to the definition of $R$. Meanwhile, $(iii)$ holds by the Pinsker inequality, and $(iv)$ holds by bounding the difference in Shannon entropy via KL divergence, [117, Theorem 2]. $\qquad\square$

**Lemma 3** ([83, 51]). For any zero-sum NG with $(\mathcal{G} = (\mathcal{N}, \mathcal{E}_r), \mathcal{A}, (r_{i,j})_{(i,j) \in \mathcal{E}_r})$, for any joint product policies $\mu, \nu$, the following holds:

$$\sum_{i \in \mathcal{N}}\left(r_i(\mu_i, \nu_{-i}) + r_i(\nu_i, \mu_{-i})\right) = 0.$$

By Lemma 3 and the definition of $r_{\tau,i}$ (Equation (25)), we can derive

$$\sum_{i \in \mathcal{N}}\left(r_{\tau,i}\big(\pi_i^{(t)}, \pi_{\tau,-i}^\star\big) + r_{\tau,i}\big(\pi_{\tau,i}^\star, \pi_{-i}^{(t)}\big)\right) = 0. \tag{37}$$

Analogous to the method used in Claim 12, we can show that the QRE, $\pi_\tau^\star$, belongs to $\prod_{i \in \mathcal{N}} \Omega_i$. Furthermore, we can bound $g_i^{(t)}(a_i)$ for all $i \in \mathcal{N}$ and $a_i \in \mathcal{A}_i$ by

$$g_i^{(t)}(a_i) \leq R - \tau\log\pi_i^{(t)}(a_i) \leq R - \tau\log\left(\frac{1}{|\mathcal{A}_i|}\exp\left(-\frac{R}{\tau}\right)\right)$$

$$= 2R + \tau\log(|\mathcal{A}_i|)$$

so that $\eta^{(t)}g_i^{(t)}(a_i) \leq 1$ for all $i \in \mathcal{N}$, $a_i \in \mathcal{A}_i$, and $t \geq 0$. Therefore, we can apply Lemma 4 below.

**Lemma 4** ([118], Theorem 2). For a convex set $\Omega \subseteq \Delta(\mathcal{A})$, $\eta\mu \preceq \mathbf{1} \in \mathbb{R}^{|\mathcal{A}|}$, and any $\pi \in \Omega$, define

$$\pi' = \underset{\widetilde{\pi} \in \Omega}{\arg\max}\left(\widetilde{\pi}^\mathsf{T}\mu - \frac{1}{\eta}\mathrm{KL}(\widetilde{\pi}, \pi)\right).$$

Then, for any $\nu \in \Omega$, the following holds:

$$(\nu - \pi)^\mathsf{T}g \leq \frac{\mathrm{KL}(\nu, \pi) - \mathrm{KL}(\nu, \pi')}{\eta} + \eta\sum_{a \in \mathcal{A}} \pi(a)g(a)^2.$$

Now, we state the theorem on the iteration complexity of Algorithm 9 to obtain an $\epsilon$-NE.

**Theorem 7.** The last iterate of Algorithm 9 requires no more than $\widetilde{\mathcal{O}}\left(nR^4/\epsilon^4\right)$ iterations to achieve an $\epsilon$-NE of $(\mathcal{G}, \mathcal{A}, (r_{i,j})_{(i,j) \in \mathcal{E}_r})$.

*Proof.* First, we bound the difference between $r_{\tau,i}(\pi^{\star}_{\tau,i}, \pi^{(t)}_{-i})$ and $r_{\tau,i}(\pi^{(t)})$ as follows:

$$r_{\tau,i}(\pi^{\star}_{\tau,i}, \pi^{(t)}_{-i}) - r_{\tau,i}(\pi^{(t)})$$

$$= \left(\pi^{\star}_{\tau,i} - \pi^{(t)}_i\right)^{\mathsf{T}} \boldsymbol{r}_i \pi^{(t)} + \tau \left(\sum_{a_i \in \mathcal{A}_i} \pi^{(t)}_i(a_i) \log \pi^{(t)}_i(a_i) - \sum_{a_i \in \mathcal{A}_i} \pi^{\star}_{\tau,i}(a_i) \log \pi^{\star}_{\tau,i}(a_i)\right)$$

$$= \left(\pi^{\star}_{\tau,i} - \pi^{(t)}_i\right)^{\mathsf{T}} g^{(t)}_i - \tau \mathrm{KL}\left(\pi^{\star}_{\tau,i}, \pi^{(t)}_i\right)$$

$$\leq \frac{\mathrm{KL}\left(\pi^{\star}_{\tau,i}, \pi^{(t)}_i\right) - \mathrm{KL}\left(\pi^{\star}_{\tau,i}, \pi^{(t+1)}_i\right)}{\eta^{(t)}} + \eta^{(t)} \sum_{a_i \in \mathcal{A}_i} \pi^{(t)}_i(a_i) \left(g^{(t)}_i(a_i)\right)^2 - \tau \mathrm{KL}\left(\pi^{\star}_{\tau,i}, \pi^{(t)}_i\right)$$

$$\leq \frac{\mathrm{KL}\left(\pi^{\star}_{\tau,i}, \pi^{(t)}_i\right) - \mathrm{KL}\left(\pi^{\star}_{\tau,i}, \pi^{(t+1)}_i\right)}{\eta^{(t)}} + \eta^{(t)} \left(2R + \tau \max_{i \in \mathcal{N}} \log(|\mathcal{A}_i|)\right)^2 - \tau \mathrm{KL}\left(\pi^{\star}_{\tau,i}, \pi^{(t)}_i\right).$$

Here, the penultimate inequality holds by Lemma 4 since $\pi_{\tau,i} \in \Omega_i$ and $\eta^{(t)} g^{(t)}_i(a_i) \leq 1$ holds for all $t > 0$ and $a_i \in \mathcal{A}_i$. Therefore, we have

$$\mathrm{KL}\left(\pi^{\star}_{\tau,i}, \pi^{(t+1)}_i\right)$$

$$\leq \left(1 - \eta^{(t)}\tau\right) \mathrm{KL}\left(\pi^{\star}_{\tau,i}, \pi^{(t)}_i\right) + \eta^{(t)}(r_{\tau,i}(\pi^{(t)}) - r_{\tau,i}(\pi^{\star}_{\tau,i}, \pi^{(t)}_{-i}))$$

$$+ (\eta^{(t)})^2 \left(2R + \tau \max_{i \in \mathcal{N}} \log(|\mathcal{A}_i|)\right)^2 \tag{38}$$

for all $i \in \mathcal{N}$ and $t \geq 0$. If we sum (38) over $i \in \mathcal{N}$, we have

$$\mathrm{KL}\left(\pi^{\star}_{\tau}, \pi^{(t+1)}\right)$$

$$\leq \left(1 - \eta^{(t)}\tau\right) \mathrm{KL}\left(\pi^{\star}_{\tau}, \pi^{(t)}\right) + n(\eta^{(t)})^2 \left(2R + \tau \max_{i \in \mathcal{N}} \log(|\mathcal{A}_i|)\right)^2$$

$$+ \sum_{i \in \mathcal{N}} \eta^{(t)}(r_{\tau,i}(\pi^{(t)}) - r_{\tau,i}(\pi^{\star}_{\tau,i}, \pi^{(t)}_{-i}))$$

$$= \left(1 - \eta^{(t)}\tau\right) \mathrm{KL}\left(\pi^{\star}_{\tau}, \pi^{(t)}\right) + n(\eta^{(t)})^2 \left(2R + \tau \max_{i \in \mathcal{N}} \log(|\mathcal{A}_i|)\right)^2$$

$$+ \sum_{i \in \mathcal{N}} \eta^{(t)}(r_{\tau,i}(\pi^{(t)}_i, \pi^{\star}_{\tau,-i}) - r_{\tau,i}(\pi^{\star}_{\tau})) \tag{39}$$

$$\leq \left(1 - \eta^{(t)}\tau\right) \mathrm{KL}\left(\pi^{\star}_{\tau}, \pi^{(t)}\right) + n(\eta^{(t)})^2 \left(2R + \tau \max_{i \in \mathcal{N}} \log(|\mathcal{A}_i|)\right)^2 \tag{40}$$

for all $t \geq 0$ where (39) holds due to (37) and $\sum_{i \in \mathcal{N}} r_{\tau,i}(\pi^{(t)}) = \sum_{i \in \mathcal{N}} r_{\tau,i}(\pi^{\star}_{\tau}) = 0$, and (40) holds due to that $\pi^{\star}_{\tau}$ is the NE for the game that having the payoff $r_{\tau,i}$. If we recursively apply (40),

we have

$$\mathrm{KL}\left(\pi_\tau^\star, \pi^{(t+1)}\right)$$

$$\leq \prod_{l=0}^{t}(1-\eta^{(l)}\tau)\mathrm{KL}(\pi_\tau^\star,\pi^{(0)}) + n\left(2R+\tau\max_{i\in\mathcal{N}}\log(|\mathcal{A}_i|)\right)^2 \sum_{l=0}^{t}(\eta^{(l)})^2 \prod_{s=l+1}^{t}(1-\eta^{(s)}\tau)$$

$$= \prod_{l=0}^{t}(1-1/(l+K))\mathrm{KL}(\pi_\tau^\star,\pi^{(0)})$$

$$+ n\left(2R+\tau\max_{i\in\mathcal{N}}\log(|\mathcal{A}_i|)\right)^2 \sum_{l=0}^{t}\frac{1}{\tau^2}(1/(l+K)^2)\prod_{s=l+1}^{t}(1-1/(s+K))$$

$$\leq K/(K+t)\mathrm{KL}(\pi_\tau^\star,\pi^{(0)})$$

$$+ n\left(2R+\tau\max_{i\in\mathcal{N}}\log(|\mathcal{A}_i|)\right)^2 \frac{1}{\tau^2}\log\left((t+K+1)/(K)\right)(1/(t+K))$$

$$= \widetilde{\mathcal{O}}\left(\left(K\sum_{i\in\mathcal{N}}\log(|\mathcal{A}_i|) + n\max\left(R^2/\tau^2, \max_{i\in\mathcal{N}}\log^2(|\mathcal{A}_i|)\right)\right)/t\right)$$

$$= \widetilde{\mathcal{O}}\left(n\max\left(R^2/\tau^2, \max_{i\in\mathcal{N}}\log^2(|\mathcal{A}_i|)\right)/t\right).$$

Therefore, if we iterate Algorithm 9 for $T$ times, by Proposition 11 and Lemma 2, we obtain an

$$\widetilde{\mathcal{O}}\left(\tau\max_{i\in\mathcal{N}}\log|\mathcal{A}_i| + \left(\tau\max_{i\in\mathcal{N}}\log|\mathcal{A}_i| + R\right)\sqrt{n\max\left(R^2/\tau^2, \max_{i\in\mathcal{N}}\log^2(|\mathcal{A}_i|)\right)/T}\right)$$

approximate NE. Therefore, if we want to obtain an $\epsilon$-NE ($\epsilon > 0$) for the matrix game in the last iterate, we need to have $\widetilde{\mathcal{O}}\left(nR^4/\epsilon^4\right)$ iteration. $\qquad\square$

### F.2.2 Diminishing regularization

One might also consider the algorithm with a diminishing choice of $\tau$. We provide Algorithm 10 for this case.

---

**Algorithm 10** MWU for zero-sum NGs with diminishing entropy regularization

---

Choose $K = (R + 2\max_{i\in\mathcal{N}}\log|\mathcal{A}_i|)^2$
Choose $\pi_i^{(0)}$ as a uniform distribution for all $i \in \mathcal{N}$
**for** timestep $t = 0, 1, \dots$ **do**
    Define $\tau^{(t)} = (t+K)^{-1/6}$ and $\eta^{(t)} = (t+K)^{-1/2}$
    Update $g_i^{(t)} = \boldsymbol{r}_i\pi^{(t)} - \tau^{(t)}\log\pi_i^{(t)}$ for all $i \in \mathcal{N}$
    Define $\Omega_i^{(t)} = \left\{\pi_i \in \Delta(\mathcal{A}_i) \mid \pi_i(a_i) \geq \frac{1}{|\mathcal{A}_i|(t+K)^2} \text{ for all } a_i \in \mathcal{A}_i\right\}$
    Update the policy as $\pi_i^{(t+1)} = \mathrm{argmax}_{\pi_i\in\Omega_i^{(t)}}\left(\pi_i^\mathsf{T}g_i^{(t)} - \frac{1}{\eta^{(t)}}\mathrm{KL}(\pi_i, \pi_i^{(t)})\right)$ for all $i \in \mathcal{N}$
**end for**

---

Let $\pi_{\tau^{(t)}}^\star$ be the unique NE in the policy space $\prod_{i\in\mathcal{N}}\Omega_i^{(t)}$ for the game that has the payoff $r_{\tau^{(t)},i}$ as

$$r_{\tau^{(t)},i}(\pi) = r_i(\pi) + \tau^{(t)}\mathcal{H}(\pi_i) - \sum_{j\in\mathcal{E}_{r,i}}\frac{\tau^{(t)}}{|\mathcal{E}_{r,j}|}\mathcal{H}(\pi_j) = \pi_i^\mathsf{T}\boldsymbol{r}_i\pi + \tau^{(t)}\mathcal{H}(\pi_i) - \sum_{j\in\mathcal{E}_{r,i}}\frac{\tau^{(t)}}{|\mathcal{E}_{r,j}|}\mathcal{H}(\pi_j).$$

Moreover, we can bound $g_i^{(t)}(a_i)$ for all $i \in \mathcal{N}$ and $a_i \in \mathcal{A}_i$:

$$g_i^{(t)}(a_i) \leq R - (t+K)^{-1/6}\log\pi_i^{(t)}(a_i) \leq R + 2(t+K)^{-1/6}\log(|\mathcal{A}_i|(t+K))$$
$$\leq R + 2\max_{i\in\mathcal{N}}\log(|\mathcal{A}_i|),$$

so that $\eta^{(t)}g_i^{(t)}(a_i) \leq 1$ for all $i \in \mathcal{N}$, $a_i \in \mathcal{A}_i$, and $t \geq 0$. Therefore, we can apply Lemma 4.

**Theorem 8.** The last iterate of Algorithm 10 requires no more than $\widetilde{\mathcal{O}}\left(n^3 R^{12}/\epsilon^6\right)$ iterations to achieve an $\epsilon$-NE of $(\mathcal{G}, \mathcal{A}, (r_{i,j})_{(i,j)\in\mathcal{E}_r})$.

*Proof.* First, we bound the difference between $r_{\tau^{(t)},i}(\pi^\star_{\tau^{(t)},i}, \pi^{(t)}_{-i})$ and $r_{\tau^{(t)},i}(\pi^{(t)})$ as follows:

$$r_{\tau^{(t)},i}(\pi^\star_{\tau^{(t)},i}, \pi^{(t)}_{-i}) - r_{\tau^{(t)},i}(\pi^{(t)})$$

$$= \left(\pi^\star_{\tau^{(t)},i} - \pi^{(t)}_i\right)^\mathsf{T} r_i \pi^{(t)} + \tau^{(t)}\left(\sum_{a_i\in\mathcal{A}_i} \pi^{(t)}_i(a_i)\log\pi^{(t)}_i(a_i) - \sum_{a_i\in\mathcal{A}_i}\pi^\star_{\tau^{(t)},i}(a_i)\log\pi^\star_{\tau^{(t)},i}(a_i)\right)$$

$$= \left(\pi^\star_{\tau^{(t)},i} - \pi^{(t)}_i\right)^\mathsf{T} g^{(t)}_i - \tau^{(t)}\mathrm{KL}\left(\pi^\star_{\tau^{(t)},i}, \pi^{(t)}_i\right)$$

$$\leq \frac{\mathrm{KL}\left(\pi^\star_{\tau^{(t)},i}, \pi^{(t)}_i\right) - \mathrm{KL}\left(\pi^\star_{\tau^{(t)},i}, \pi^{(t+1)}_i\right)}{\eta^{(t)}}$$

$$+ \eta^{(t)}\sum_{a_i\in\mathcal{A}_i}\pi^{(t)}_i(a_i)\left(g^{(t)}_i(a_i)\right)^2 - \tau^{(t)}\mathrm{KL}\left(\pi^\star_{\tau^{(t)},i}, \pi^{(t)}_i\right)$$

$$\leq \frac{\mathrm{KL}\left(\pi^\star_{\tau^{(t)},i}, \pi^{(t)}_i\right) - \mathrm{KL}\left(\pi^\star_{\tau^{(t)},i}, \pi^{(t+1)}_i\right)}{\eta^{(t)}}$$

$$+ \eta^{(t)}\left(R + 2(t+K)^{-1/6}\max_{i\in\mathcal{N}}\log(|\mathcal{A}_i|(t+K))\right)^2 - \tau^{(t)}\mathrm{KL}\left(\pi^\star_{\tau^{(t)},i}, \pi^{(t)}_i\right).$$

Here, the penultimate inequality holds by Lemma 4 since $\pi_{\tau^{(t)},i}\in\Omega^{(t)}_i$ and $\eta^{(t)}g^{(t)}_i(a_i)\leq 1$ holds for every $t > 0$ and $a_i\in\mathcal{A}_i$. Therefore, we have

$$\mathrm{KL}\left(\pi^\star_{\tau^{(t+1)},i}, \pi^{(t+1)}_i\right)$$

$$\leq \left(1 - \eta^{(t)}\tau^{(t)}\right)\mathrm{KL}\left(\pi^\star_{\tau^{(t)},i}, \pi^{(t)}_i\right) + \eta^{(t)}(r_{\tau^{(t)},i}(\pi^{(t)}) - r_{\tau^{(t)},i}(\pi^\star_{\tau^{(t)},i}, \pi^{(t)}_{-i})) \qquad (41)$$

$$+ (\eta^{(t)})^2\left(R + 2(t+K)^{-1/6}\max_{i\in\mathcal{N}}\log(|\mathcal{A}_i|(t+K))\right)^2$$

$$+ \mathrm{KL}\left(\pi^\star_{\tau^{(t+1)},i}, \pi^{(t+1)}_i\right) - \mathrm{KL}\left(\pi^\star_{\tau^{(t)},i}, \pi^{(t+1)}_i\right)$$

for all $i\in\mathcal{N}$ and $t\geq 0$. If we sum (41) over $i\in\mathcal{N}$, we have

$$\mathrm{KL}\left(\pi^\star_{\tau^{(t+1)}}, \pi^{(t+1)}\right)$$

$$\leq \left(1 - \eta^{(t)}\tau^{(t)}\right)\mathrm{KL}\left(\pi^\star_{\tau^{(t)}}, \pi^{(t)}\right) + n(\eta^{(t)})^2\left(R + 2(t+K)^{-1/6}\max_{i\in\mathcal{N}}\log(|\mathcal{A}_i|(t+K))\right)^2$$

$$+ \sum_{i\in\mathcal{N}}\eta^{(t)}(r_{\tau^{(t)},i}(\pi^{(t)}) - r_{\tau^{(t)},i}(\pi^\star_{\tau^{(t)},i}, \pi^{(t)}_{-i})) + \mathrm{KL}\left(\pi^\star_{\tau^{(t+1)}}, \pi^{(t+1)}\right) - \mathrm{KL}\left(\pi^\star_{\tau^{(t)}}, \pi^{(t+1)}\right)$$

$$= \left(1 - \eta^{(t)}\tau^{(t)}\right)\mathrm{KL}\left(\pi^\star_{\tau^{(t)}}, \pi^{(t)}\right) + n(\eta^{(t)})^2\left(R + 2(t+K)^{-1/6}\max_{i\in\mathcal{N}}\log(|\mathcal{A}_i|(t+K))\right)^2$$

$$\qquad (42)$$

$$+ \sum_{i\in\mathcal{N}}\eta^{(t)}(r_{\tau^{(t)},i}(\pi^{(t)}_i, \pi^\star_{\tau^{(t)},-i}) - r_{\tau^{(t)},i}(\pi^\star_{\tau^{(t)}})) + \mathrm{KL}\left(\pi^\star_{\tau^{(t+1)}}, \pi^{(t+1)}\right) - \mathrm{KL}\left(\pi^\star_{\tau^{(t)}}, \pi^{(t+1)}\right)$$

$$\leq \left(1 - \eta^{(t)}\tau^{(t)}\right)\mathrm{KL}\left(\pi^\star_{\tau^{(t)}}, \pi^{(t)}\right) + n(\eta^{(t)})^2\left(R + 2(t+K)^{-1/6}\max_{i\in\mathcal{N}}\log(|\mathcal{A}_i|(t+K))\right)^2$$

$$+ \mathrm{KL}\left(\pi^\star_{\tau^{(t+1)}}, \pi^{(t+1)}\right) - \mathrm{KL}\left(\pi^\star_{\tau^{(t)}}, \pi^{(t+1)}\right), \qquad (43)$$

for all $t \geq 0$ where (42) holds due to (37), and (43) holds due to that $\pi_\tau^\star$ is the NE for the game with the payoff $r_{\tau^{(t)},i}$. If we recursively apply (43), we have

$$\mathrm{KL}\left(\pi_{\tau^{(t+1)}}^\star, \pi^{(t+1)}\right)$$

$$\leq \prod_{l=0}^{t}(1 - \eta^{(l)}\tau^{(l)})\mathrm{KL}(\pi_\tau^\star, \pi^{(0)})$$

$$\quad + n\left(R + 2(t+K)^{-1/6}\max_{i \in \mathcal{N}}\log(|\mathcal{A}_i|(t+K))\right)^2 \sum_{l=0}^{t}(\eta^{(l)})^2 \prod_{s=l+1}^{t}(1 - \eta^{(s)}\tau^{(s)})$$

$$\quad + \sum_{l=0}^{t}\prod_{s=l+1}^{t}(1 - \eta^{(s)}\tau^{(s)})\left(\mathrm{KL}\left(\pi_{\tau^{(t+1)}}^\star, \pi^{(t+1)}\right) - \mathrm{KL}\left(\pi_{\tau^{(t)}}^\star, \pi^{(t+1)}\right)\right)$$

$$\leq \prod_{l=0}^{t}(1 - (t+K)^{-2/3})\mathrm{KL}(\pi_\tau^\star, \pi^{(0)})$$

$$\quad + n\left(R + 2(t+K)^{-1/6}\max_{i \in \mathcal{N}}\log(|\mathcal{A}_i|(t+K))\right)^2 \sum_{l=0}^{t}(1/(l+K)) \prod_{s=l+1}^{t}(1 - (s+K)^{-2/3})$$

$$\quad + \sum_{l=0}^{t}\prod_{s=l+1}^{t}(1 - (s+K)^{-2/3})\left(\mathrm{KL}\left(\pi_{\tau^{(t+1)}}^\star, \pi^{(t+1)}\right) - \mathrm{KL}\left(\pi_{\tau^{(t)}}^\star, \pi^{(t+1)}\right)\right)$$

$$\underset{(i)}{\leq} \widetilde{\mathcal{O}}\left(n\max(R^2, \max_{i \in \mathcal{N}}\log(|\mathcal{A}_i|t)t^{-1/3})(t+K)^{-1/3} + \max_{i \in \mathcal{N}}\log^3(|\mathcal{A}_i|t)(t+K)^{-1/3}\right)$$

$$= \widetilde{\mathcal{O}}\left(nR^2t^{-1/3}\right).$$

Here, (i) holds because

$$\prod_{l=0}^{t}(1 - (t+K)^{-2/3})\mathrm{KL}(\pi_\tau^\star, \pi^{(0)}) = \widetilde{\mathcal{O}}\left(\exp\left(-t^{1/3}\right)\right)$$

holds,

$$\sum_{l=0}^{t}(1/(l+K)) \prod_{s=l+1}^{t}(1 - (s+K)^{-2/3}) = \widetilde{\mathcal{O}}((t+K)^{-1/3})$$

holds by [119, Lemma 4], and

$$\sum_{l=0}^{t}\prod_{s=l+1}^{t}(1 - (s+K)^{-2/3})\left(\mathrm{KL}\left(\pi_{\tau^{(t+1)}}^\star, \pi^{(t+1)}\right) - \mathrm{KL}\left(\pi_{\tau^{(t)}}^\star, \pi^{(t+1)}\right)\right)$$

$$\leq \max_{i \in \mathcal{N}}\log^3(|\mathcal{A}_i|(t+K))(t+K)^{-1/3}$$

holds by [119, Lemma 4, Lemma 15]. Therefore, if we iterate Algorithm 10 for $T$ times, by Proposition 11 and Lemma 2, we obtain

$$\widetilde{\mathcal{O}}\left(T^{-1/6}\max_{i \in \mathcal{N}}\log|\mathcal{A}_i| + \left(T^{-1/6}\max_{i \in \mathcal{N}}\log|\mathcal{A}_i| + R\right)\sqrt{nR^2T^{-1/3}}\right)$$

approximate NE. Therefore, if we want to obtain $\epsilon$-NE ($\epsilon > 0$) for the matrix game, we need to have $\widetilde{\mathcal{O}}\left(n^3R^{12}/\epsilon^6\right)$ iterations. $\qquad\square$

## G    Experimental Results

We now present experimental results for the learning dynamics/algorithms investigated before.

### G.1 Fictitious-play property of zero-sum NMGs

We present an experiment for the fictitious-play property in Section 5. We experimented with an infinite-horizon $\gamma$-discounted zero-sum NMG ($\mathcal{G} = (\mathcal{N}, \mathcal{E}_Q), \mathcal{S}, \mathcal{A}, \mathbb{P}, (r_i)_{i \in \mathcal{N}}, \gamma)$, where $\mathcal{N} = \{0, 1, 2\}$, $\mathcal{S} = \{0, 1\}$, $\mathcal{A}_i = \{0, 1\}$ for all $i \in \mathcal{N}$, $\mathbb{P}(s' \,|\, s, \boldsymbol{a}) := \mathbb{F}_0(s' \,|\, s, a_0)$, $\mathbb{F}_0(0 \,|\, 0, 1) = \mathbb{F}_0(0 \,|\, 1, 0) = 0.8$, $\mathbb{F}_0(0 \,|\, 1, 1) = \mathbb{F}_0(0 \,|\, 0, 0) = 0.2$, $r_{0,1}(0) = -r_{1,0}(0)^\intercal = \begin{bmatrix} 1 & 2 \\ 4 & 3 \end{bmatrix}$, $r_{0,2}(0) = -r_{2,0}(0)^\intercal = \begin{bmatrix} 4 & 3 \\ 2 & 1 \end{bmatrix}$, $r_{0,1}(1) = -r_{1,0}(1)^\intercal = \begin{bmatrix} 4 & 3 \\ 2 & 1 \end{bmatrix}$, $r_{0,2}(1) = -r_{2,0}(1)^\intercal = \begin{bmatrix} 1 & 2 \\ 4 & 3 \end{bmatrix}$, $\alpha_t = \frac{1}{t^{0.55}}$, $\beta_t = \frac{1}{t^{0.75}}$, and $\gamma = 0.99$. We iterated $2^{28}$ times for the experiments. The result is demonstrated in Figure 3 (a). Note that the gray and black lines indicate the sum of the values for states 0 and 1, which asymptotically go to 0.

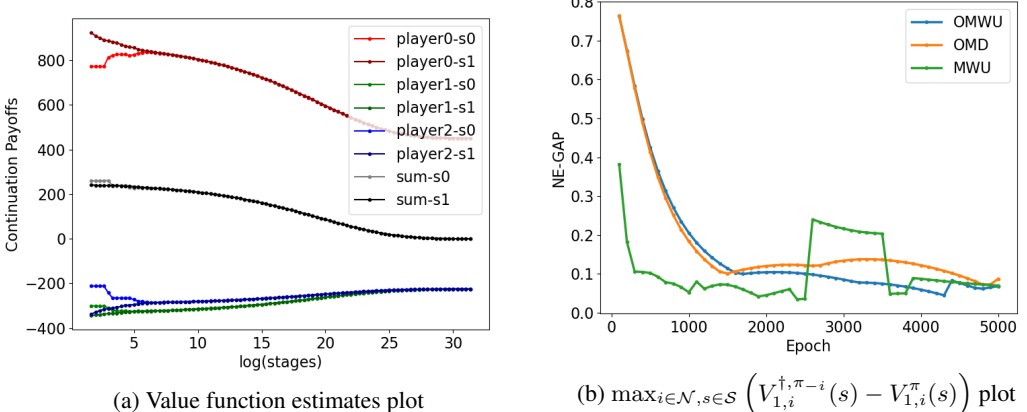

(a) Value function estimates plot      (b) $\max_{i \in \mathcal{N}, s \in \mathcal{S}} \left( V_{1,i}^{\dagger, \pi_{-i}}(s) - V_{1,i}^{\pi}(s) \right)$ plot

Figure 3: (a) Fictitious play experiment. The red and dark red lines indicate player 0's value function estimates for states 0 and 1, respectively. The green and dark green lines indicate player 1's value function estimates for states 0 and 1, respectively. The blue and dark blue lines indicate player 2's value function estimates for states 0 and 1, respectively. The gray and black lines indicate the sum of each player's value function estimates for states 0 and 1, respectively. $x$-axis denotes the logarithm with base 2 of the number of iterates (stages) and $y$-axis denotes the value function estimates. (b) Value-iteration-based algorithms with (OMWU, OMD, MWU) NE-ORACLE subroutines. The blue, orange, and green lines indicate $\max_{i \in \mathcal{N}, s \in \mathcal{S}} \left( \max_{\pi'_i \in \Delta(\mathcal{A}_i)} V_{1,i}^{\pi'_i, \pi_{-i}}(s) - V_{1,i}^{\pi}(s) \right)$ value of the OMWU, OMD, and MWU NE-ORACLE, respectively. $x$-axis denotes the number of iteration of NE-ORACLE subroutine, and $y$-axis denotes the NE-Gap.

### G.2 Value-iteration with different NE-ORACLEs

We present an experiment for zero-sum NMGs with different NE-ORACLEs in Section 6. We experimented with a zero-sum NMG ($\mathcal{G} = (\mathcal{N}, \mathcal{E}_Q), \mathcal{S}, \mathcal{A}, H, (\mathbb{P}_h)_{h \in [H]}, (r_{h,i})_{h \in [H], i \in \mathcal{N}})$ where $\mathcal{N} = \{0, 1, 2\}$, $\mathcal{E}_Q = \{(1, 2), (1, 0), (2, 0)\}$, $\mathcal{S} = \{0, 1\}$, $\mathcal{A}_i = \{0, 1\}$ for all $i \in \mathcal{N}$, $H = 5$, $\mathbb{P}_h(s' \,|\, s, \boldsymbol{a}) := \sum_{i \in \mathcal{N}} \mathbb{F}_i(s' \,|\, s, a_i)$, $\mathbb{F}_0(s' \,|\, s, a_0) = 0$, $\mathbb{F}_1(s' \,|\, s, a_1) = \frac{1}{3}\mathbb{P}_1(s' \,|\, s, a_1)$, $\mathbb{F}_2(s' \,|\, s, a_2) = \frac{2}{3}\mathbb{P}_2(s' \,|\, s, a_2)$, $\mathbb{P}_1(0 \,|\, 0, 1) = \mathbb{P}_1(0 \,|\, 1, 0) = \mathbb{P}_2(0 \,|\, 0, 1) = \mathbb{P}_2(0 \,|\, 1, 0)$ is determined randomly, $\mathbb{P}_1(0 \,|\, 1, 1) = \mathbb{P}_1(0 \,|\, 0, 0) = \mathbb{P}_2(0 \,|\, 1, 1) = \mathbb{P}_2(0 \,|\, 0, 0)$ is determined randomly, and $r_{h,i}$ is determined randomly such that it makes $(\mathcal{G}, \mathcal{A}, (r_{h,i}(s))_{i \in \mathcal{N}})$ a zero-sum NG for every $h$. We set $\tau = 0.05$ for both OMWU and MWU. We set $\eta = 1/(36H)$ for both OMWU and OMD. We iterated the algorithm for $T = 5000$ times. The result is plotted in Figure 3 (b).

# H Limitations and Broader Impact

**Limitations and future directions.** One main focus of our paper is on introducing the new model of multi-player zero-sum Markov games, and studying the performance of basic learning and (centralized) computation algorithms. We did not intend to develop more advanced decentralized multi-agent RL algorithms with finite-sample/iteration guarantees, which would be one of our immediate next steps. Our convergence guarantees for fictitious-play dynamics in the infinite-horizon setting only hold for the star-shaped structure, and it would be interesting to explore other network structures that possess the fictitious-play property, and how they mirror the conditions for the computational hardness result.

**Broader impact.** Our work mainly focuses on the theoretical aspects of multi-player Markov games with networked local interactions. We expect no negative social impacts from our results.