# OpenReview forum: "Multi-Player Zero-Sum Markov Games with Networked Separable Interactions"
_NeurIPS.cc/2023/Conference — NeurIPS 2023 poster_

### Official Review · Reviewer_AdnB · 2023-06-21

**Soundness:** 3 good
**Presentation:** 2 fair
**Contribution:** 2 fair
**Rating:** 4
**Confidence:** 4

**Summary:**

The paper considers a subclass of Markov Games (MG) that have local interactions and a zero-sum structure for their Q value functions. This subclass generalizes zero-sum polymatrix games. It's shown that the marginal distribution of a Markov CCE for these games is a Markov  NE. It's then shown that computing a stationary Markov CCE/NE is PPAD-hard. It's then shown that a fictitious play type algorithm converges to a stationary Markov CCE for the special case where the local interactions are given by a star graph where the player at the center of the star controls the state transitions. It's then shown that another algorithm based on optimistic multiplicative weight updates converges to a non-stationary Markov

**Strengths:**

Markov game is a challenging class of games with limited results. The paper makes some progress for this class of games. The analysis is rigorous and non-trivial. The technical aspects of the results are well-discussed.

**Weaknesses:**

The motivation for this class of games is unclear and weak. The motivation for the results of this class of games is also vague. The paper is dry and technical with little insight. The presentation is notationally heavy and some of the statements are a bit sloppy.

I'm not sure what meaning the proposed class of games has beyond its technical definition and the paper does very little to expose such meaning. The definition is done on the Q values, which is awkward and unintuitive, and then proposition 1 basically states that a better definition exists. In a polymatrix zero-sum game, the decomposition of the reward makes simple sense. To generalize them to MG, the paper here adds that the transition probabilities need to be decomposable - which doesn't seem nice or intuitive anymore beyond trivial special cases (one player controls the transitions, please see Q3 below). The paper presents one example (even though the title is "Examples of M(Z)NMGs") of such a game which is highly contrived, and even for this example, the strong bipartite graph structure condition is needed. This is very unconvincing regarding how useful this class of games is.

The results for stationary equilibrium are a bit disappointing. The hardness result would be more interesting if it was already agreed that this class of games is interesting - but otherwise, it just looks like a reason not to consider this class of games proposed here. A star structure with the central player controlling the transitions basically means that the N-1 different games are almost separate (up to the mutual state). No motivation is provided for wanting to compute a stationary equilibrium, so these disappointing results are uncalled for. If nothing meaningful can be said there, and there is no reason to believe this case is of special importance, I don't see why the stationary case is any interesting.

The statements of the definition and proposition are mostly messy. They are written as one block of text even if they are split into several cases, and if some equations are more important than others. Therefore they are unnecessarily hard to digest.

More Comments:

The acronyms are very confusing because they never follow the first letters of the words.

In Definition 1, Multi-player MG is italic but "with networked local interactions" isn't, making it look like only the italic part is the issue here.

"every stage game with continuation payoffs being the payoff matrices qualifies as an" - continuation payoffs?

Proposition 1 (b) - by always, you mean for every discount factor? this is a bit confusing as stated.

The definition of N_c can use some words to explain the meaning. This is also true for other definitions in the paper.

The caption of Figure 1 doesn't explain what's going on (not even that the red and blue are the nodes in the set N_c). Also, shouldn't the arrow by an equality?

"based on the PCP for PPAD conjecture" - I guess you mean, if the conjecture holds, then... otherwise nothing can be based on a conjecture

Line 165: if we do not know a priori about  - no "about" needed

What's the point of the title of Subsection 5.1? that's the only case and only subsection

Line 367 - double period

The statement of Theorem 2 is lacking and misleading. The main assumption is the star structure (and a single player driving the state transitions), and it's not stated here.



**Questions:**

Q1: Most of the paper analyzes the computation of stationary Markov CCE/NE.  Why is this interesting that stationary equilibrium is hard to compute if a non-stationary one isn't? Why do we want to compute the equilibrium in the first place? The research question proposed in Lines 36-37 is already answered by the non-stationary part of the paper.

Q2: Why does \mathcal{E}_r appear in a  MZNMG? (e.g., Figure 1 t comes from a polymatrix zero-sum game, but a MZNMG is not defined using a polymatrix zero-sum game. Why is another graph even needed? shouldn't the reward dependencies be dictated by \mathcal{E}_Q? why would these two graphs be different?

Q3: Are there interesting cases beyond a single (state-dependent) player driving the state transitions? Given that it was done in [50], I think that a detailed comparison with [50] is needed.

**Limitations:**

The technical limitations in this paper are discussed. There is no potential negative societal impact since the paper is computational.

---

> ### Author Rebuttal · Authors · 2023-08-08
>
> Thank you for finding our analysis rigorous and non-trivial, and the technical parts well-discussed. We acknowledge your main comments regarding *motivations*, and sincerely hope our responses below could help you appreciate the significance of the class of games and equilibrium we consider.
>
> **Motivation of the game model and comparing to [50] (Q3)).** As we stated in the introduction, the study of this class of model mainly came from two important motivations in “game theory” and “learning in games”: 1) the emerging behavior of natural *learning dynamics*, e.g., fictitious-play (FP); 2) the *computation tractability* of equilibrium. We push the boundaries in both of these two aspects. Before our work, the only known types of “Markov games (MG)” where FP can reach the equilibrium are either zero-sum [1-3], or identical-interest [4-5]. It was unknown if there are other MGs in which fictitious-play dynamics would reach equilibrium in the long-run. Similarly, before our work, regarding equilibrium computation complexity we know that: a) computing Markov NE (both stationary and non-stationary) is tractable (in polynomial time) for zero-sum and potential MGs; b) computing non-stationary Markov CCE is tractable for general-sum MGs; c) computing stationary Markov CCE is intractable for general-sum MGs [6]. It was unknown if there exist other types of MGs where (Markov) NE maybe *tractable*, and/or Markov stationary CCE maybe *tractable*. The motivation of our work follows exactly from the seminal works of [7-10] (push the boundary on *what types of games have fictitious-play property*), and that of [11-12] (push the boundary on *what types of games may have computationally tractable algorithms for finding the NE*), in the normal-form game setting. We believe these are strong motivations.
>
> For Definition 1, we have established the structural results in Prop. 1, regarding the if and only if condition for an MG to be qualified as an MZNMG, in terms of its reward and transition dynamics. Essentially, what our model requires is the transition dynamics being an “ensemble” of some transition dynamics that is controlled by each controller agent individually. We emphasize that this “ensemble” dynamics is more than just “single-controller” case in [50], i.e., “one player controls the transitions”, which covers “switched/turn-based” case (though even the single-controller case is an important class of MGs in its own right, see [50]’s Ch 3 and 6, and is definitely not a “trivial” one).
>
> Note that all our examples (*note that we have more than just “one” example, as we stated at the beginning of Sec. 3.2*), which generalize polymatrix matrix games to the Markov setting, satisfy this structure naturally. Hence, we respectfully disagree that the class of games we introduced is vacuous.
>
> Finally, and perhaps encouragingly, we remark that “after” submission of our paper to NeurIPS, a new work [14] was posted on arxiv, which also concerns the generalization of zero-sum polymatrix games to the MG setting. This independent work carries exactly the same motivation as ours. Furthermore, we note that [14] “defines” the zero-sum polymatrix MGs as the ones where: 1) the reward is zero-sum polymatrix, and 2) the transition dynamics is ”switched-controller/turn-based”. This model aligns with and is covered by ours. We hope both of our independent works may *strengthen the motivation of considering this type of MG in the community*.
>
>
> **The motivation for studying stationary equilibrium and Q1).** For infinite-horizon discounted MGs, “stationary” equilibrium is a very natural solution concept to consider. In fact, we believe it is more natural than the non-stationary one, which mainly started to attract attention in the recent literature of *multi-agent RL*, which exclusively focused on the “finite-horizon episodic” setting for ease of exposition and studying the “exploration” problem in RL. Specifically, starting from Shapley’s seminal work [15] that introduced this MG model, to the classical MG literature in economics [16-20], and to the classical literature of multi-agent RL in MGs [21-22], they all focused on studying “stationary” equilibrium in this setting. It is natural since in “single-agent MDPs”, such a policy class is the most widely studied one in infinite-horizon settings, and it should be important to ask about the computational tractability for this class of solutions in the multi-agent game setting.
>
> Finally, “stationary” equilibrium and its stark contrast to “non-stationary” equilibrium in terms of computation complexity have been a central topic in several recent breakthrough results [23-24]. It was shown that “stationary” equilibrium can be computationally intractable, while the “non-stationary” counterpart can be computed using backward induction. This is particularly interesting since it is in stark contrast to the single-agent setting, where there is no such difference. Hence, we believe studying stationary equilibrium in MGs is a well-motivated topic. We believe studying the computation of this type of equilibria can help us paint a more complete picture of equilibrium computation for this new class of MGs, and better understand the fundamental limit of this model.
>
> **Q2 Answer).** We intended to keep $E_r$ in Figure 1, in order to demonstrate the fact that the graph that characterizes the separable property of the “reward” can be different from the graph that characterizes the separable property of Q-functions. We will make sure to introduce this notation property and explain it more explicitly.
>
> We will also correct the minor grammar issues/typos and presentation issues.
>
> Thank you very much again for the insightful comments for helping improve our paper. Please let us know if there are any other comments we can address to help you improve the rating of our paper.
>
> [14] "Zero-sum Polymatrix Markov Games: Equilibrium Collapse and Efficient Computation of Nash Equilibria." arXiv:2305.14329.

---

> > ### Author Response · Authors · 2023-08-18
> > **Thank you for your comment**
> >
> > Thank you very much for your additional comments. First, we sincerely apologize if our previous response has caused any confusion -- we did not mean to "lecture" about equilibria and stationary equilibria. Our response exactly came from your question in the previous round: "why we want to compute an equilibrium ..., and then perhaps it will become clear why certain types of equilibria are more desirable than others", without referring to any literature on Markov games directly since they might be "subjective", as the reviewer has pointed out. We intended to justify these equilibria in a general sense, i.e., it is some solution concept that might be of interest to all (Markov) games (including the ones we consider, as a special case).
> >
> > Your new comments are well-taken, and we will make sure to address them by providing more explanations in the next version of the paper. We just would like to add the following points briefly:
> >
> > 1. As you also mentioned, "in light of the hardness of Markov games, simplifying assumptions are made (like focusing on a certain type of equilibria", we are exactly doing so by considering "Markov equilibria", as the seminal work [1] and many other works we mentioned did. We believe this is an important first step to studying any type of Markov games. We don't necessarily see "studying certain fundamental and commonly accepted equilibria" as an "assumption". That said, justifying *other kinds of equilibria* in this model is definitely an interesting and exciting next step.
> >
> > 2. As we mentioned before, we never intended (nor claimed in paper or rebuttal) that "stationary" equilibria are "better" in some sense (i.e., "more natural", "more desirable"), and favor it over other types of equilibria. We just believe this is **one** important kind of solution concept one may not want to miss in Markov games (starting from [1]), and that was exactly why we established results for **both stationary and non-stationary** solutions.
> >
> > 3. Compared with [22,40], we believe our "analysis" does not "follow the existing ideas". Specifically, [22] established the PPAD-hardness for general-sum stochastic games, while our analysis focused on constructing the "reduction" to the types of games studied in [22]; [40] studied the "equilibrium collapse" for "matrix" polymatrix games, using linear programming (LP) argument, while we did not use LP to show the results for the "Markov" case, but uses dynamic programming. In fact, interestingly, the recent arxiv work [2] after our submission exactly stated that the LP idea in [40] does not work in the Markov case. Thus, we believe it might be fairer to say our results "build upon" these, but with completely different technical ideas/novelties, instead of "following the existing ideas".
> >
> > 4. Thanks for the feedback on the presentation and writing, and we will make sure to improve them in the next version.
> >
> > Thank you again for your valuable feedback.
> >
> > [1] Lloyd S Shapley. Stochastic games. Proceedings of the national academy of sciences, 449 39(10):1095–1100, 1953.
> >
> > [2] Fivos Kalogiannis and Ioannis Panageas. "Zero-sum Polymatrix Markov Games: Equilibrium Collapse and Efficient Computation of Nash Equilibria." arXiv preprint arXiv:2305.14329 (2023).

---

> ### Author Response · Authors · 2023-08-10
> **references for rebuttal**
>
> [1] David S Leslie, Steven Perkins, and Zibo Xu. Best-response dynamics in zero-sum stochastic games. Journal of Economic Theory, 189:105095, 2020.
>
> [2] Muhammed Sayin, Kaiqing Zhang, David Leslie, Tamer Basar, and Asuman Ozdaglar. Decentralized Q-learning in zero-sum Markov games. In NeurIPS, 2021.
>
> [3] Muhammed O Sayin, Francesca Parise, and Asuman Ozdaglar. Fictitious play in zero-sum stochastic games. SIAM Journal on Control and Optimization, 60(4):2095–2114, 2022.
>
> [4] Muhammed O Sayin, Kaiqing Zhang, and Asuman Ozdaglar. Fictitious play in markov games with single controller. In ACM Conference on Economics and Computation (EC), pages 919–936, 2022.
>
> [5] Lucas Baudin and Rida Laraki. Fictitious play and best-response dynamics in identical interest and zero-sum stochastic games. In ICML, 2022.
>
> [6] Constantinos Daskalakis, Noah Golowich, and Kaiqing Zhang. The complexity of Markov equilibrium in stochastic games. In COLT, 2023.
>
> [7] Koichi Miyasawa. On the convergence of the learning process in a 2x2 non-zero-sum game. Economic Research Program, Princeton University, Research Memorandum, 33, 1961.
>
> [8] Dov Monderer and Lloyd S Shapley. Fictitious play property for games with identical interests. Games and Economic Behavior, 68:258–265, 1996.
>
> [9] Aner Sela. Fictitious play in “one-against-all” multi-player games. Economic Theory, 14:635–651, 1999.
>
> [10] Ulrich Berger. Fictitious play in 2xn games. Journal of Economic Theory, 120(2):139–154, 2005.
>
> [11] Yang Cai and Constantinos Daskalakis. On minmax theorems for multiplayer games. In SODA, 2011.
>
> [12] Yang Cai, Ozan Candogan, Constantinos Daskalakis, and Christos Papadimitriou. Zero-sum polymatrix games: A generalization of minmax. Mathematics of Operations Research, 41(2):648–655, 2016.
>
> [13] Jerzy Filar and Koos Vrieze. Competitive Markov decision processes. Springer Science & Business Media, 2012.
>
> [14] Fivos Kalogiannis and Ioannis Panageas. "Zero-sum Polymatrix Markov Games: Equilibrium Collapse and Efficient Computation of
> Nash Equilibria." arXiv preprint arXiv:2305.14329 (2023).
>
> [15] Lloyd S Shapley. Stochastic games. Proceedings of the national academy of sciences, 449 39(10):1095–1100, 1953.
>
> [16] Arlington M. Fink. Equilibrium in a stochastic n-person game. Journal of Science of the Hiroshima University, series A-I (mathematics), 28(1):89–93, 1964.
>
> [17] Eilon Solan and Nicolas Vieille. Stochastic games. Proceedings of the National Academy of Sciences (PNAS), 112(45):13743–13746, 2015.
>
> [18] Eric Maskin and Jean Tirole. A theory of dynamic oligopoly, I: Overview and quantity competition with large fixed costs. Econometrica: Journal of the Econometric Society, pages 549–569, 1988.
>
> [19] Eric Maskin and Jean Tirole. A theory of dynamic oligopoly, II: Price competition, kinked demand curves, and edgeworth cycles. Econometrica: Journal of the Econometric Society, pages 571–599, 1988.
>
> [20] Masayuki Takahashi. Equilibrium points of stochastic non-cooperative n-person games. Journal of Science Hiroshima University Series A-I, 28:95–99, 1964.
>
> [21] Michael L. Littman. Markov games as a framework for multi-agent reinforcement learning. Machine Learning Proceedings 1994, pages 157–163. Elsevier, 1994.
>
> [22] Junling Hu, and Michael P. Wellman. "Nash Q-learning for general-sum stochastic games." Journal of machine learning research 4, no. Nov (2003): 1039-1069.
>
> [23] Constantinos Daskalakis, Noah Golowich, and Kaiqing Zhang. The complexity of markov 473 equilibrium in stochastic games. In COLT, 2023.
>
> [24] Yujia Jin, Vidya Muthukumar, and Aaron Sidford. The complexity of infinite-horizon general-sum stochastic games. In ITCS, 2022.

---

> > ### Comment · Reviewer_AdnB · 2023-08-16
> > **Response to Rebuttal**
> >
> > Thank you for your response.
> >
> > First, I apologize for missing the other examples in the appendix. They are indeed helpful to justify the studied class of games. I'm therefore raising my score to 4. I still find the proposed class of games limited, as the examples demonstrate. Even though I understand that the limitations are a result of the intractability of more general Markov games, to make a paper interesting, it needs to consider an interesting class of games.
> >
> > My concern regarding the motivation for a stationary equilibrium remains. The response frequently uses the existence of other papers as motivation for this paper. I find this highly subjective and suboptimal. It would be better if the questions studied in this paper could be motivated independently using objective arguments. For example, saying that a stationary equilibrium is "more natural" and is motivated because other papers studied it is not quite ideal, to put it mildly. The appropriate way is to first clearly motivate why we want to compute an equilibrium (to predict the outcome of the game? or to instruct the players to play it? etc), and then perhaps it will become clear why certain types of equilibria are more desirable than others.

---

> > > ### Author Response · Authors · 2023-08-16
> > > **Reply (2)**
> > >
> > > **Why stationary equilibria**:
> > >
> > > First, we would like to emphasize that, we never claimed (in our paper, or in rebuttal) that *stationary equilibrium* is “more desirable” than “non-stationary equilibrium”. We believe both are natural and important classes of equilibria and that was exactly why we **established results for both** in the paper.
> > >
> > > Second, we believe stationary equilibrium has the following properties that make it worth being studied:
> > >
> > > 1. **Easy to describe and understand**: Stationary equilibria, in which the policies do not change over time, are easier to describe and understand. In contrast, non-stationary equilibria can involve complex time-dependent strategies and require the space that grows with time to describe.
> > >
> > > 2. **Memory efficiency**: Related to point 1, stationary equilibria typically require fewer memory resources, as agents do not need to constantly update their strategies. Non-stationary equilibria, with time-varying strategies, can be memory intensive, as it has equilibria that are time-varying.
> > >
> > > 3. **Ease of Implementation**: Stationary policies are easier to implement in practice, as agents do not need to adapt their strategies over time. Non-stationary policies may require complex adaptation mechanisms, making their implementation more challenging.
> > >
> > > 4. **Fixed point computation**: Stationary equilibrium, if can be computed, usually comes from solving a fixed-point equation, e.g., for the value-iteration operator. This gives us a standard routine to find a solution concept for Markov games, which facilitates the design of other learning algorithms, e.g., Q-learning.
> > >
> > > 5. **Fundamental difference from non-stationary equilibrium**: In the single-agent MDP setting, it is well-known that both stationary and non-stationary optimal policies can be found efficiently, for infinite-horizon discounted settings. In stark contrast, given by the recent results [2,3], in the multi-agent Markov game setting, computing *stationary* equilibrium (even CCE) is **computationally intractable**, while computing *non-stationary* equilibrium is still tractable (using backward induction). We believe such a contrast is very interesting and illustrates the fundamental challenges in **multi-agent** sequential decision-making. We wanted to understand this difference in our setting also.
> > >
> > > Combined with all the classical literature we have referred to, in conclusion, we believe stationary equilibria is a reasonable solution concept to consider in Markov/stochastic games. We will try to add these emphases in the final version of our paper. Please feel free to let us know if there are any other questions that may affect your evaluation of our paper, we are more than happy to address them. Thank you very much.
> > >
> > >
> > > [1] H. Peyton Young, Strategic learning and its limits, OUP Oxford, 2004.
> > >
> > > [2] Constantinos Daskalakis, Noah Golowich, and Kaiqing Zhang. The complexity of Markov equilibrium in stochastic games. In COLT, 2023.
> > >
> > > [3] Yujia Jin, Vidya Muthukumar, and Aaron Sidford. The complexity of infinite-horizon general-sum stochastic games. In ITCS, 2022.

---

> > > > ### Comment · Reviewer_AdnB · 2023-08-18
> > > > **Another Response**
> > > >
> > > > Thank you for the illuminating lecture about equilibria and stationary equilibria. I think that the motivation in the paper aligns poorly with the ideas you describe here as I will try to explain below. I also think that in a rebuttal phase, it's more productive to stick with details that are relevant to the paper at hand.
> > > >
> > > > Would you say that an approximate Markov stationary NE/CCE is a good predictor? why would players converge to that? it asks them to use Markov strategies (why would selfish players ignore the past?), it's approximate (note that the epsilon does not reveal the *Euclidean* distance between the equilibrium to the exact one), and it's not even the perfect version, so given some state realization, why wouldn't selfish players deviate?
> > > >
> > > > The same applies to designing incentives. Mechanism design needs to guarantee players cannot manipulate the mechanism. The equilibria you're studying, as I explained above, are inherently easily manipulated, definitely in a dynamic game where players can use the history of the game.
> > > >
> > > > (I see "Analyzing Social Phenomena" as a special case of predicting behavior.)
> > > >
> > > > I disagree with "For Better Decision-Making" and "Facilitating Negotiation". Equilibria are typically inefficient from a global point of view. If you mean that analyzing them can predict outcomes and that the prediction helps external decision-making, then it's again a special case of "Predicting Behavior" which I discussed above. The suitable framework for "Facilitating Negotiation" seems to be cooperative game theory (e.g., coalitions and bargaining) which has nothing to do with this paper.
> > > >
> > > > The memory efficiency and implementation complexity of stationary equilibria can be only slightly better than some non-stationary equilibria that only switch infrequently, or in a periodic behavior with a short period (e.g., odd and even turns). Your result doesn't quantity the amount of "non-stationarity" needed for an equilibrium to be computable so these issues are irrelevant.
> > > >
> > > > The fact that stationary equilibria may be computed using fixed point iterations, or that in general they're easier to describe and understand, is also irrelevant to "predicting behavior" since the players themselves converge to equilibria in other means and are not looking to describe and understand their equilibrium. I would even guess that players are more likely to adopt local dynamic programming (backward induction) to compute their strategies, which, as you mentioned, can easily lead to non-stationary strategies.
> > > >
> > > > I now read the examples in the appendix more carefully. They seem a little motivated and contrived, more or less like the example in the body of the paper.
> > > >
> > > > I do think that there is significant technical content in this paper. But in light of the existing literature (e.g., [22] and [40]). the technical effort is mainly incremental and tries to reconstruct these results for this specific class of game using similar techniques (of the level of generality between [40] and [22]). Here I'm not making claims about the difficulty of doing so, but about the originality involved. It is also understandable that in light of the hardness of Markov games, simplifying assumptions are made (like focusing on a certain type of equilibria, even if they're not great predictions), as is indeed done in other papers. But different papers can still utilize similar strong assumptions and come up with results that vary in their level of interest and depth. With strong assumptions, it's extra important that the results would be exciting and interesting in at least one strong sense. The class of games studied here is very limited and specific, as the examples serve to show, and the analysis follows existing ideas. Together with the messy presentation and writing, especially around the technical details, I do not believe the paper is ready for publication.

---

> ### Author Response · Authors · 2023-08-16
> **Reply (1)**
>
> Thank you for your response, and for starting to appreciate the motivation we study this *class of games*. We're excited to push the limits of our understanding in the realms of "learning in games" and "equilibrium tractability," both of which are cornerstone topics in (algorithmic) game theory. If you have any further questions about this part, please don't hesitate to reach out.
>
>
> Regarding **why we study stationary equilibrium**:
>
>
> First, we want to clarify that our justification isn't solely based on references, but have also mentioned in the previous rebuttal that “It is natural since in “single-agent MDPs”, such a policy class is the most widely studied one in infinite-horizon settings, and it should be important to ask about the computational tractability for this class of solutions in the multi-agent game setting.” “It was shown that “stationary” equilibrium can be computationally intractable, while the “non-stationary” counterpart can be computed using backward induction. This is particularly interesting since it is in stark contrast to the single-agent setting, where there is no such difference.”
>
>
> Second, regarding the comment on subjectivity, we believe that referring to a robust body of *classical literature* is an *objective* measure. The importance of this solution concept has been consistently acknowledged by the community (not by us since these are not necessarily our works).
>
>
> We understand your additional comments, and respond as below:
>
> **Why equilibria**:
>
> Computing equilibria in (Markov) games is essential in real-world applications, with the following examples:
>
> 1. **Predicting Behavior**: In financial markets, understanding how multiple traders interact and influence prices is crucial. By computing equilibria, one can predict traders' strategies, anticipate market movements, and make informed investment decisions.
>
> 2. **For Better Decision-Making**: In healthcare, multiple stakeholders (patients, providers, payers) interact with varying objectives. Computing equilibria helps understand these interactions, supporting better decision-making in treatment, insurance, and healthcare policies.
>
> 3. **Designing Incentives and Regulations**: In environmental conservation, agents (countries, corporations) have different interests regarding resource usage and pollution. Equilibria can inform the design of incentives and regulations that promote sustainable practices while considering agents' objectives. Finding the equilibrium so that agents with misaligned objectives may achieve certain social objectives is exactly the main theme in mechanism design, a core topic in game theory.
>
> 4. **Analyzing Social Phenomena**: In social sciences, understanding how individuals interact and influence each other is crucial. Equilibria provides a framework to analyze social phenomena like opinion dynamics, information diffusion, and collective behavior. In fact, in Economics, the issue of how market prices and demands come into equilibrium is a long-standing problem [1].
>
> 5. **Facilitating Negotiation**: In international relations, countries negotiate on issues like trade, security, and diplomacy. Equilibria help identify mutually acceptable agreements, facilitating successful negotiations that balance the interests of all parties involved.
>
> The significance of finding *equilibrium* in (algorithmic) game theory, dated back to the time of Cournot and Nash, is definitely more than the bullets we mentioned above.

---

### Official Review · Reviewer_xjXv · 2023-06-26

**Soundness:** 3 good
**Presentation:** 2 fair
**Contribution:** 2 fair
**Rating:** 6
**Confidence:** 3

**Summary:**

The paper studies the problem of learning equilibria in multi-player zero-sum Markov games with networked local interactions (abbreviated as MZNMGs for short). In such games, the set of Markov CCEs collapses into that of Markov NEs. The paper shows that finding approximate Markov stationary CCEs is PPAD-hard in MZNMGs with infinite horizon, unless the network structure has a specific star shape. Then, the paper proves that in games with such a structure fictitious-play-based dynamics are guaranteed to converge to a Markov stationary NE. Moreover, the paper also shows that the negative PPAD-hardness result can be circumvented by considering Markov non-stationary CCEs, providing finite-iteration convergence guarantees for an algorithm based on optimistic multiplication weight updates.

**Strengths:**

ORIGINALITY

The paper proposes a new class of Markov games and it shows that such a class of games enjoys some appealing properties in terms of convergence of learning dynamics.

QUALITY

The claims seem sound (but I did not check the proofs in details).

CLARITY

The paper is well written.

SIGNIFICENCE

The problem of learning equilibria in Markov games has receiving considerable attention over the last years, and the results presented in the paper could be of interest to both researchers in algorithmic game theory and those in multi-agent RL.

**Weaknesses:**

ORIGINALITY

The ideas behind the presented results and the techniques used to prove them seem adaptations and combinations of already known results and techniques. The authors should more specifically address the novelty of their results/techniques.

CLARITY

The notation used in the paper is quite cumbersome, and in many spots it is easy to get lost in symbols. I think that these issues are inherent in the model studied in the paper, since it encompasses many elements, but I encourage the authors to try to lighten the notation as much as possible.

MINOR COMMENTS

- The experimental results seem an unneeded extra in this paper, I would consider removing them (actually, they are only discussed in the abstract and in the last section of the Appendix).
- The discussion on related works in neglecting the recent works addressing the problem of learning correlated equilibria in sequential games. These are related to Markov games and should be adequately discussed.
- It is not clear what you mena by single controller.
- Line 107: It should be $\Delta(\mathcal{A})$ rather than $\Delta(\mathcal{A}_i)$.
- Line 122 and onward, the notation $\mu_i, \pi_{-I}$ has not been introduced.
- Line 162, it is not clear why $I \neq j$.
- Please give more textual intuition to Propositions 1 and 2.
- Lines 260-261, discuss more why Markovianity is important, it is not clear otherwise.

**Questions:**

1) What is the relation between the results in this paper and the following recent work?

Foster, Dylan J., Noah Golowich, and Sham M. Kakade. "Hardness of Independent Learning and Sparse Equilibrium Computation in Markov Games." arXiv preprint arXiv:2303.12287 (2023).

**Limitations:**

None.

---

> ### Author Rebuttal · Authors · 2023-08-08
>
> Thank you for finding our paper sound, well-written and significant. We respond to your reviews point-by-point as follows.
>
> **Originality.** Thank you for your comment. We agree that some of the ideas and techniques are based on existing works. However, non-trivial ideas and techniques are proposed in the paper, which include but are not limited to: 1). The if and only if conditions of transition and rewards, for a Markov game to be qualified as a MZNMG (Prop .1); 2). Reduction of our MZNMG model to general-sum MGs, and the proof for computation hardness; 3). Identifying the structures, i.e., the star-shaped ones, under which equilibrium computation and FPP are possible; 4). Reformulating the star-shaped MZNMG value-iteration operator as a “minimax” form to enable the convergence of fictitious-play; 5). New algorithm and proofs for non-stationary NE computation in MZNMGs. We will emphasize more on these points in our updated version.
>
> We also emphasize that the techniques of single-agent MDP theory or two-player zero-sum MG theory are not directly applicable to every theorem's proof due to the multi-agent structure. Moreover, we'll expand Section 6, on the convergence results of not only OMWU but also MWU (which is not analyzed in the previous literature) and various algorithms in our updated version of the paper.
>
>
> **Clarity.** Thanks for pointing out that the notation is “cumbersome”, and thanks for understanding that the complication “inherently came from the complication of the model” we studied. We will try our best to lighten the notation in the updated version.
>
> **Minor comments.**
> 1. Experiments: Thanks for the suggestion. We thought adding some experiments to validate our theory would be a plus to a theory-oriented paper. We can add back more details in the main paper when we have more space in the final version.
> 2. Related work: Thank you very much for bringing up the recent works on learning CE in sequential/extensive games. We will surely include them in our updated version.
> 3. Notation typos: Thanks for the careful reading. We will clean up the typos and make them clear.
> 4. More intuition: Thanks for the suggestion, we will add more explanations regarding the intuition of proving Prop. 1 and 2. Regarding the importance of Markovianity, as our proof of Prop. 3 is based on backward induction over states, and it is the Markovian policies that preserve the polymatrix structure while inducing in a backward fashion. We will add more details on this also.
>
> **Relationship to Foster et al.** Thanks for bringing up this recent work. This work also established PPAD-hardness for computation, but for having “no-regret” algorithms in general Markov games. It has also provided “statistical” hardness results (even regardless of the computation tractability) for no-regret learning in Markov games without knowing the model. Note that the setting in Foster et al is “finite-horizon”, and the equilibrium they end up finding is “non-stationary”. The authors showed that even in this case, achieving “no-regret” is hard. In contrast, our hardness result is for “computation” only, and it is for Markov stationary CCE-computation. Note that to compute CCE, one may use “no-regret” learning algorithms, but it is not necessarily always the case – one may use algorithms that are not necessarily no-regret to obtain CCE, e.g., Daskalakis et al. and our paper. “No-regret” (in the face of arbitrary adversaries) is a more challenging goal to achieve (than (CC)equilibrium computation).
>
> Finally, we remark, also as stated in Foster et al. Sec. 6.4, that the two results are *not comparable* – stationary Markov CCE as we considered are not “sparse” (see their Definition 3.1), which is key to their hardness result; while they showed the hardness without imposing the “stationarity” condition (as they focused on the “finite-horizon” setting). We will add this discussion in our updated version.
>
> Thank you very much again for the insightful comments for helping improve our paper. We hope the responses have addressed them satisfactorily. Please let us know if there are any other comments we can address to help you improve the rating of our paper.

---

> ### Author Response · Authors · 2023-08-18
> **Response to Reviewer xjXv**
>
> Dear Reviewer xjXv,
>
> Thank you very much again for reviewing our paper. Since we have not heard back from you, we were wondering if the response has addressed your questions, and if there are any other comments we may need to address. Please do not hesitate to let us know.
>
> Thank you,
>
> The Authors

---

> > ### Comment · Reviewer_xjXv · 2023-08-20
> >
> > I would like to thank the Authors for their detailed response to my questions and concerns. Given this, and after reading all the other reviews, I will increase my score accordingly.

---

### Official Review · Reviewer_J94i · 2023-07-07

**Soundness:** 3 good
**Presentation:** 4 excellent
**Contribution:** 3 good
**Rating:** 7
**Confidence:** 4

**Summary:**

The paper focuses on the problem of computing both in a centralized and in a decentralized way equilibria for a special class of stochastic games. These are games that may be found in multiple states. At each state players have a set of available actions, that once played will define both the reward that each player achieves and the next state in which the game will be found. It is known that with only two players or multi-player potential games equilibria of these games can be efficiently computed by fictitious play. The paper consider multi-player non-potential games such that the utility that a player achieves by playing an action can be decomposed as the sum of the utilities that this action will give in two-players games against a few of other players. Indeed, it is known that these games in their normal form (non stochastic non repeated) enjoy a lot of common properties with the two-player zero sum games, such as the possibility to efficiently compute an equilibirum.

Unfortunately, the answer of the paper is essentially negative: computing an equilibrium for this class of stochastic games is hard even for coarse correlated equilibrium, unless the utility relationships is such that there is a center player c such that the utility of all remaining players depends on c only. In this case, fictitious play is known to converge to the equilibrium under opportune assumptions on the rate at which the relevant learning value are updated. Finally, it is showed that is possible to compute through multiplicative weights updates a dynamics converging to an approximation of a non-stationary equilibrium, i.e., an equilibrium that is not a fixed point independent of history.

**Strengths:**

The problem attracted the interest of the community in recent years, and hence it is relevant. Moreover it is a well-motivated class of problems, both for their applications and for the theoretical foundations (as stated above in the summary).

The contribution is clearly stated, and the paper is quite easy to read.

**Weaknesses:**

The results are essentially negative and positive cases have quite limited applicztion.
E.g., Prop 3 (stating correlated equilibria are the same as Nash equilibria) only holds for decomposable  transition matrix, that in turn mean that there must be at least a single player that depends on all other player, that is often not the case (usually local interactions are very local).
Thm 1 allows equilibrium computation only for a further restriction of this class. And convergence of fictitious play holds under further assumptions.

**Questions:**

Does decomposable tranisition matrix mean that N_c >= 1? If so, I believe that would make the definition more easy to grasp to write down this fact.

Remove double point at the end of page 8.

**Limitations:**

See above

---

> ### Author Rebuttal · Authors · 2023-08-08
>
> Thank you for finding our results well-motivated, contributions clearly stated, and the paper easy to follow. We respond to your reviews point-by-point as follows.
>
> **The results are essentially negative and positive cases have limited applications.** Regarding Prop. 3, first, the result applies to all MZNMGs as long as it is not a “two-player” case (which is a reduced case, as it is just a standard two-player Markov game), since by our Prop. 1, an MZNMG with “more than two-players” should have the decomposability of the transition dynamics. Second, a decomposable transition dynamics does not necessarily mean “there must be at least a single player that depends on all other players”, as we emphasized in our paper that it can have an ensemble structure. Moreover, In Prop. 1, we allowed the set $\mathcal{N}_c$ to be empty. Sorry for the confusion of the statement, and we have updated them in the manuscript in the supplementary material. These conditions might sound restrictive, but by our Prop. 1, they are the “necessary” conditions for an MG to be an MZNMG.
>
> Finally, we remark that *after* submitting our paper, an independent work [1] was posted on arxiv, which also concerns zero-sum Markov polymatrix games. Notably, the “definition” of the game therein that enables “equilibrium collapse” and “equilibrium computation” is based on the “switched/turn-based” controller transition dynamics, which is covered by our “ensemble/decomposable” dynamics, according to our remarks in lines 191-205 at the end of page 4.
>
> **Conditions that enable fictitious-play property.** Our “star-shaped” condition that enables fictitious-play is inspired by our hardness result in Theorem 1, as a graph that does not contain a triangle or 3-path subgraph has to be a star shape (see our Prop. 5). Without such a condition, it is unclear how to construct a “contracting” value-iteration operator, which is essential in the current analysis of the fictitious play in MZNMGs. We discussed the importance of the condition in the remark at the end of page 8.
>
> **Does decomposable transition dynamics mean $\mathcal{N}_c\geq1$?** Sorry for the confusion. When saying “decomposable”, we actually allowed N_c=0, which corresponds to the other cases discussed in Prop. 1. Being aware of the confusion, we have cleaned up and updated the statement of Prop 1 in the manuscript in the supplementary material (and colored the changes). We will also keep the changes in the updated version.
>
> **Remove the double-point at the end of page 8.** We will do it in the updated version.
>
> Thank you very much again for the helpful comments on improving our paper. We hope the responses have addressed them satisfactorily. Please let us know if there are any other comments we can address to help you reevaluate our paper.
>
>
> References:
>
> [1] Fivos Kalogiannis and Ioannis Panageas. "Zero-sum Polymatrix Markov Games: Equilibrium Collapse and Efficient Computation of Nash Equilibria." arXiv preprint arXiv:2305.14329 (2023).

---

> > ### Comment · Reviewer_J94i · 2023-08-14
> >
> > I see that I was misunderstanding the definition of decomposable transition matrix.
> >
> > By hoping that this aspect will be clarified in the final version, I am prone to increase my score.

---

### Official Review · Reviewer_GndM · 2023-07-10

**Soundness:** 3 good
**Presentation:** 3 good
**Contribution:** 3 good
**Rating:** 7
**Confidence:** 4

**Summary:**

The authors define a notion of Markov Games with local interactions with the extra assumption that are zero-sum (inspired from the polymatrix/graphical games that appeared in papers in Algorithmic Game Theory). They manage to show that Markovian Coarse Correlated Equilibria correspond to Markovian Nash Equilibria when marginalized (called the phenomenon collapsing) in their setting. They also show that approximating stationary and Markovian NE  is PPAD-hard and they finally provide an algorithm that gives Markovian non-stationary NE for their class of Markov Games.

**Strengths:**

The authors manage to define another class of Markov games in which finding NE (Markovian but not stationary) is tractable. The collapsing phenomenon is quite interesting and the idea of using QRE to get Markovian non-stationary NE is natural and elegant.

**Weaknesses:**

The model assumes some decomposability of the probability transition matrix that makes the model a bit complicated to parse. It seems though that the model captures interesting settings. The hardness result is not surprising and seems straightforward from Daskalakis et al.

**Questions:**

Can you elaborate more on the uniqueness of the QRE? If this is the case for infinite horizon settings, why your algorithm does not work for infinite horizon and you need to truncate after log(1/eps) episodes?

---

> ### Author Rebuttal · Authors · 2023-08-08
>
> Thank you for finding our equilibrium collapsing result interesting, and the algorithmic idea natural and elegant. We respond to your reviews point-by-point as follows.
>
> **Hardness result seems not surprising and straightforward from Daskalakis et al.** We agree that the result *relies on* Daskalakis et al, but respectfully disagree that it is straightforward from it. To show the result, we needed to carefully construct the reward functions for the players in MZNMGs, and identify the “sufficient” (and later turns out to be “necessary”) conditions for the Markov stationary CCE to be computationally intractable, i.e., the network structure is not a star shape. The necessary part is justified by our result on the contracting property of the VI operator (see the response to Reviewer SApy, and in lines 350-367 in the Remark on page 8). This “star-shaped” condition also inspired us to study the fictitious-play property under it, and justified the computation of “non-stationary” CCE later, separating what is possible and what is not in this class of games. We believe the result is important and not a trivial application of Daskalakis et al. 2022.
>
> Moreover, these results have important implications for the computational hardness literature. Consider the static polymatrix game as an example. Its fascinating aspect lies in the dichotomy between the computational **hardness** of computing the Nash Equilibrium (NE) of two-player general-sum static games, and the computational **tractability** of computing the NE of two-player zero-sum static games. This contrast underlines the significance of polymatrix games from a complexity perspective. In the realm of Markov games, our study contributes by exploring special cases of *multi-player zero-sum* games. This new class of games adds on a new dimension to the complexity spectrum, as we can draw interesting contrasts between these cases and their static-game counterparts. By bridging this gap, we hope to deepen the understanding of computational complexity results in the field of Markov games.
>
>
> **Uniqueness of QRE.** It is correct that in the **finite-horizon** setting, the QRE, which is non-stationary, is unique. This can be shown by backward induction – the QRE for the matrix game (i.e., the MZNG) at the last time step is unique, due to reference [62] in the paper; this fact can then be used to show the uniqueness of the previous timestep by induction. This is also the reason why we truncated after log(1/\epsilon) steps and compute “non-stationary” equilibrium in Sec. 6.
>
> In the **infinite-horizon** discounted setting, the QRE of the MZNMG is not necessarily unique (and *we never claimed so*). In general, it is unclear if the “QRE + value-iteration” operator is contracting, or possesses any good property that leads to the “uniqueness” of the fixed point (which corresponds to the QRE of the Markov game). This is also eluded by our hardness result – if the “QRE + value-iteration” operator is contracting, then an “approximate” stationary CCE can be obtained by iterating this operator and choosing a small enough regularization, contradicting our hardness result.
>
> Finally, we also remark that, the “uniqueness” of the QRE does not necessarily mean the existence of an easy way to “compute” the equilibrium, either. In particular, uniqueness may not necessarily come from the contracting property of the VI-operator, or some other properties that may lead to efficient computation algorithms. We will add these discussions in the updated version of the paper.
>
> Thank you very much again for the helpful and insightful comments on improving our paper. We hope the responses have addressed them satisfactorily.

---

> > ### Comment · Reviewer_GndM · 2023-08-13
> > **Thanks for the response**
> >
> > Thank you for your detailed response. I would like to increase my score to 7. Please include all the comments of QRE in your final version.

---

### Official Review · Reviewer_SApy · 2023-07-16

**Soundness:** 3 good
**Presentation:** 3 good
**Contribution:** 3 good
**Rating:** 6
**Confidence:** 4

**Summary:**

The paper studies multi-player zero-sum Markov games with networked interactions, generalizing the well-studied class of polymatrix normal-form games. They first show that Markov coarse correlated equilibria (CCE) collapse to Markov Nash equilibria (NE), allowing to focus on computing Markov CCE for which they provide polynomial-time algorithms based on optimistic multiplicative weights update. They also rely on recent advances to show that the stronger notion of stationary CCE is PPAD-hard, unless the network is star-shaped. For the latter class of Markov games, they establish that fictitious play converges asymptotically to the set of stationary NE.

**Strengths:**

The paper studies a natural subclass of Markov games inspired by a number of positive results on polymatrix zero-sum normal-form games. The authors motivate this class with a number of concrete examples and applications, and they make concrete contributions towards understanding the complexity of computing equilibria in that class of games. In particular, they show that certain positive results in normal-form games can be extended to their Markov games counterpart as well, eluding a number of hardness results in general Markov games. These are some of the few positive results for Markov NE under a non-trivial class of multi-player games, so I believe the results will be valued by the community.

The paper also appears sound; I did not find any notable issue in the proofs. The paper is also well-written and organized, and the key ideas are nicely exposed in the main body.

**Weaknesses:**

There are a couple of issues regarding the exposition/novelty of the results. First, the authors claim that they provide the first guarantees even in zero-sum polymatrix normal-form games under fictitious-play dynamics, but this does not appear to be the case. In particular, see the paper "Fictitious play in Networks." (Note that the latter paper also provides guarantees under discrete-time FP.) In fact, the results of the latter paper apply without the restrictive assumption of a star-shaped topology. So the authors should carefully clarify the connection with the aforementioned paper. I should note that this does not affect significantly the narrative of the paper since the results on the more general class of Markov games are novel and non-trivial even in light of that other paper. For completeness, it would be good to see if your techniques provide asymptotic guarantees for polymatrix zero-sum normal-form games without the star-shaped assumption.

The second issue relates to the novelty of the results in Section 6. In light of the collapse of Markov CCE with Markov NE, one can obtain immediately a polynomial-time algorithm using any algorithm for computing Markov CCE (e.g., the one by Daskalakis et al. 2022). In contrast, instead of using known algorithms, the authors design a new algorithm whose analysis is quite technically involved; it is not clear why the authors did not simply rely on known algorithms. At the very least, the authors should clarify that getting a Markov NE follows directly by the shown collapse; the current write-up can cause confusion. The designed algorithm based on OMWU has the nice property of getting a rate of $1/T$, which is certainly not the case with other algorithms for computing Markov CCE, but this is not highlighted at all. Regarding the last point, if one only cares about centralized algorithms, it should be possible to use the ellipsoid against hope algorithm of Papadimitriou and Roughgarden (along with backward induction) to get an exact Markov CCE (and hence Markov NE).



**Questions:**

Some minor issues:

1. The exposition in Section 3 regarding Markov games with network interactions appears to be more complex than what is actually needed. I would recommend slightly revising and simplifying the exposition in Section 3.1
2. Is there a polynomial algorithm for computing stationary NE under a star-shaped topology? The results based on FP do not give any non-asymptotic guarantees
3. A related paper worth inclusing: “Fast Convergence of Optimistic Gradient Ascent in Network Zero-Sum Extensive Form Games”
4. Propositions and theorems are typically in italic, but I leave this up to the authors
5. Line 373: approximates -> approximate
6. Line 668: correlated equilibria (CCE)
7. Line 718: do -> take
8. Line 719: gives -> which gives
9. Line 1001: It maybe good to point to Appendix E here for the definition of a differential inclusion as it is quite non standard


**Limitations:**

The authors have adequately addressed the limitations.

---

> ### Author Rebuttal · Authors · 2023-08-08
>
> **Comparison with prior work**. Thank you very much for bringing up the work “Fictitious play in networks” to our attention. We were not aware of the paper, and will add the reference in our updated version. We will also revise the statement regarding the first FP in zero-sum polymatrix games in the paper. Compared with the work, we allow more general stepsizes beyond the $1/T$ one. We also would like to point out that for the matrix case, we did *not* require the topology of the network to be of “star-shape” for the FPP to hold. The star-shaped condition is only required for the *Markov game* case.
>
> As pointed out by the reviewer, this does not change the narrative of our paper, as our focus was on the Markov game case, and viewed the one for the matrix case as a bonus result. Thank you again for kindly pointing out the reference.
>
>
> **Results in Sec. 6.** Indeed, given the equilibrium collapse result, one can use the existing algorithms for computing Markov CCE to find Markov NE, e.g., as in Daskalakis et al., 2022. We developed different algorithms as we believe they could constitute new contributions to the work. We wanted to include more solid technical results than “just applying” existing algorithms. We also wanted to have some algorithm that is natural and not completely “centralized”. We will emphasize this point to avoid confusion.
>
> A final remark is that, the algorithm in Daskalakis et al., 2022 is specifically designed for “learning” Markov CCE without model knowledge, a more challenging setting than the “equilibrium computation” setting we consider here. Thus, directly applying it will lead to worse iteration complexity than our equilibrium computation algorithms. As pointed out by the reviewer, Our designed algorithm based on OMWU indeed offers $\tilde{O}(1/\epsilon)$ convergence rate, distinguishing it from other algorithms for computing Markov CCE with the general structure. We will make these points clear in our updated version.
>
> **Minor issues.**
>
> 1. We will simplify the exposition in Sec. 3.1
>
> 2. Thanks for raising this very good point. Yes, we can use value-iteration to compute Markov stationary NE for the star-shaped case, for a fixed $\gamma$. The key to obtaining the fictitious-play property, as we pointed out in lines 350-367 in the Remark on page 8, is to *re-formulate* the NE computation in this case as a **minimax optimization** problem for each stage, and show the corresponding value-iteration operator is “contracting”. With this contracting property, one can readily show that VI can find the Markov NE efficiently. We did not emphasize this as it is folklore to obtain efficient computation using contracting VI operators. We will stress this point in our updated version.
> \\
> 3. Thank you very much for bringing up the reference. We will include the discussion in our updated version.
>
> 4. Thanks for the suggestion. We will fix this formatting issue.
>
> 5-8. Thanks for the careful reading. We will fix these typos.
>
> 9. We will add the definition of differential inclusion in the updated version.
>
> Thank you very much again for the valuable comments on improving our paper. We hope the responses have addressed them satisfactorily.

---

> > ### Comment · Reviewer_SApy · 2023-08-10
> >
> > I thank the authors for the detailed response.

---

### Decision · Program_Chairs · 2023-09-21

**Decision:**

Accept (poster)

**Comment:**

Accept with a large number of expert and clearly supporting reviews. The authors are encouraged to address the reviewers' suggestions in improving the final version of their paper.